# Transformer Key-Value Memories Are Nearly as Interpretable as Sparse Autoencoders

**Mengyu Ye**
Tohoku University
ye.mengyu.s1@dc.tohoku.ac.jp

**Jun Suzuki**
Tohoku University & RIKEN
jun.suzuki@tohoku.ac.jp

**Tatsuro Inaba**[*]
MBZUAI
tatsuro.inaba@mbzuai.ac.ae

**Tatsuki Kuribayashi**
MBZUAI
tatsuki.kuribayashi@mbzuai.ac.ae

## Abstract

Recent interpretability work on large language models (LLMs) has been increasingly dominated by a feature-discovery approach with the help of proxy modules. Then, the quality of features *learned* by, e.g., sparse auto-encoders (SAEs), is evaluated. This paradigm naturally raises a critical question: do such learned features have better properties than those *already represented* within the original model parameters, and unfortunately, only a few studies have made such comparisons systematically so far. In this work, we revisit the interpretability of feature vectors stored in feed-forward (FF) layers, given the perspective of FF as key-value memories, with modern interpretability benchmarks. Our extensive evaluation revealed that SAE and FFs exhibits a similar range of interpretability, although SAEs displayed an observable but minimal improvement in some aspects. Furthermore, in certain aspects, surprisingly, even vanilla FFs yielded better interpretability than the SAEs, and features discovered in SAEs and FFs diverged. These bring questions about the advantage of SAEs from both perspectives of feature quality and faithfulness, compared to directly interpreting FF feature vectors, and FF key-value parameters serve as a strong baseline in modern interpretability research[2].

## 1 Introduction

Transformer-based language models (LMs) have exhibited outstanding performance on a wide variety of tasks [10, 35, 1, 44, 45], whereas their underlying mechanisms remain opaque [47, 37, 50, 17, 38, 18, 34, 28, 31]. This issue has been tackled in the interpretability field, and in earlier days, the field has typically adopted a *top-down* approach, where, given candidate features or algorithms, e.g., syntactic structure, it has been inspected where in the original model those are encoded. Nowadays, as a variety of capabilities emerge in larger LMs, the question tends to be more on the *bottom-up*, feature-discovery side: what kind of features are encoded in the model?; and how can we discover and control them? This feature-discovery age has brought two trends to the interpretability community simultaneously: (i) training an external proxy module dedicated to this purpose, namely, sparse auto-encoder (SAEs), to decompose neuron activations into simpler basic features [53, 8, 24, 30, 21, 16] (**proxy-based analysis**), and (ii) developing new comprehensive interpretability benchmarks [36, 27] to test the quality of discovered features.

---

[*]Work done at Tohoku University.

[2]Project page: https://muyo8692.com/projects/ff-kv-sae

This paper explores one overlooked question in the field, to what extent a proxy-based, *artificial* decomposition of neuron activations empirically benefits the model interpretation. In other words, feed-forward (FF) layers naturally implement the decomposition of neural activation into a set of feature vectors, through the lens of FF as key-value memories [17](**FF-KV analysis**), why not first evaluate such *organic* features in FFs with the newly developed interpretability benchmarks? Proxy-based and FF-KV analysis have complementary advantages, and thus, there is no immediate reason to dismiss the latter. For example, while some proxy-based methods have a theoretical motivation to handle superposition, they also have limitations that FF-KV analysis can automatically bypass: proxy modules can additionally expose biases to the interpretation, e.g., specific features are repeatedly found [8, 49, 11]; the external proxy hallucinates features [22]; and additional computation costs are needed to interpret the model. Furthermore, the FF activations are reported to be naturally sparse even without any regularization [29]. Thus, if FF-KV and SAE analyses yield comparable results, there are several advantages (more simply put, from Occam's Razor principle) to adopting the former FF-KV analyses.

To gauge the (dis)similarities between FF-KVs' and SAEs' interpretability level, we perform both automatic evaluation and manual feature analyses. Automatic evaluation with SAEBENCH demonstrates surprising similarities between the two approaches. The evaluation scores fell into a similar range in all eight metrics in SAEBENCH, and the inter-metrics tendencies are also paralleled, e.g., causal intervention scores are poorer than feature disentanglement scores in both methods. One can even observe some advantages of FF-KVs; for example, features in the original FFs tend to avoid feature overlapping, resulting in better absorption scores [11] (i.e., less redundancy) than those in SAEs. These comparable quality is further supported by human manual evaluation of feature qualities. Conceptual features can be found with almost equal ease from both FF-KV features and SAEs. These tentatively conclude that features from FF-KVs and SAEs serve a quite similar level of interpretability from both quantitative and qualitative perspectives.

In our analysis, we further investigate the faithfulness of proxy-discovered features, considering FF-KV features as gold, how large is the overlap between the feature sets of the original FF-KV module and that of the proxy module? We analyzed such an overlap with Transcoder (TC), the closest counterpart to FF-KV, as a proxy model, and revealed that the majority of TC features do not have similar counterparts in the original FF module. This aligns with the existing report that SAEs can interpret even random Transformers [22], and perhaps the proxy module hallucinates new features rather than translating the workings of the original FF module, encouraging further research on the faithfulness of the learned features, with FF-KV features as grounding points. To sum up, our study reveals that proxy-based methods such as SAEs empirically offer very limited advantage over the direct analysis of FFs (i.e., key-value memories). That is, the theoretical advantage of SAEs is not observed empirically, at least through the lens of the current evaluation scheme, and encourages the inclusion of FF-KV features as a strong baseline when assessing feature-discovery methods in the interpretability field.

## 2 Background

### 2.1 Related Work

**Dictionary Learning and LLMs Interpretation.** Dictionary learning has been proposed to address polysemanticity of the representation [3, 52, 42, 40, 14, 5, 20], and this has been applied to interpret the internal activations of LLMs, represented by sparse autoencoders (SAEs) [53, 8, 24, 30, 21, 16, 25]. Specifically, these introduce a proxy module to decompose and reconstruct a model's activation, and seek interpretable features in it. Apparently, promising results were observed in earlier days: the learned features are highly interpretable and can be directly used to steer the model's behavior [46]: modification on a feature will either eliminate the corresponding behavior, or enhance it.

**Mixed Reports on SAE Features.** Although the SAEs get increasing attention, concurrent works have brought skeptical views on their success. For example, SAE-based feature steering quality is inferior to simple baselines utilizing activations [51]; SAEs can learn meaningful features even from a randomly initialized Transformer [22]; and they exhibit no clear advantage in downstream tasks and sometimes underperform linear probes that use the model's raw activations [9, 26]. This study, at a high level, provides additional support for such criticisms of the general advantage of SAEs.

**Interpretability of Feed-Forward (FF) Layers.** There have been a fair number of studies to interpret the feed-forward (FF) layer in Transformers directly [13, 40, 2]. The closest work to ours is Geva et al. [17], where FFs can be viewed as key-value memories, and they are interpretable and useful to control the model output. Recent work also indicates that activations in FFs are already sparse [29], and their neurons can be manipulated [51]; these motivate our work to contextualize the bare FF interpretability with SAE works.

## 2.2 Sparse Autoencoder for Transformer Interpretability

**Transformer.** Transformer architecture is a stack of multiple modules, such as attention mechanisms, feed-forward (FF) layers, and normalization layers. There have recently been increasing endeavors to interpret, especially, neuron activations around FF layers, such as SAEs. Henceforth, vector denotes a row vector.

**SAE.** SAE decomposes and reconstructs the neuron activations, typically after the FF layer (residual stream). That is, let $\boldsymbol{x}_{\mathrm{FF_{out}}} \in \mathbb{R}^{d_{\mathrm{model}}}$ be neuron activations after the FF layer, and $d_{\mathrm{SAE}}$ denotes the dimension of SAE features. SAE decomposes the neuron activations $\boldsymbol{x}_{\mathrm{FF_{out}}}$ and reconstructs it $\hat{\boldsymbol{x}}_{\mathrm{FF_{out}}}$ as follows:

$$\hat{\boldsymbol{x}}_{\mathrm{FF_{out}}} \approx \boxed{\mathrm{ReLU}(\boldsymbol{x}_{\mathrm{FF_{out}}} \boldsymbol{W}_{\mathrm{enc}} + \boldsymbol{b}_{\mathrm{enc}})} \ \boxed{\boldsymbol{W}_{\mathrm{dec}}} + \boldsymbol{b}_{\mathrm{dec}} \ , \quad (1)$$

with $\boldsymbol{W}_{\mathrm{enc}} \in \mathbb{R}^{d_{\mathrm{model}} \times d_{\mathrm{SAE}}}$, $\boldsymbol{W}_{\mathrm{dec}} \in \mathbb{R}^{d_{\mathrm{SAE}} \times d_{\mathrm{model}}}$, $\boldsymbol{b}_{\mathrm{enc}} \in \mathbb{R}^{d_{\mathrm{SAE}}}$, and $\boldsymbol{b}_{\mathrm{dec}} \in \mathbb{R}^{d_{\mathrm{model}}}$ in the SAE module. $\mathrm{ReLU}(\cdot) : \mathbb{R}^d \to \mathbb{R}^d$ denotes an element-wise ReLU activation. Each activation dimension is treated as a potentially interpretable feature, and the matrix maps each feature dimension to its feature vector in the representation space. This module is trained so that the activations are as sparse as possible with a sparsity loss to disentangle the potentially polysemantic input neurons.

**Transcoder.** Notably, as an alternative to SAEs and perhaps the closest attempt to this study, *Transcoders* have recently been proposed [12]. This approximates the original FF by training a sparse MLP as a proxy to predict FF output $\boldsymbol{x}_{\mathrm{FF_{out}}}$ from FF input $\boldsymbol{x}_{\mathrm{FF_{in}}}$, and its internal activations ($\in \mathbb{R}^{d_{\mathrm{TC}}}$) are evaluated in the same way as the standard SAE. Still, their work [12] did not clearly evaluate how interpretable the original FF's internal activations are, and this work complements this overlooked question.

## 2.3 Feed-Forward Layer as Key-Value Memories

Feed-forward layers in Transformers once project the FF input $\boldsymbol{x}_{\mathrm{FF_{in}}} \in \mathbb{R}^{d_{\mathrm{model}}}$ to $d_{\mathrm{FF}}$-dimensional representation ($d_{\mathrm{model}} < d_{\mathrm{FF}}$), applies an element-wise non-linear activation $\boldsymbol{\phi}(\cdot) : \mathbb{R}^d \to \mathbb{R}^d$, and projects it back, as follows:

$$\boldsymbol{x}_{\mathrm{FF_{out}}} = \boxed{\boldsymbol{\phi}(\boldsymbol{x}_{\mathrm{FF_{in}}} \boldsymbol{W}_K + \boldsymbol{b}_K)} \ \boxed{\boldsymbol{W}_V} + \boldsymbol{b}_V = \sum_{i \in d_{\mathrm{FF}}} \boxed{\boldsymbol{\phi}(\boldsymbol{x}_{\mathrm{FF_{in}}} \boldsymbol{W}_K)_{[i]}} \ \boxed{\boldsymbol{W}_{V[:,i]}} + \boldsymbol{b}_V \ , \quad (2)$$

where $\boldsymbol{W}_K \in \mathbb{R}^{d_{\mathrm{model}} \times d_{\mathrm{FF}}}$ and $\boldsymbol{W}_V \in \mathbb{R}^{d_{\mathrm{FF}} \times d_{\mathrm{model}}}$ are learnable weight matrices, and $\boldsymbol{b}_K \in \mathbb{R}^{d_{\mathrm{FF}}}$, $\boldsymbol{b}_V \in \mathbb{R}^{d_{\mathrm{model}}}$ are learnable biases. $d_{\mathrm{FF}}$ is typically set as $4d_{\mathrm{model}}$. One interpretation of the FF layer is a knowledge retrieval module; that is, the module first creates keys (activations) from an input $\boldsymbol{x}_{\mathrm{FF_{in}}}$ and then aggregates their associated values (feature vectors) . Existing studies have analyzed what kind of concept is stored in each feature vector of $\boldsymbol{W}_{V[:,i]}$ and when they are activated by $\boldsymbol{\phi}(\boldsymbol{x}_{\mathrm{FF_{in}}} \boldsymbol{W}_K)_{[i]}$ [17].

# 3 Comparing FF-KVs with SAEs

The feed-forward key-value memory module (FF-KV) inherently performs the same operation as SAEs (although it is somewhat obvious, given that both adopt the MLP architectures): it first decomposes the neuron activations into feature vectors and then aggregates them. This naturally raises a question about how similar the decomposition *naturally* made by FF-KVs is to that *learned* by the proxy module, e.g., SAEs. We examine several variants of FF-KV-based feature discovery methods[3].

---

[3]See Appendix A for the details on the implementations

### 3.1 Methods

**FF-KV.** The vanilla FF key-value memories are evaluated with the SAE evaluation framework, treating the key activations as features and the value vectors as feature vectors.

**TopK FF-KV.** To encourage the alignment with SAE research, we also introduce sparsity to activations in FFs by applying a top-$k$ activation function to the key vector, although it has been reported that the vanilla FFs' activations are somewhat already sparse [29]. This keeps only the $k$ neurons with the $k$ largest activations in each inference, zeroing out the activation for the rest. We call this **TopK FF-KV**, defined as follows:

$$\boldsymbol{x}_{\text{FF}_{\text{out}}} \approx \text{Top-}k(\phi(\boldsymbol{x}_{\text{FF}_{\text{in}}}\boldsymbol{W}_K + \boldsymbol{b}_K)) \ \boldsymbol{W}_V + \boldsymbol{b}_V \ . \tag{3}$$

**Normalized FF-KV.** The feature vectors of SAE are typically normalized, whereas those in FF are not. If a particular feature vector $\boldsymbol{W}_{V[i,:]}$ has a large norm, the magnitude of its corresponding activation may be underestimated. To handle this potential concern, we normalize each row of $\boldsymbol{W}_V$, and the discounted vector norm is weighted to activations. We refer to the method with this post-correction as **Normalized (TopK) FF-KV**:

$$\boldsymbol{x}_{\text{FF}_{\text{out}}} \approx \text{Top-}k(\phi(\boldsymbol{x}_{\text{FF}_{\text{in}}}\boldsymbol{W}_K + \boldsymbol{b}_K) \odot \boldsymbol{s}) \ \tilde{\boldsymbol{W}}_V + \boldsymbol{b}_V \ , \tag{4}$$

$$\boldsymbol{s} = [\|\boldsymbol{W}_{V[1,:]}\|_2, \|\boldsymbol{W}_{V[2,:]}\|_2, \cdots, \|\boldsymbol{W}_{V[d_{FF},:]}\|_2] \in \mathbb{R}^{d_{\text{FF}}} \ . \tag{5}$$

Here, $\tilde{\boldsymbol{W}}_V = \text{diag}(\boldsymbol{s})^{-1}\boldsymbol{W}_V$, where $\text{diag}(\cdot)$ expands a vector $\mathbb{R}^d$ to a diagonal matrix $\mathbb{R}^{d \times d}$.

### 3.2 Inference and Feature Discovery

Once a method to obtain activations from the models is determined, one can get an activation history over a certain set of text. Here, we introduce several notations before going to the experiments.

**Notations.** Feature activations are analyzed through feeding specific texts to models, and the exact text contents will vary depending on evaluation metrics. Let us denote a set of input texts as $\mathcal{S} = [\boldsymbol{s}_1, \cdots, \boldsymbol{s}_n]$, where each text consists of multiple tokens $\boldsymbol{s}_k = [w_{(k,1)}, w_{(k,2)}, \cdots, w_{(k,m)}] \in \mathcal{S}$, which are used to get feature activations. For brevity, we flatten and re-index the tokens as $[w_1, w_2, \cdots, w_l]$; one can recover the original indices $(i, j)$ indicating text id and token position via $\sigma : \mathbb{N}_{[1,l]} \to \mathbb{N}_{[1,n]} \times \mathbb{N}_{[1,m]}$, e.g., $\sigma(2) = (1, 2)$. For each token $w_t$, we first collect feature activations $\boldsymbol{a}_t \in \mathbb{R}^{d_{\text{coder}}}$ with a particular method, such as SAE. Here, $d_{\text{coder}}$ should be $d_{\text{SAE}}$, $d_{\text{TC}}$, or $d_{\text{FF}}$, depending on the methods; in other words, each method can maximally yield $d_{\text{coder}}$ numbers of features $\mathcal{F} = [f_1, \cdots, f_{d_{\text{coder}}}]$. Repeatedly collecting the activations over inputs $[w_1, \cdots, x_l]$ gives an activation history matrix $A \in \mathbb{R}^{l \times d_{\text{coder}}}$, where each row corresponds to each token $x_t$, and each column corresponds to each feature (neuron) $f_p \in \mathcal{F}$, respectively. $A_{[:,p]} \in \mathbb{R}^l$ presents where a feature $f_p$ was activated in $\mathcal{S}$. $\mathcal{S}_p = \{\boldsymbol{t}_i \in T \mid A_{t,p} > 0 \text{ and } i, \_ = \sigma(t)\} \subseteq \mathcal{S}$ represents the text subset associated with the feature $f_p$.

**SwiGLU activation.** Modern LMs adopt a SwiGLU gating function [41] for the non-linear activation of FFs $\phi(\cdot)$. The existing work [54] showed the compatibility of SwiGLU activation with FF-KV analysis, and thus, the above methods (§ 3.1) can be naturally implemented with SwiGLU. For example, on top of the SwiGLU activation, the TopK FF-KV can be written as follows:

$$\boldsymbol{x}_{\text{FF}_{\text{out}}} \approx \text{Top-}k((\boldsymbol{W}_G \boldsymbol{x}_{\text{FF}_{\text{in}}}) \odot \mathbf{Swish}(\boldsymbol{W}_K \boldsymbol{x}_{\text{FF}_{\text{in}}})) \ \boldsymbol{W}_V + \boldsymbol{b}_V \ , \tag{6}$$

$$= \sum_{i \in d_{\text{FF}}} \text{Top-}k((\boldsymbol{W}_G \boldsymbol{x}_{\text{FF}_{\text{in}}}) \odot \mathbf{Swish}(\boldsymbol{W}_K \boldsymbol{x}_{\text{FF}_{\text{in}}}))_{[i]} \ \boldsymbol{W}_{V[:,i]} + \boldsymbol{b}_V \ , \tag{7}$$

where, $\boldsymbol{W}_G \in \mathbb{R}^{d_{\text{FF}} \times d_{\text{model}}}$ is the gating matrix.

## 4  Experiment 1: Automatic Evaluation

We evaluate FF-KVs, SAEs, and Transcoders using the metrics from SAEBENCH [27]. We also report the Feature Alive Rate as a complementary statistic.

## 4.1 Evaluation Metrics

Here, we give a high-level description of each metric, and details are shown in Appendix B.

**Feature Alive Rate** aggregates how many features are alive, out of $d_{\text{coder}}$ features. A positive value of $A_{t,p}$ is regarded as the activation of $p$-th feature in $x_t$. An indicator function, $\chi : X_{[:,p]} \mapsto 1$ if $\max(X_{[:,p]}) > 0$ else 0, judges if the feature $f_p$ is *alive* (activated at least once) and the following score is calculated: $\frac{\sum_{j=1}^{d_{\text{coder}}} \chi(A_{[:,j]})}{d_{\text{coder}}}$. A score of 1 indicates that all features are activated at least once.

**Explained Variance** evaluates how well the proxy module reconstructs the original activations, and the FF-KV methods (without proxies) can automatically get a perfect score (=1) since this is the original module as is.

**Absorption Score** evaluates how many features a particular simple concept (e.g., word starting with "S") is split into. A higher value implies that many features are needed to emulate the targeted single concept, and thus, the feature set is redundant.

**Sparse Probing** evaluates the existence of specific informative features (e.g., sentiment) and their generalizability to held-out data. This is measured based on the accuracy of probe classifiers trained on the activation patterns to predict the properties of unseen inputs for the proxy module.

**Auto-Interpretation Score** evaluates how easily the activation patterns of the feature can be summarized in natural language (e.g., "a feature related to accounting"). Specifically, given a text subset $\mathcal{S}_p$ for the feature $f_p$,[4] an LLM is requested to summarize the feature concept and then predict the (binary) feature activation on the held-out set based on the summary, following Paulo et al. [36]. A score of 1 indicates that features can be perfectly summarized, and their activations are predictable.

**Spurious Correlation Removal (SCR)** evaluates how well spuriously correlated two features (e.g., gender and profession) are disentangled from different features, using the SHIFT [32] data. A score of 1 indicates a perfect disentanglement. Notably, its extension, Targeted Probe Perturbation (TPP) score, was also invented as a supplemental metric in SAEBENCH [27]. The TPP results are shown in Appendix and yielded consistent results with SCR. Notably, SCR and TPP consider top-$K$ activations in evaluations (we adopt $K = 20$ here, following existing studies [27])[5], and results for different $K$ are shown in Appendix B.

**RAVEL** [23] further evaluates the feature overlap and disentanglement, but slightly from a different angle from SCR and TPP. Specifically, this concerns the separability and controllability of multiple different attributes of the same entity (e.g., Japan $\xrightarrow{\text{continent}}$ Asia, Japan $\xrightarrow{\text{capital}}$ Tokyo). The RAVEL score can be decomposed into two complementary scores: (i) Isolation score—the probability that all non-edited attributes remain unchanged; and (ii) Causality score—the probability that the edit successfully changes the target attribute. We report both scores for the RAVEL results to clarify the fine-grained properties.

## 4.2 LMs and Proxy Modules

**LMs.** We evaluate FFs in all layers of five LMs: Gemma-2-2B, Gemma-2-9B [43], Llama-3.1-8B [45], GPT-2 [39], and Pythia-70M [4]. Due to space limitations, the results of the middle layers of Gemma-2-2B (layer 13) and Llama-3.1-8B (layer 16) are shown in the main part of this paper, and the results for other layers and models are shown in Appendix C. We also target randomly initialized LLMs as baselines, given the assertion that SAEs can even interpret randomly initialized Transformers. In our main experiments, we set $k = 10$ for the TopK FF-KV, and we additionally compare results under varying $k$ values.

**SAEs.** We use pretrained SAEs from Gemma Scope [30] (width 16k) for Gemma-2-2B and Gemma-2-9B, Llama Scope [21] (width 32k) for Llama-3.1-8B, and SAELens [7] for GPT-2. All SAEs are trained on FF outputs.

---

[4]It is also marked at which token in the text the feature was activated.

[5]We found SCR and TPP scores are highly unstable, and one may have to treat them as supplementary results. See Appendix B for details.

| Model | SAE Type | Coder Status | | Concept Detection | |
|---|---|---|---|---|---|
| | | Feat. Alive ↑ | Expl. Var. ↑ | Absorption ↓ | Sparse Prob. ↑ |
| Gemma-2 2B | SAE | $0.988_{\pm 0.000}$ | $0.699_{\pm 0.000}$ | $0.087_{\pm 0.173}$ | $0.846_{\pm 0.161}$ |
| | Transcoder | $1.000_{\pm 0.000}$ | $0.637_{\pm 0.000}$ | $0.025_{\pm 0.116}$ | $0.854_{\pm 0.149}$ |
| | FF-KV | $0.999_{\pm 0.000}$ | $1.000_{\pm 0.000}$ | $0.000_{\pm 0.001}$ | $0.827_{\pm 0.158}$ |
| | FF-KV (Norm.) | $1.000_{\pm 0.000}$ | $1.000_{\pm 0.000}$ | $0.000_{\pm 0.001}$ | $0.826_{\pm 0.160}$ |
| | TopK-FF-KV | $0.984_{\pm 0.000}$ | $0.160_{\pm 0.000}$ | $0.000_{\pm 0.001}$ | $0.768_{\pm 0.168}$ |
| | TopK-FF-KV (Norm.) | $0.984_{\pm 0.000}$ | $0.160_{\pm 0.000}$ | $0.000_{\pm 0.000}$ | $0.768_{\pm 0.168}$ |
| | Random Transformer | $1.000_{\pm 0.000}$ | $1.000_{\pm 0.000}$ | $0.007_{\pm 0.013}$ | $0.798_{\pm 0.067}$ |
| Llama-3.1 8B | SAE | $1.000_{\pm 0.000}$ | $0.594_{\pm 0.000}$ | $0.097_{\pm 0.332}$ | $0.879_{\pm 0.123}$ |
| | Transcoder | - | - | - | - |
| | FF-KV | $1.000_{\pm 0.000}$ | $1.000_{\pm 0.000}$ | $0.000_{\pm 0.001}$ | $0.876_{\pm 0.098}$ |
| | FF-KV (Norm.) | $1.000_{\pm 0.000}$ | $1.000_{\pm 0.000}$ | $0.000_{\pm 0.000}$ | $0.876_{\pm 0.098}$ |
| | TopK-FF-KV | $0.992_{\pm 0.000}$ | $0.238_{\pm 0.000}$ | $0.000_{\pm 0.001}$ | $0.832_{\pm 0.150}$ |
| | TopK-FF-KV (Norm.) | $0.992_{\pm 0.000}$ | $0.238_{\pm 0.000}$ | $0.000_{\pm 0.001}$ | $0.832_{\pm 0.150}$ |
| | Random Transformer | $1.000_{\pm 0.000}$ | $1.000_{\pm 0.000}$ | $0.002_{\pm 0.006}$ | $0.837_{\pm 0.084}$ |

| Model | SAE Type | Feature Explanation | Feature Disentanglement | | |
|---|---|---|---|---|---|
| | | Autointerp ↑ | RAVEL-ISO ↑ | RAVEL-CAU ↑ | SCR (k=20) ↑ |
| Gemma-2 2B | SAE | $0.782_{\pm 0.274}$ | $0.985_{\pm 0.027}$ | $0.002_{\pm 0.006}$ | $0.170_{\pm 0.191}$ |
| | Transcoder | $0.790_{\pm 0.270}$ | $0.940_{\pm 0.040}$ | $0.010_{\pm 0.017}$ | $0.104_{\pm 0.178}$ |
| | FF-KV | $0.710_{\pm 0.246}$ | $0.952_{\pm 0.035}$ | $0.012_{\pm 0.021}$ | $0.041_{\pm 0.094}$ |
| | FF-KV (Norm.) | $0.706_{\pm 0.255}$ | $0.952_{\pm 0.035}$ | $0.012_{\pm 0.021}$ | $0.041_{\pm 0.120}$ |
| | TopK-FF-KV | $0.772_{\pm 0.276}$ | $0.943_{\pm 0.039}$ | $0.009_{\pm 0.015}$ | $0.045_{\pm 0.105}$ |
| | TopK-FF-KV (Norm.) | $0.773_{\pm 0.269}$ | $0.942_{\pm 0.038}$ | $0.009_{\pm 0.014}$ | $0.029_{\pm 0.134}$ |
| | Random Transformer | $0.679_{\pm 0.248}$ | - | - | $0.004_{\pm 0.022}$ |
| Llama-3.1 8B | SAE | $0.817_{\pm 0.272}$ | $0.993_{\pm 0.016}$ | $0.002_{\pm 0.007}$ | $0.219_{\pm 0.323}$ |
| | Transcoder | - | - | - | - |
| | FF-KV | $0.751_{\pm 0.248}$ | $0.955_{\pm 0.044}$ | $0.007_{\pm 0.012}$ | $0.048_{\pm 0.070}$ |
| | FF-KV (Norm.) | $0.749_{\pm 0.245}$ | $0.954_{\pm 0.044}$ | $0.007_{\pm 0.012}$ | $0.046_{\pm 0.071}$ |
| | TopK-FF-KV | $0.807_{\pm 0.267}$ | $0.954_{\pm 0.044}$ | $0.006_{\pm 0.011}$ | $0.030_{\pm 0.045}$ |
| | TopK-FF-KV (Norm.) | $0.807_{\pm 0.256}$ | $0.955_{\pm 0.043}$ | $0.006_{\pm 0.010}$ | $0.029_{\pm 0.043}$ |
| | Random Transformer | $0.656_{\pm 0.237}$ | - | - | $0.053_{\pm 0.239}$ |

Table 1: Overview of the SAEBENCH evaluation results for the middle layer of Gemma-2-2B (layer 13) and Llama-3.1-8B (layer 16). Results are reported as mean $\pm 2$ standard errors of the mean over multiple seeds/settings. Norm. represent the normalized FF-KV. We also present the scores for a randomly initialized FF layers, which serve as the baseline. No substantial difference between FF-KV and SAEs is observed.

**Transcoders.** We use pretrained Transcoders (TCs) from Gemma Scope [30] for Gemma-2-2B and one from the original paper [12] for GPT-2, respectively.[6]

## 4.3 Results

**Overall.** Table 1 shows the results for each interpretability method. First of all, the SAE-based results and FF-KV results rendered similar tendencies. In each metric, the absolute difference between the scores from different methods is typically much smaller than seed/layer variance. In addition, the difficulty of each task (metric) is aligned across the tasks; for example, SAEs and FF-KVs achieved higher RAVEL-Isolation scores than RAVEL-Causality scores. These results suggest that, even with the activations in the original FF module, comparable interpretability can be realized compared to proxy-based methods, i.e., SAEs and Transcoders.

---

[6]See Appendix D for more details on the SAEs and Transcoders we use.

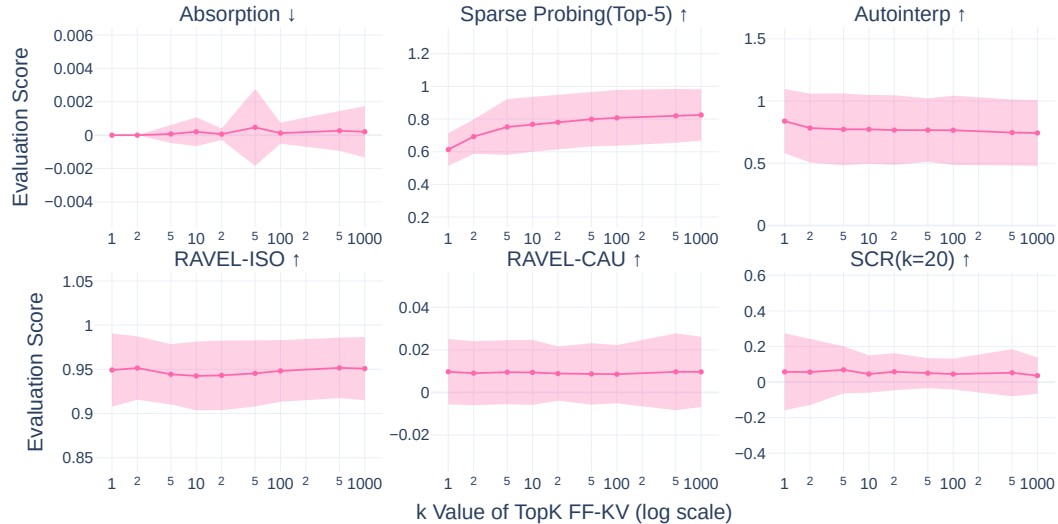

Figure 1: Evaluation scores for TopK FF-KVs at Layer 13 on Gemma-2-2B, under a different sparsity value $k$. A lower $k$ indicates a higher sparsity. Shaded areas denote $\pm 2$ standard errors of the mean, computed across multiple seeds and evaluation settings.

**Inter-Methods Similarity.** To mention specific similarity among the methods, causal intervention (RAVEL-Causality) was difficult for both SAEs and FF-KVs; that is, FF-KV methods inherit the limitation of SAEs. In contrast, feature isolation is well realized in FF-KVs, similarly to SAEs, even without any specific feature disentanglement regulation in FF-KVs, based on the high scores in RAVEL-isolation. Later layers tend to yield a high RAVEL-isolation score, and vice versa, especially in the case of FF-KVs. This shows a parallel with the existing observation that FFs in later layers have more semantic features [17], and attributes targeted in the RAVEL dataset might not be well-shaped in earlier layers.

**Inter-Methods Difference.** To highlight the differences among the methods, FF-KV methods can achieve perfect explained variance by definition (i.e., zero reconstruction loss as the original model is directly analyzed), whereas SAEs cannot. In addition, FF-KVs exhibited better absorption scores; that is, a simple single concept is not overly split into multiple concepts in FF-KVs than in SAEs, and in this sense, features in FF-KVs are less redundant. Sparse-probing results are comparable or slightly better in FF-KVs; representative features are encoded and generalizable to the same extent in both SAEs and FF-KVs. SAEs achieved slightly but consistently better Auto-interpretablity and SCP (although around zero) scores, which are only the advantage of SAEs compared to FF-KV-based analyses.

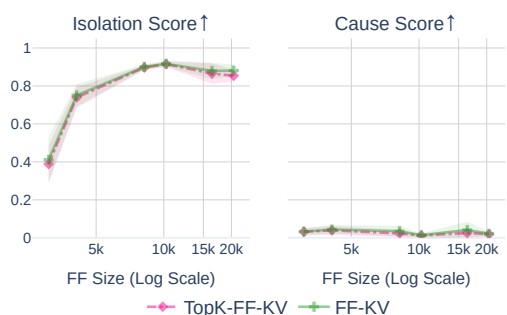

Figure 2: Relationship between FF hidden dimension size (model scale) and RAVEL scores.

**FF-KV Variants.** Within the FF-KV variants, TopK and normalization effects were generally small. Vanilla FF-KV features already exhibited a reasonable interpretability.

**TopK Effects.** Figure 1 shows the relationship between the $k$ value of the TopK FF-KV (x-axis) and the SAEBENCH evaluation scores (y-axis). The increase of sparsity level leads to inconsistent results, for example, a higher sparse probing (top-5) score but a lower autointerpretation score, suggesting that higher sparsity is not always better, at least for interpretation FF-KV.

**FF-Scaling Effects.**    Figure 2 shows the relationship between FF hidden representation size (model scale; x-axis) and RAEL interpretability scores (y-axis). These results suggest that FFs with a larger hidden dimension size do not always get better interpretability results, suggesting that just extending the hidden dimension size of FFs into that of SAEs does not lead to better interpretability. See Appendix E for other metrics.

## 5    Experiment 2: Human Evaluation

Our results in Section 4 suggest that FFs' internal activations have overall comparable interpretability to SAEs/Transcoders based on automatic evaluations. In this section, we further perform a follow-up manual inspection on the interpretability of features extracted from layer 13 of Gemma-2-2B's FF-KV, SAE, and Transcoder. We specifically explore the following questions: 1) Do features from the FF-KV, SAE, and Transcoder appear equally interpretable to humans? 2) How accurately can humans infer the origin of the feature?

### 5.1    Settings

We randomly sampled 50 features each from the FF-KV, TopK FF-KV, SAE, and Transcoder of Gemma-2-2B model, yielding a total of 200 features. Each feature $f_p$ is presented with its top-ten associated texts $\in \mathcal{S}_p$ based on the activation magnitude over a subset of OpenWebText corpus [19] (200M tokens in total). From the annotator's view, the presentation order of features is randomly shuffled, and their origins remain hidden throughout the experiment. The annotations in this section were conducted by one of the authors.

Table 2: Number of superficial, conceptual, and uninterpretable features.

| Coder | Superficial | Conceptual | Uninterp. |
|---|---|---|---|
| FF-KV | 6 | 8 | 36 |
| K-FF-KV | 9 | 9 | 32 |
| SAE | 6 | 9 | 35 |
| TC | 16 | 11 | 23 |

**Interpretability Evaluation.**    One annotator judges the qualitative quality of a feature using three categories: 1) *superficial Feature*: activates on shallow surface patterns (e.g., particular word, such as "the," punctuation, digits); 2) *Conceptual Feature*: activates on higher-level concepts spanning multiple tokens (e.g., sentiment, topics); or, 3) *Uninterpretable*: exhibits no clear activation pattern[7].

**Feature Origin Judgment.**    We also designed a task to predict from which module a feature originates, only based on the feature activation patterns in 10 texts, with the same data, to exploratorily find any difference between these activation patterns. If annotators can not guess which module is used to obtain the given feature, the used methods would have the same level of feature extraction ability. One annotator conducted this analysis, and as preliminary training, the annotator had first learned several activation patterns in the held-out set, paired with their module names.

Table 3: Origin judgment accuracies of features.

| Origin | Judging Acc. |
|---|---|
| FF-KV | 0.86 |
| K-FF-KV | 0.28 |
| SAE | 0.13 |
| TC | 0.18 |

### 5.2    Results

**Interpretability Evaluation.**    The results are presented in Table 2. First, the number of conceptual features is nearly the same across the four interpretability methods. In this sense, the quality of the obtained features is comparable. Transcoders could find a larger number of features that are interpretable (superficial or conceptual), but the ratio of superficial features is higher than in the other methods.

**Feature Origin Judgment.**    Table 3 shows the results. The annotator could not correctly predict the original model, except for the FF-KV methods. Through interviewing the annotator, we found that they could identify the FF-KV features by relying on superficial patterns in the magnitude and variance of feature activations (FF-KV tends to have a small value with high variance), rather than the represented concepts. Using TopK FF-KV (K-FF-KV) alleviates this distinction pattern, and thus,

---

[7]We provide the actual text we use to annotate in Appendix F.

if one wants to render a visualization of activations similar to that of SAEs, TopK FF-KV should be preferred. The low accuracies for K-FF-KV, SAE, and TC support that their discovered features and activations are similar to each other, as the human evaluator could not distinguish them.

## 6 Analysis: Feature Alignments

We analyze how similar features discovered by proxy methods, e.g., SAEs, are to the FF-KV ones.

### 6.1 Settings

To investigate the alignment of features from different interpretability methods, we specifically focus on those from FF-KV and Transcoder (not SAE, as the closest counterpart to FF-KV). In this analysis, we used layer 13 of Gemma-2-2B, which showed reasonable performance in the automatic evaluation experiment. Given an $r$-th feature vector in the FF $W_{V[:,r]}$, we find the index $u$ of the most aligned feature vector in $W_{\mathrm{dec}}$ from Transcoder, based on their cosine similarity: $u = \arg\max_k(W_{V[:,r]} \cdot W_{\mathrm{dec}[:,k]})$. Note that when analyzing features in TC, the searching direction will be opposite: $\arg\max_k(W_{\mathrm{dec}[:,r]} \cdot W_{V[:,k]})$. We call these max-cosine scores MCS.

We first perform a quick check of the correlation between MCS and the semantic feature alignment. For each MCS bin, we sampled ten feature pairs. Then, for feature pairs $(f_{p_1}, f_{p_2})$ within a specific MCS range, annotators manually inspected their alignment. Similarly to the previous experiment, each feature $f_p$ is accompanied by ten associated texts $\in \mathcal{S}_p$ from a subset of the OpenWebText corpus [19]. We consider a pair *matched* if these three criteria meet: 1) The two features generally represent the same concept (e.g., sentiment, topic); OR 2) The associated texts for the two features $(\mathcal{S}_{p_1}, \mathcal{S}_{p_2})$ exhibits $8/10$ overlap; OR 3) The topics of the texts from two features coincide. The annotations in this section were independently conducted by a different author from that of § 5.

### 6.2 Results

**Validity of Cosine-Based Alignments.** The results of alignment analysis are shown in Figure 3. This clearly shows that higher cosine similarity entails their semantic alignment. Based on these results, we tentatively regard a feature pair with cosine similarity above 0.9 as *aligned*, and the similarity below 0.3 as *unaligned* in the following analyses[8].

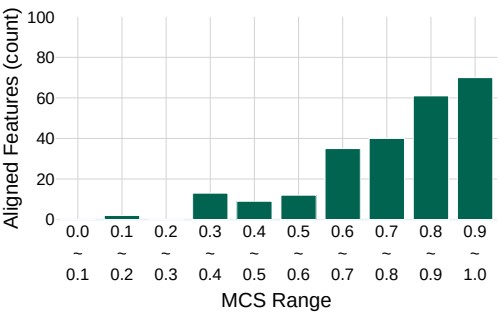

Figure 3: Histogram of aligned features numbers distribution for each MCS bin, between the FF and Transcoder.

**Large Number of Unaligned Features.** Based on the above criteria with cosine similarity, 41% (=3,780/9,216) and 66% (=10,835/16,384) features in FF-KV and Transcoder are unaligned with each other, respectively. In contrast, 5.7% (=527/9,216) and 3.2% (=527/16,384) features are regarded as aligned in FF-KV and Transcoder, respectively. That is, there are a large number of unaligned features between FF and Transcoder, clarifying that the same level of interpretability from different methods in automatic evaluation (§ 4) was not simply due to their similar feature sets. In the next paragraph, we manually analyze the unaligned features.

**Feature Complementarity.** We manually analyze three sets of features: (a) aligned features (FF-KV∩Transcoder), (b) FF-KV features not aligned with any Transcoder feature (FF-KV\Transcoder), and (c) Transcoder features not aligned with any FF-KV feature (Transcoder\FF-KV). The analysis target is the same as § 5; the features are classified into three categories of superficial, conceptual, and uninterpretable. Table 4 shows the distribution of feature categories in each feature set. First and interestingly, the unaligned features both in FFs and TC have a fair amount of conceptual ones. In particular, 32% of features that are unaligned with FFs were conceptual.

---

[8]See Appendix F for examples of the feature pairs we use for annotation as well as how we decide the threshold.

**Discussion.** Why are there so many unaligned features? One optimistic view is that TC successfully decomposed FF features into simpler ones, resulting in decomposed features being orthogonal to the original FF features, although this may offer a potential side effect of feature absorption, which is suggested by relatively large number of superficial features in TC\FF-KV and an already good absorption scores achieved by FF-KVs (Table 1). One more pessimistic view is that Transcoder invented completely new features that are not in the original FF-KV, which is in line with the fact that SAE can interpret even randomly initialized Transformers [22]. Our analysis alone can not fully distinguish between the two cases, but this fact of frequent misalignment deserves a motivation to further explore the faithfulness of the learned features in proxy modules. Features in the FF-KVs will serve as grounding points to evaluate such faithful evaluation, on top of our first extensive attention to FF-KVs in the context of SAE research.

Table 4: Number of superficial, multi-token conceptual, and uninterpretable features in aligned/unaligned features between FF (FF-KV) and Transcoders (TC).

| Coder | Superficial | Concept. | Uninterp. |
|---|---|---|---|
| FF-KV∩TC | 7 | 16 | 27 |
| FF-KV\TC | 1 | 8 | 41 |
| TC\FF-KV | 6 | 14 | 23 |

## 7 Conclusion

In this work, we revisit the interpretability of feature vectors *already represented* in the feed-forward (FF) module, as a strong baseline to SAEs. Our results show that the original FF feature vectors already exhibit reasonable interpretability comparable to that of sparse autoencoders (SAEs) and Transcoders on both comprehensive benchmark and human evaluations. We further demonstrate that a large portion of the features between the FF and the Transcoder are not aligned, and manual analysis suggests a potential feature over-splitting or hallucination of new features in the proxy module. To sum up, our results demonstrate that SAEs and Transcoders offer only limited advantages over the direct analysis of feed-forward key–value (FF-KV) representations. This finding highlights the lack of a strong and simple baseline within the interpretability community and underscores the importance of including FF-KV analysis as a fundamental reference point for evaluating interpretability methods. It also encourages future work to consider both model-internal parameters and proxy-module parameters when pursuing feature-discovery-based interpretability of large language models.

**Limitations.** The feature dimension of SAEs and Transcoders we used was fixed; more diverse configurations should be examined in the comparison, although publicly available pre-trained SAEs/Transcoders are limited, and prior work shows that simply scaling width does not necessarily improve SAEs and that there is not a universally best architecture choice [27]. Not all models are accompanied by Transcoder results: still, training Transcoders on all layers of, e.g., 9B-parameter models is prohibitively costly under an academic budget. We conducted only a few qualitative case studies on the effect of $k$ for TopK FF-KVs and on FF size. Although our analysis showed a discrepancy between FF-KV and Transcocder features, the interpretation of this difference (faithfulness of the learned features) remains unclear; future work should elaborate on this point.

**Impact Statement.** Our findings indicate that SAEs and Transcoders do not consistently outperform the original feature vectors in FFs with respect to interpretability. We underscore the need to reassess both the interpretability and, potentially, the reproducibility of the previously reported advantages of SAEs. In a sense, our study supports the use of inherently black-box neural LLMs while setting aside the interpretability issue, as their FFs appear to possess a certain degree of interpretability. Nevertheless, one of the ultimate objectives of this line of research should remain the development of models that are interpretable by design.

**Ethics Statement.** Our research primarily relied on publicly available models and datasets, and strictly adhered to their respective licenses (see Table 7). For human evaluation, we collected data as described in § 5.1. The data were collected with participant consent, and we ensured that responses were anonymized to prevent them from being traced back to individuals. To promote transparency and reproducibility, we have made the collected data, along with all code used in our experiments, publicly accessible. Comprehensive details of our experimental setup are provided in each section and the appendix to ensure reproducibility.

## Acknowledgment

**Author Contribution.** M. Ye led the research project, implemented the experimental pipeline, conducted the SAEBENCH evaluation, designed the human evaluation task, and carried out one of the human evaluation studies. T. Kuribayashi initially proposed the idea of comparing the transcoder with FF-KV augmented by a TopK activation function and was deeply involved in regular discussions throughout the project. T. Inaba provided valuable feedback on implementation, conducted one of the human evaluation studies, and actively contributed to project discussions. J. Suzuki provided overarching guidance and feedback at all stages of the project as well as computational resources.

M. Ye drafted the initial version of the manuscript. T. Kuribayashi offered extensive feedback and revisions on the writing. T. Inaba authored the impact and ethics statements. J. Suzuki contributed valuable insights into the overall framing and positioning of the paper.

**Acknowledgments.** We want to express our gratitude to the members of the Tohoku NLP Group for their insightful comments. And special thanks to Charles Spencer James for assisting in determining the MCS threshold for text alignment through human annotation. This work was supported by the JSPS KAKENHI Grant Number JP24H00727, JP25KJ06300; JST Moonshot R&D Grant Number JPMJMS2011-35 (fundamental research); JST BOOST, Japan Grant Number JPMJBS2421.

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

# A Implementation Details on FF-KVs

## A.1 Overall Framework

We implement FF-KVs use the custom_sae class provided by SAEBENCH [7]. To faithfully reproduce the activation of the FF sublayers, we apply the hook-based approach. The *encode* method simulates the FF's forward pass up to its neuron activations. It takes an input tensor **x**, injects it at the FF's input hook point, and captures the subsequent neuron activations using another hook. Conversely, the *decode* method simulates the FF's transformation from its neuron activations to its output. It accepts a tensor of neuron activations, injects them at the corresponding hook point, and captures the FF's final output. A *forward* method is also provided, performing the full pass through the FF block using hooks to inject input and capture the final output. This framework allows for the direct examination of an FF's feature extraction and signal reconstruction capabilities as if it were an SAE, providing a unique lens for interpreting learned representations within large language models.

## A.2 FF-KV Implement Details

The core principle is to map the FF's operations to the conceptual stages of an SAE:

**Input.**  Activations entering the FF block serve as the input to our pseudo-SAE.

**Feature Representation (Encoding).**  The FF's internal neuron activations, captured after the non-linear activation function and any gating, are interpreted as the SAE's latent features. The dimensionality of this feature space is equivalent to the FF's hidden dimension. The effective "encoder weights" are the FF's input weights.

**Reconstruction (Decoding).**  The FF's output, which is typically added to the residual stream, is considered the reconstructed input. The effective "decoder weights" and "decoder bias" are the FF's output weights and output bias, respectively.

## A.3 TopK FF-KV Implement Details

The TopK FF-KV extends the FF-KV framework by enforcing a strict TopK sparsity on the intermediate feature representations. The only modification from a FF-KV resides in the *encode* method. After obtaining the FF's internal neuron activations, a k-sparsity constraint is applied. For each input position (e.g., token in a sequence), the method identifies the $k$ neuron activations with the largest absolute values. All other neuron activations for that position are set to zero. This results in a feature vector where, at most, $k$ dimensions are non-zero.

## A.4 Normalized FF-KV Implement Details

The FF block's original output weights (which serve as the "decoder weights") are $L_2$-normalized along their feature dimension. The original norms of these weight vectors are stored. The encode and forward methods remain unchanged from their respective base FF-KV implementations. The key difference lies in the decode method. Before applying the normalized "decoder weights", the input feature activations are first scaled by the stored original norms. This step ensures that the magnitude of the reconstructed output appropriately reflects the original scaling of the FF block's output projections, despite the normalization. For models that include a final normalization step after the FF output (e.g., Gemma-2 Models), this step is also applied to maintain fidelity with the original model's computation.

## A.5 Transcoder Implement Details

We load the Transcoder weight into the JumpReLU class of SAEBENCH. While evaluating it, we follow the instructions of Gemma-Scope paper [30] to load model weights with folding applied.

# B  Details on Metrics

We provide detailed definitions of the metrics used in our main results, alongside the experimental settings for each metric.

**Feature Alive Rate.**  This metric belongs to the "core" evaluations in SAEBENCH. It counts how many features are alive out of the $d_{\mathrm{coder}}$ total features. A feature is deemed active when its activation exceeds 0. The evaluation is conducted on a 4 M-token subset of OpenWebText [19].

The metric is especially relevant for TopK FF-KVs employing a TOPK activation, ensuring that the mechanism does not repeatedly select only a small subset of neurons.

**Explained Variance.**  Also in the "core" suite, this metric is computed on a 0.4 M-token subset of OpenWebText [19].

**Absorption Score.**  Feature absorption [11] stems from *feature splitting* [8, 49], in which newly uncovered features become overly specific. A concrete example is a feature that activates only on "U.S. cities except New York and Los Angeles."

The metric targets a first-letter classification task, measuring situations where the main feature for a letter fails to fully capture the concept of "first letter", and other features compensate. Specifically, it evaluates all 26 letters with the prompt "*{word}* has the first letter:".

Given primary features $S_{\mathrm{main}}$ (e.g., selected via sparse probing) and auxiliary features $S_{\mathrm{abs}}$, the absorption score for one input is

$$\mathrm{Absorption} = \frac{\sum_{i \in S_{\mathrm{abs}}} a_i\, d_i \cdot p}{\sum_{i \in S_{\mathrm{abs}}} a_i\, d_i \cdot p + \sum_{i \in S_{\mathrm{main}}} a_i\, d_i \cdot p},$$

where $a_i$ is the activation, $d_i$ the unit decoder direction, and $p$ the ground-truth probe direction. We use the default hyperparameters.

**Sparse Probing.**  Sparse probing, introduced by Gurnee et al. [20], evaluates the alignment between individual features and a prespecified concept $c$. It has a hyperparameter $K$ that specifies how many features are used when training the probe. For each feature $h_j$,

$$s_j = \left| \mathbb{E}_{x \in \mathcal{X}_+}[h_j(x)] - \mathbb{E}_{x \in \mathcal{X}_-}[h_j(x)] \right|, \tag{8}$$

where $\mathcal{X}_+$ and $\mathcal{X}_-$ denote inputs with and without $c$, respectively. The top $K$ features by $s_j$ serve as inputs to a logistic-regression probe; the probe's test accuracy constitutes the sparse-probing score. We again employ the default hyperparameters.

**Auto-Interpretation Score.**  This evaluation has two phases: generation and scoring. In the generation phase, it obtain SAE activations, annotate each token with its activation value for the feature under consideration, and prompt an LLM to generate explanations based on these annotated activation patterns. The scoring phase constructs a test set for each feature containing 14 examples, exactly two of which are activated texts. The LLM must label each of the 14 texts as activated or not; the resulting prediction accuracy yields the auto-interpretation score.

The dataset is a subset of the copyright-free version of the Pile [15] (monology/pile-uncopyrighted), comprising 2 M tokens. GPT-4o [35] is used both to generate explanations and to predict activations.

**SCR and TPP**  Following SHIFT [32], the **SCR** evaluation proceeds as follows. A baseline classifier $C_{\mathrm{base}}$ is trained on data containing both true and spurious correlations. We then zero–ablate the $K$ features most attributable to the spurious signal and re-measure accuracy on a balanced set:

$$\mathrm{SCR} = \frac{A_{\mathrm{abl}} - A_{\mathrm{base}}}{A_{\mathrm{oracle}} - A_{\mathrm{base}}},$$

where $A_{\mathrm{abl}}$ is the post-ablation accuracy, $A_{\mathrm{base}}$ the baseline accuracy, and $A_{\mathrm{oracle}}$ the accuracy of an oracle probe trained on the true concept.

**TPP.** We extend Targeted Probe Perturbation to the multi-class setting. For $m$ classes, let $C_j$ be a linear probe that classifies concept $c_j$ with accuracy $A_j$. Let $A_{i,j}$ denote the accuracy of probe $C_j$ after ablating the $K$ most contributive features for class $c_i$. The TPP score is then

$$\text{TPP} = \frac{1}{m}\sum_{i=1}^{m}\big(A_{i,i} - A_i\big) - \frac{1}{m(m-1)}\sum_{i \neq j}\big(A_{i,j} - A_j\big),$$

so higher SCR and TPP values indicate stronger disentanglement.

**Stability Caveats.** Both metrics are highly sensitive to the choice of $K$. Across different $K$ values, SCR can range from below $0.1$ to above $0.4$. For TPP the variation is even larger: for the same coder, scores span from under $0.1$ (SAE, $K = 2$, Figure 14) to over $0.4$ (SAE, $K = 50$, Figure 18). Error bands obtained from multiple sub-runs are also wide—not only for SAEs (e.g., SAE on Llama-3.1 in Figure 14, and on Pythia in Figure 15) but likewise for FF-KVs (e.g., Pythia in Figure 14 and Figure 15).

Based on these empirical observations, we interpret SCR and TPP scores with caution.

**RAVEL.** Unlike SCR and TPP, we find RAVEL to be consistent across multiple models and coders, and the results align with the scores reported in the original work [23]. This stability suggests that RAVEL is comparatively insensitive to hyperparameter choices and dataset splits, making it a reliable baseline when assessing disentanglement. Accordingly, we place greater weight on RAVEL when synthesizing conclusions across metrics.

## C   Detailed Results on SAEBENCH

We provide detailed evaluation result with error bars indicate 95% confidence intervals ($\pm 2$ SEM), compute as $\text{SEM} = \sqrt{\frac{\sum(x_i - \bar{x})^2}{n(n-1)}}$ where n is the number of runs on different datasets for each metric in each layer. Note that error bars are not applicable for the feature alive metric, as they are counts for features activated at least once.

### C.1   Detailed Results on More Models

Figures 5, 6, 10, 13, 7, 8, 9, and 17 present the detailed results for all models.

### C.2   Detailed Results on Various Hyperparameter Choices

For metrics that have multiple hyperparameter choices for $k$, we provide detailed results for all tested hyperparameters.

- For SCR, the available $k$ values are $2, 5, 10, 20, 50, 100$, and $500$; the corresponding results are shown in Figures 14, 15, 16, 17, 18, 19, and 20.
- For TPP, the available $k$ values are the same, and all results are shown in Figures 21, 22, 23, 24, 25, 26, and 27.
- For Sparse Probing, the available $k$ values are $1, 2$, and $5$; the results are shown in Figures 11, 12, and 13.

## D   Details on SAEs/Transcoders used

For both SAEs and Transcoders from Gemma-Scope, we use the *canonical* versions, whose average $L_0$ sparsity is close to $100$, which are believed to be reasonably useful[9]. The SAEs are loaded through SAELens [7] with the following keys: "gemma-scope-2b-pt-mlp-canonical" for Gemma-2-2B, "gemma-scope-9b-pt-mlp-canonical" for Gemma-2-9B, and "llama_scope_lxm_8x" for Llama-3.1-8B.

For Transcoders, since no canonical versions have been explicitly defined and the SAELens release we use does not yet support loading them, we manually select checkpoints from the Gemma-Scope

---

[9]This statement can be found on Gemma Scope's collection page on HuggingFace (link).

| Layer | ID |
| --- | --- |
| 0 | layer_0/width_16k/average_l0_115/params.npz |
| 1 | layer_1/width_16k/average_l0_104/params.npz |
| 2 | layer_2/width_16k/average_l0_87/params.npz |
| 3 | layer_3/width_16k/average_l0_96/params.npz |
| 4 | layer_4/width_16k/average_l0_88/params.npz |
| 5 | layer_5/width_16k/average_l0_87/params.npz |
| 6 | layer_6/width_16k/average_l0_95/params.npz |
| 7 | layer_7/width_16k/average_l0_70/params.npz |
| 8 | layer_8/width_16k/average_l0_92/params.npz |
| 9 | layer_9/width_16k/average_l0_72/params.npz |
| 10 | layer_10/width_16k/average_l0_88/params.npz |
| 11 | layer_11/width_16k/average_l0_108/params.npz |
| 12 | layer_12/width_16k/average_l0_111/params.npz |
| 13 | layer_13/width_16k/average_l0_89/params.npz |
| 14 | layer_14/width_16k/average_l0_81/params.npz |
| 15 | layer_15/width_16k/average_l0_78/params.npz |
| 16 | layer_16/width_16k/average_l0_87/params.npz |
| 17 | layer_17/width_16k/average_l0_112/params.npz |
| 18 | layer_18/width_16k/average_l0_99/params.npz |
| 19 | layer_19/width_16k/average_l0_89/params.npz |
| 20 | layer_20/width_16k/average_l0_88/params.npz |
| 21 | layer_21/width_16k/average_l0_102/params.npz |
| 22 | layer_22/width_16k/average_l0_117/params.npz |
| 23 | layer_23/width_16k/average_l0_116/params.npz |
| 24 | layer_24/width_16k/average_l0_96/params.npz |
| 25 | layer_25/width_16k/average_l0_110/params.npz |

Table 5: Mapping of layers to their corresponding IDs

collection and download the corresponding weights from HuggingFace (link). These checkpoints are chosen according to the same criteria as the *canonical* SAEs, and their exact filenames are listed in Table 5. We also directly download the weight from the Transcoder proposal work [12] on HuggingFace (Link). To the best of our knowledge, there are no Transcoders publicly available for Gemma-2-9B and Llama-3.1-8B, and Pythia-70M.

# E    Detailed Results on FF Scaling

Table 6 shows the results on all metrics regarding to various FF intermediate sizes. Scores are not showing noticeable improvement except for RAVEL and absorption. Trends shown in SCR somehow understandable: these metrics highly depend on the ground truth probing performance, which is not always stable. Sparse probing result is also understandable, since sparse probing on FF from a random transfer can achieve a reasonable score, the probs can learn unintended signal in the dataset, rather than the true feature.

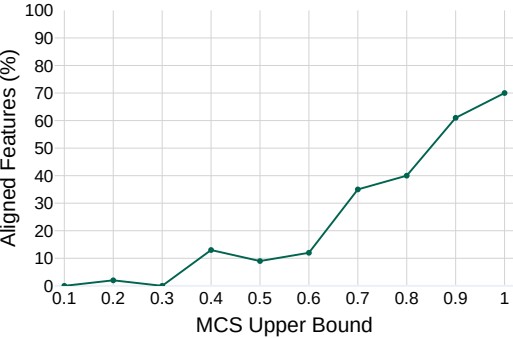

Figure 4: Proportion of aligned features as a function of the max-cosine score (MCS).

# F    Feature Examples

We provide example features for each annotation and coder, visualizing the top input examples that most strongly activate each feature. All features are extracted from layer 13 of Gemma-2-2B. To analyze the relationship between alignment and the max-cosine score (MCS), we divide the MCS

Table 6: Evaluation scores for different size of Pythia models' FF and TopK FF-KVs.

| FF Size | SAE Type | Coder Status | | Concept Detection | |
|---|---|---|---|---|---|
| | | Feat. Alive ↑ | Expl. Var. ↑ | Absorption ↓ | Sparse Prob. ↑ |
| 2048 | FF-KV | $1.000_{\pm0.000}$ | $1.000_{\pm0.000}$ | $0.060_{\pm0.083}$ | $0.802_{\pm0.173}$ |
| | TopK-FFKV | $1.000_{\pm0.000}$ | $0.227_{\pm0.000}$ | $0.064_{\pm0.127}$ | $0.717_{\pm0.204}$ |
| 3072 | FF-KV | $1.000_{\pm0.000}$ | $1.000_{\pm0.000}$ | $0.013_{\pm0.038}$ | $0.826_{\pm0.140}$ |
| | TopK-FFKV | $0.999_{\pm0.000}$ | $0.129_{\pm0.000}$ | $0.014_{\pm0.036}$ | $0.779_{\pm0.153}$ |
| 4096 | FF-KV | $1.000_{\pm0.000}$ | $1.000_{\pm0.000}$ | $0.003_{\pm0.007}$ | $0.803_{\pm0.128}$ |
| | TopK-FFKV | $1.000_{\pm0.000}$ | $0.082_{\pm0.000}$ | $0.006_{\pm0.011}$ | $0.765_{\pm0.126}$ |
| 8192 | FF-KV | $1.000_{\pm0.000}$ | $1.000_{\pm0.000}$ | $0.003_{\pm0.009}$ | $0.812_{\pm0.194}$ |
| | TopK-FFKV | $1.000_{\pm0.000}$ | $0.277_{\pm0.000}$ | $0.003_{\pm0.009}$ | $0.770_{\pm0.186}$ |
| 10240 | FF-KV | $1.000_{\pm0.000}$ | $1.000_{\pm0.000}$ | $0.001_{\pm0.004}$ | $0.850_{\pm0.117}$ |
| | TopK-FFKV | $1.000_{\pm0.000}$ | $0.090_{\pm0.000}$ | $0.002_{\pm0.009}$ | $0.783_{\pm0.144}$ |
| 16384 | FF-KV | $1.000_{\pm0.000}$ | $1.000_{\pm0.000}$ | $0.001_{\pm0.003}$ | $0.870_{\pm0.119}$ |
| | TopK-FFKV | $1.000_{\pm0.000}$ | $0.047_{\pm0.000}$ | $0.001_{\pm0.004}$ | $0.807_{\pm0.155}$ |
| 20480 | FF-KV | $1.000_{\pm0.000}$ | $1.000_{\pm0.000}$ | $0.002_{\pm0.005}$ | $0.818_{\pm0.164}$ |
| | TopK-FFKV | $1.000_{\pm0.000}$ | $0.195_{\pm0.000}$ | $0.000_{\pm0.001}$ | $0.735_{\pm0.157}$ |

| FF Size | SAE Type | Feature Explanation | Feature Disentanglement | | |
|---|---|---|---|---|---|
| | | Autointerp ↑ | RAVEL-ISO ↑ | RAVEL-CAU ↓ | SCR (k=20) ↑ |
| 2048 | FF-KV | $0.727_{\pm0.256}$ | - | - | $-0.056_{\pm0.464}$ |
| | TopK-FFKV | $0.766_{\pm0.277}$ | - | - | $0.000_{\pm0.131}$ |
| 3072 | FF-KV | $0.734_{\pm0.252}$ | $0.411_{\pm0.272}$ | $0.033_{\pm0.043}$ | $0.093_{\pm0.195}$ |
| | TopK-FFKV | $0.731_{\pm0.271}$ | $0.389_{\pm0.218}$ | $0.032_{\pm0.043}$ | $0.017_{\pm0.114}$ |
| 4096 | FF-KV | $0.708_{\pm0.252}$ | $0.750_{\pm0.132}$ | $0.044_{\pm0.051}$ | $0.017_{\pm0.054}$ |
| | TopK-FFKV | $0.716_{\pm0.263}$ | $0.739_{\pm0.123}$ | $0.039_{\pm0.053}$ | $-0.001_{\pm0.028}$ |
| 8192 | FF-KV | $0.714_{\pm0.256}$ | $0.900_{\pm0.032}$ | $0.035_{\pm0.064}$ | $0.047_{\pm0.099}$ |
| | TopK-FFKV | $0.712_{\pm0.271}$ | $0.897_{\pm0.031}$ | $0.023_{\pm0.040}$ | $0.016_{\pm0.038}$ |
| 10240 | FF-KV | $0.707_{\pm0.259}$ | $0.916_{\pm0.037}$ | $0.013_{\pm0.025}$ | $0.049_{\pm0.130}$ |
| | TopK-FFKV | $0.704_{\pm0.278}$ | $0.915_{\pm0.033}$ | $0.013_{\pm0.026}$ | $-0.017_{\pm0.065}$ |
| 16384 | FF-KV | $0.702_{\pm0.254}$ | $0.879_{\pm0.095}$ | $0.041_{\pm0.098}$ | $0.023_{\pm0.056}$ |
| | TopK-FFKV | $0.701_{\pm0.265}$ | $0.865_{\pm0.122}$ | $0.025_{\pm0.046}$ | $0.003_{\pm0.018}$ |
| 20480 | FF-KV | $0.693_{\pm0.252}$ | $0.881_{\pm0.072}$ | $0.021_{\pm0.031}$ | $0.030_{\pm0.050}$ |
| | TopK-FFKV | $0.698_{\pm0.270}$ | $0.854_{\pm0.069}$ | $0.019_{\pm0.028}$ | $0.022_{\pm0.064}$ |

range into ten bins (e.g., 0.1–0.2, 0.2–0.3). From each bin, we sample ten features, each associated with ten pairs of input examples (for a total of 100 examples per bin), and annotate the proportion of aligned features within each bin. As shown in Figure 4, features with an MCS below 0.3 are almost never aligned, whereas those above 0.9 exhibit over 60% alignment.

## F.1 Superficial Features

We show examples of "superficial" features here.

- **Figure 29** shows the first FF-KV feature we annotate as superficial, activating on "now".

- **Figure 30** shows the first TopK FF-KV feature we annotate as superficial, focused on "the".

- **Figure 31** shows the first SAE feature we annotate as superficial, activating on "return" in code.

- **Figure 32** shows the first Transcoder feature we annotate as superficial, activating on alphanumeric token combinations.

## F.2 Conceptual Features

We illustrate features that activate on higher-level concepts or semantic themes.

- **Figure 33** shows the first FF-KV feature we annotate as conceptual, activating on coastal concepts.
- **Figure 34** shows the first TopK FF-KV feature we annotate as conceptual, linked to recipes and desserts.
- **Figure 35** shows the first SAE feature we annotate as conceptual, activating on country and region names.
- **Figure 36** shows the first Transcoder feature we annotate as conceptual, activating on college degree concepts.

## F.3 Uninterpretable Features

We also show examples of features without clear patterns.

- **Figure 37** shows the first FF-KV feature we annotate as uninterpretable.
- **Figure 38** shows the first TopK FF-KV feature we annotate as uninterpretable.
- **Figure 39** shows the first SAE feature we annotate as uninterpretable.
- **Figure 40** shows the first Transcoder feature we annotate as uninterpretable.

## F.4 Aligned Features

**Figure 41** shows the first FF-KV feature we annotate as uninterpretable.

## F.5 Unaligned Features

**Figure 42** shows the first FF-KV feature we annotate as uninterpretable.

Table 7: The list of assets used in this work.

| Asset Type | Asset Name | Link | License | Citation |
|---|---|---|---|---|
| Code | SAEBench | ⌂ | Not specified | [27] |
| Code | TransformerLens | ⌂ | MIT License | [33] |
| Code | SAELens | ⌂ | MIT License | [7] |
| Dataset | OpenWebText | Link | CC0 1.0 Universal | [19] |
| SAE | Gemma-Scope-2B-pt-mlp | google/gemma-scope-2b-pt-mlp | Apache 2.0 | [30] |
| SAE | Gemma-Scope-9B-pt-mlp | google/gemma-scope-9b-pt-mlp | Apache 2.0 | [30] |
| SAE | Llama-Scope-3.1-8B-LXM-8x | fnlp/Llama3_1-8B-Base-LXM-8x | Not specified | [21] |
| SAE | GPT2-Small-32k-mlp-out | jbloom/GPT2-Small-OAI-v5-32k-mlp-out-SAEs | Not specified | [6] |
| SAE | Pythia-70m-deduped-mlp | ctigges/pythia-70m-deduped__mlp-sm_processed | Not specified | [48] |
| Transcoder | Gemma-Scope-2B-pt-transcoders | google/gemma-scope-2b-pt-transcoders | Apache 2.0 | [30] |
| Model | GPT-2-small | openai-community/gpt2 | MIT License | [39] |
| Model | Gemma-2-2B | google/gemma-2-2b | Gemma License | [43] |
| Model | Gemma-2-9B | google/gemma-2-9b | Gemma License | [43] |
| Model | Llama-3.1-8B | meta-llama/Llama-3.1-8B | Llama 3 Community License | [45] |
| Model | Pythia-70M-deduped | EleutherAI/pythia-70m-deduped | Apache 2.0 | [4] |
| Model | Pythia-160M-deduped | EleutherAI/pythia-160m-deduped | Apache 2.0 | [4] |
| Model | Pythia-410M-deduped | EleutherAI/pythia-410m-deduped | Apache 2.0 | [4] |
| Model | Pythia-1.4B-deduped | EleutherAI/pythia-1.4b-deduped | Apache 2.0 | [4] |
| Model | Pythia-2.8B-deduped | EleutherAI/pythia-2.8B-deduped | Apache 2.0 | [4] |
| Model | Pythia-6.9B-deduped | EleutherAI/pythia-6.9B-deduped | Apache 2.0 | [4] |
| Model | Pythia-12B-deduped | EleutherAI/pythia-12B-deduped | Apache 2.0 | [4] |

# G   Use of Existing Assets

Table 7 shows the assets being used in this paper, with the type, name, link, license, and citation for each asset used in the paper.

# H   Compute Statement

Most experiments presented in this paper were run on a cluster consisting of the NVIDIA H200 GPUs with 141GB of memory. All experiments on models are run using a single 141GB memory GPU. Evaluation time per layer differs largely on model size, with Pythia-70M, it takes approximately 1 hour, and for larger models like Gemma-2-9B, it takes approximately 4 hours per layer. The total GPU time for this work is approximately 1400 hours, including exploratory research stage.

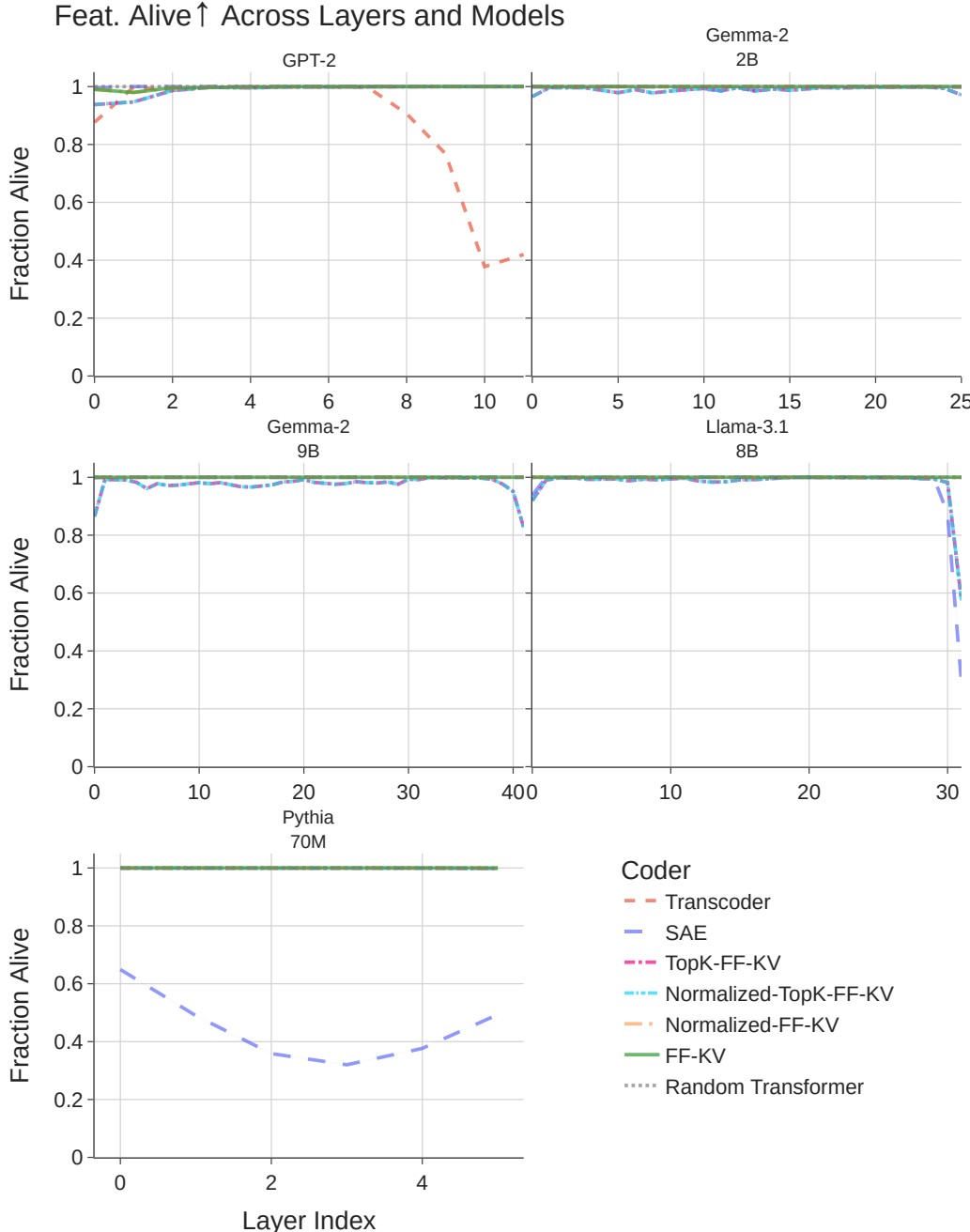

Figure 5: Detailed feature alive scores on all tested models, across all layers.

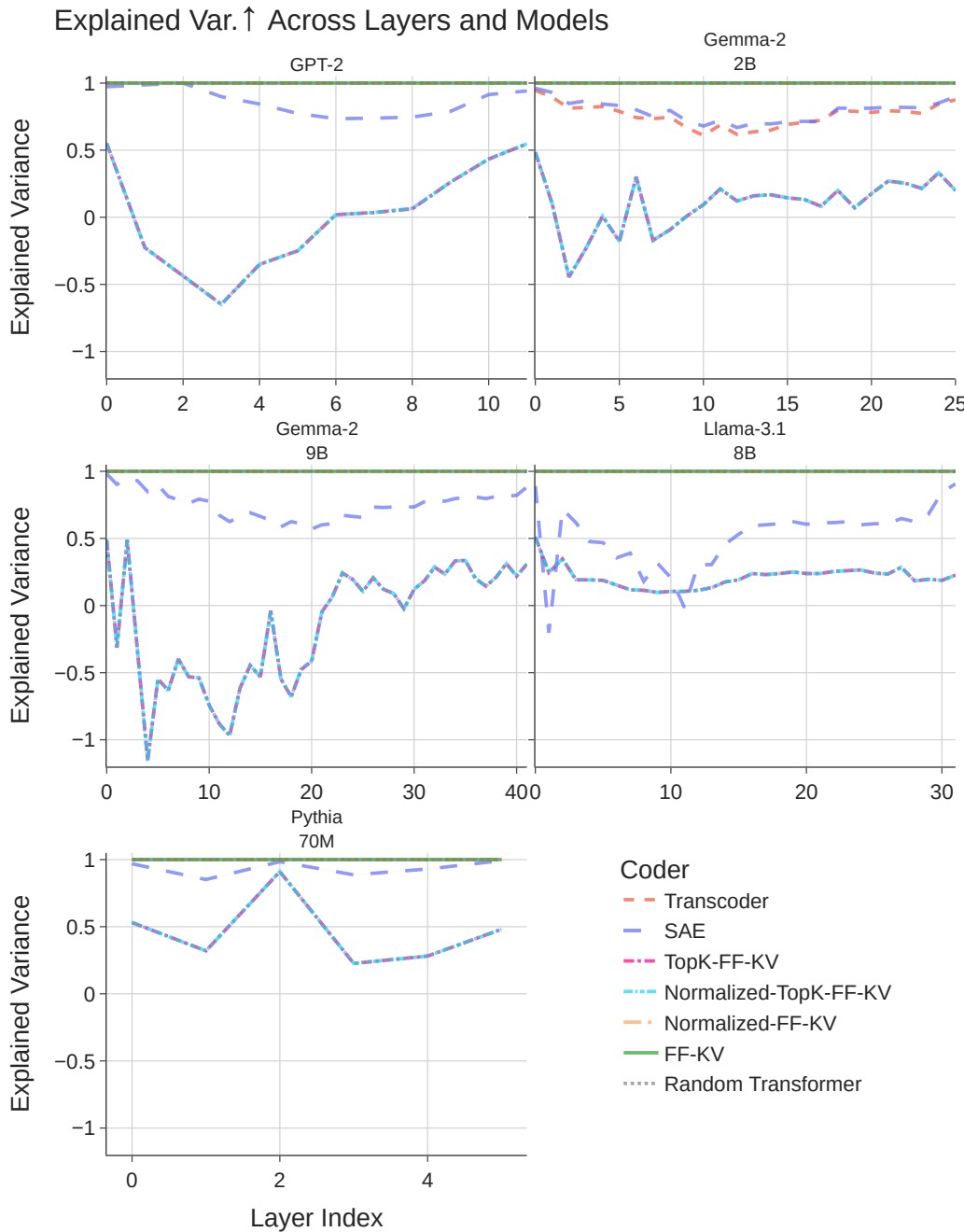

Figure 6: Detailed explained variance scores on all tested models, across all layers.

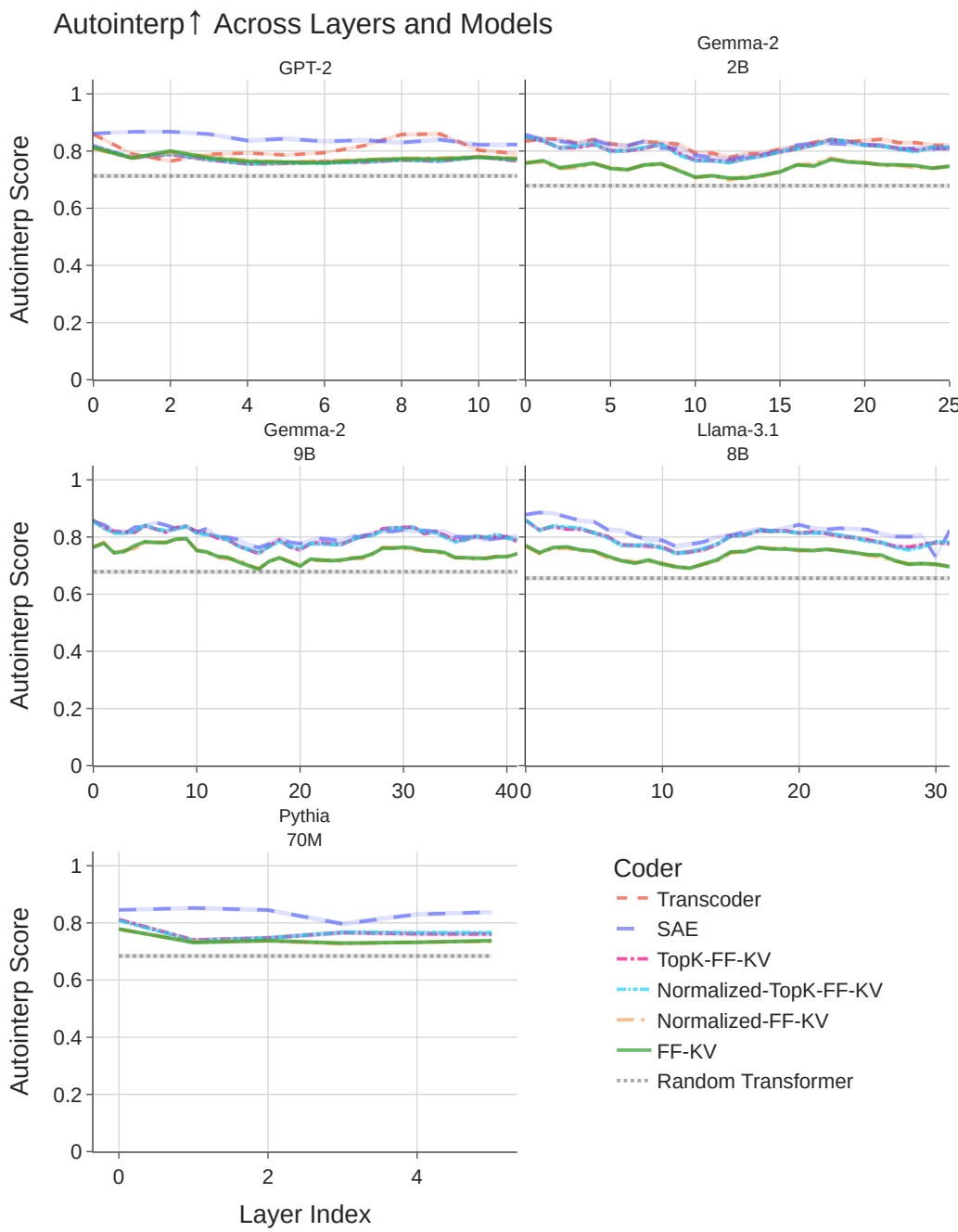

Figure 7: Detailed auto-interpretation scores on all tested models, across all layers.

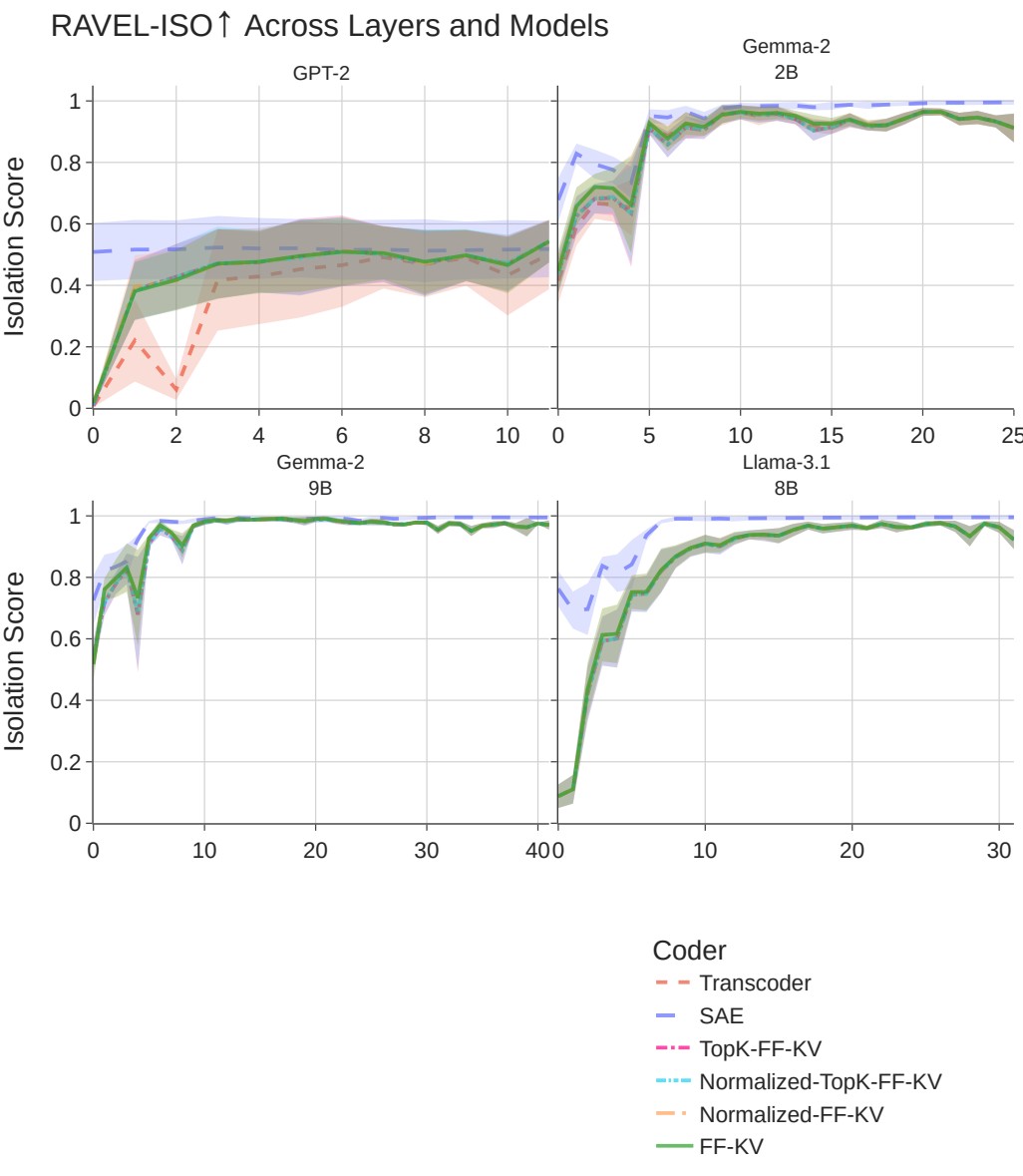

Figure 8: Detailed RAVEL scores on all tested models, across all layers.

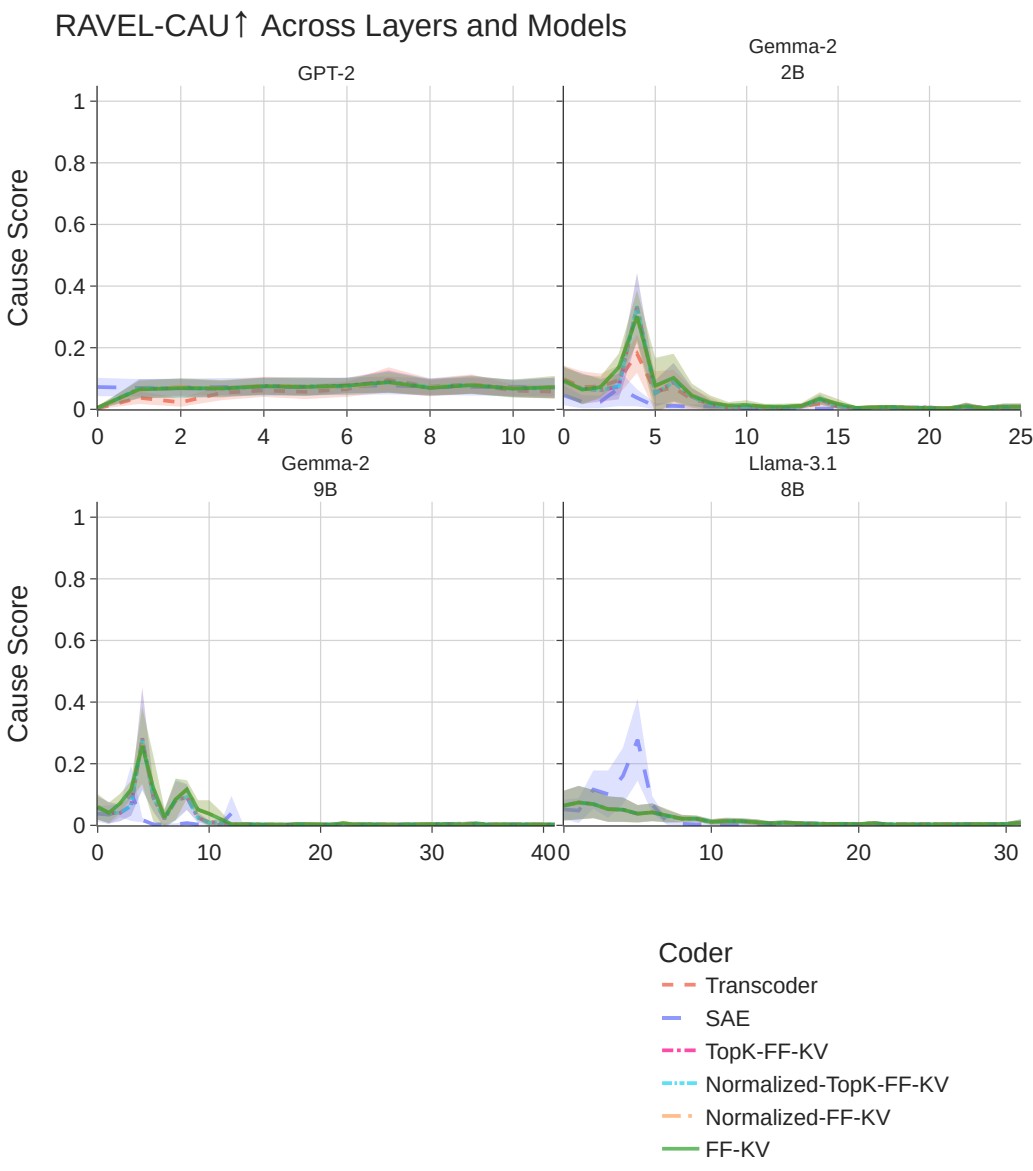

Figure 9: Detailed RAVEL scores on all tested models, across all layers.

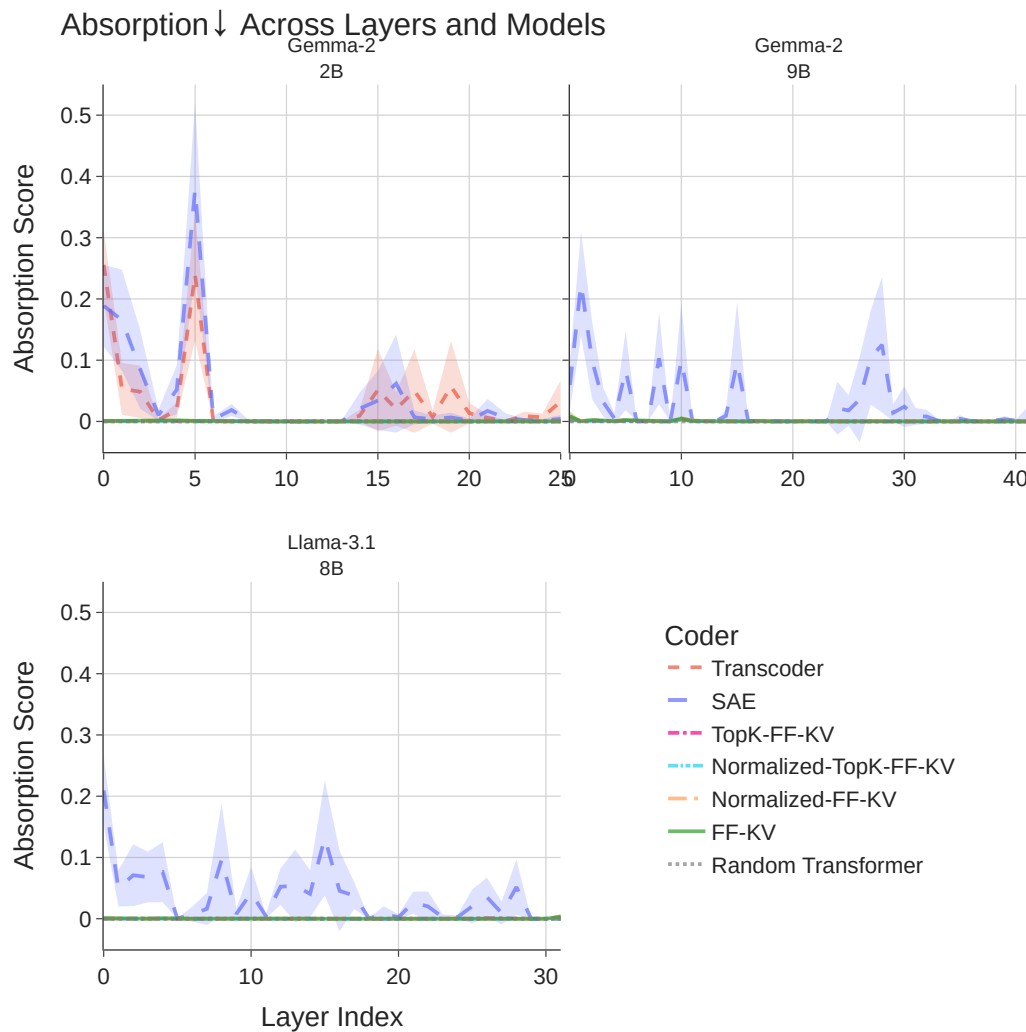

Figure 10: Detailed absorption scores on all tested models, across all layers.

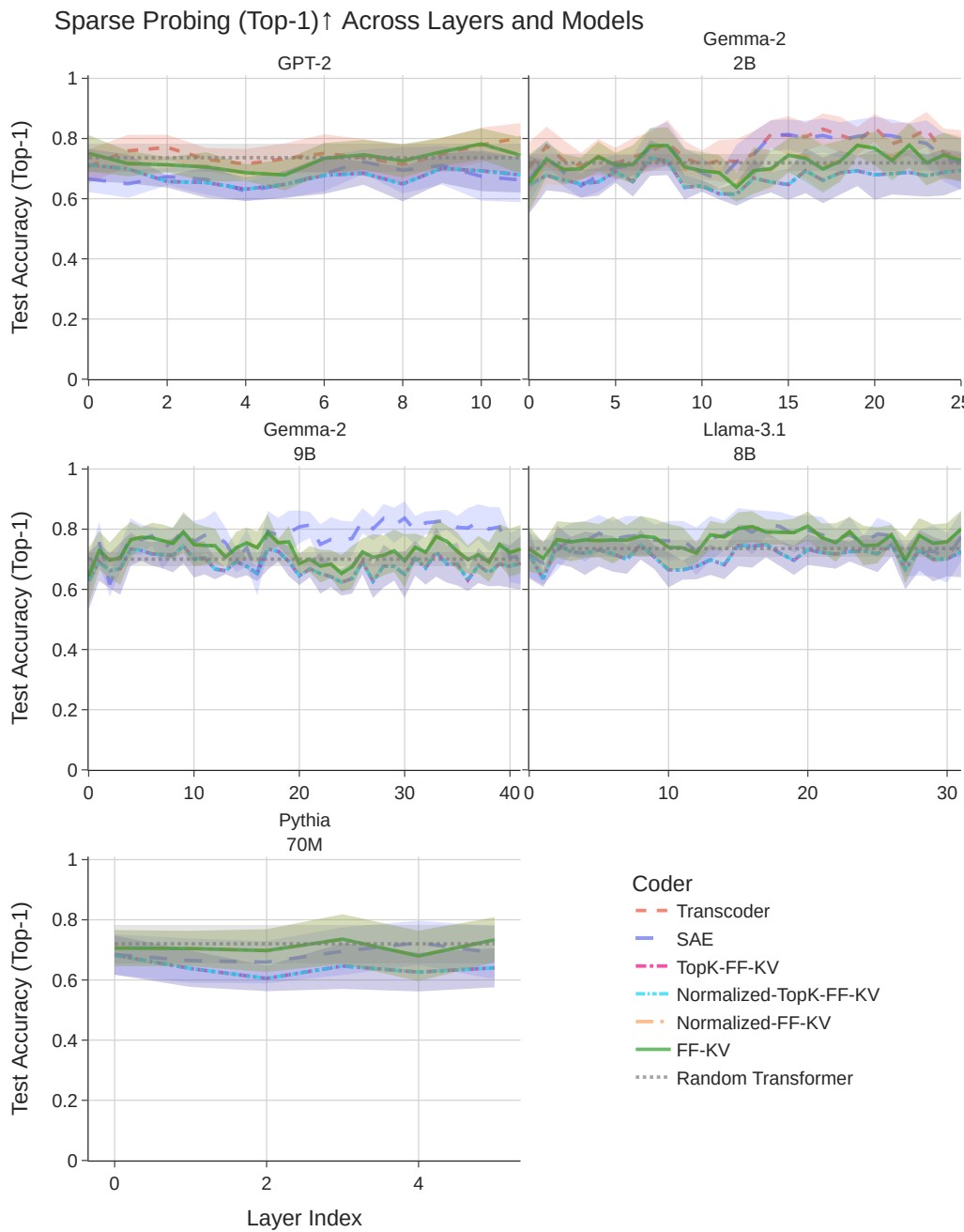

Figure 11: Detailed sparse probing scores on all tested models, across all layers, and various hyperparameter ($K$) choices.

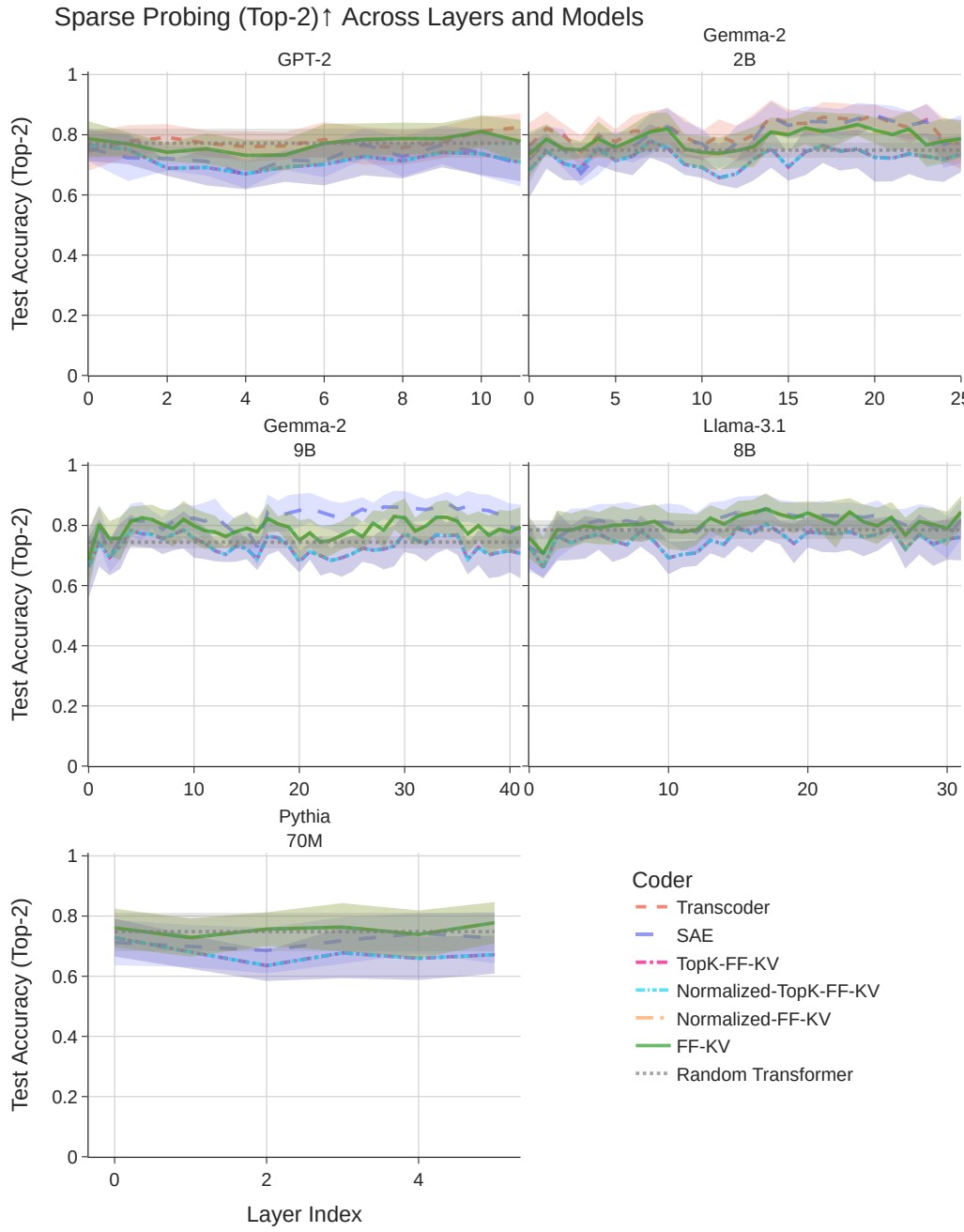

Figure 12: Detailed sparse probing scores on all tested models, across all layers, and various hyperparameter $(K)$ choices.

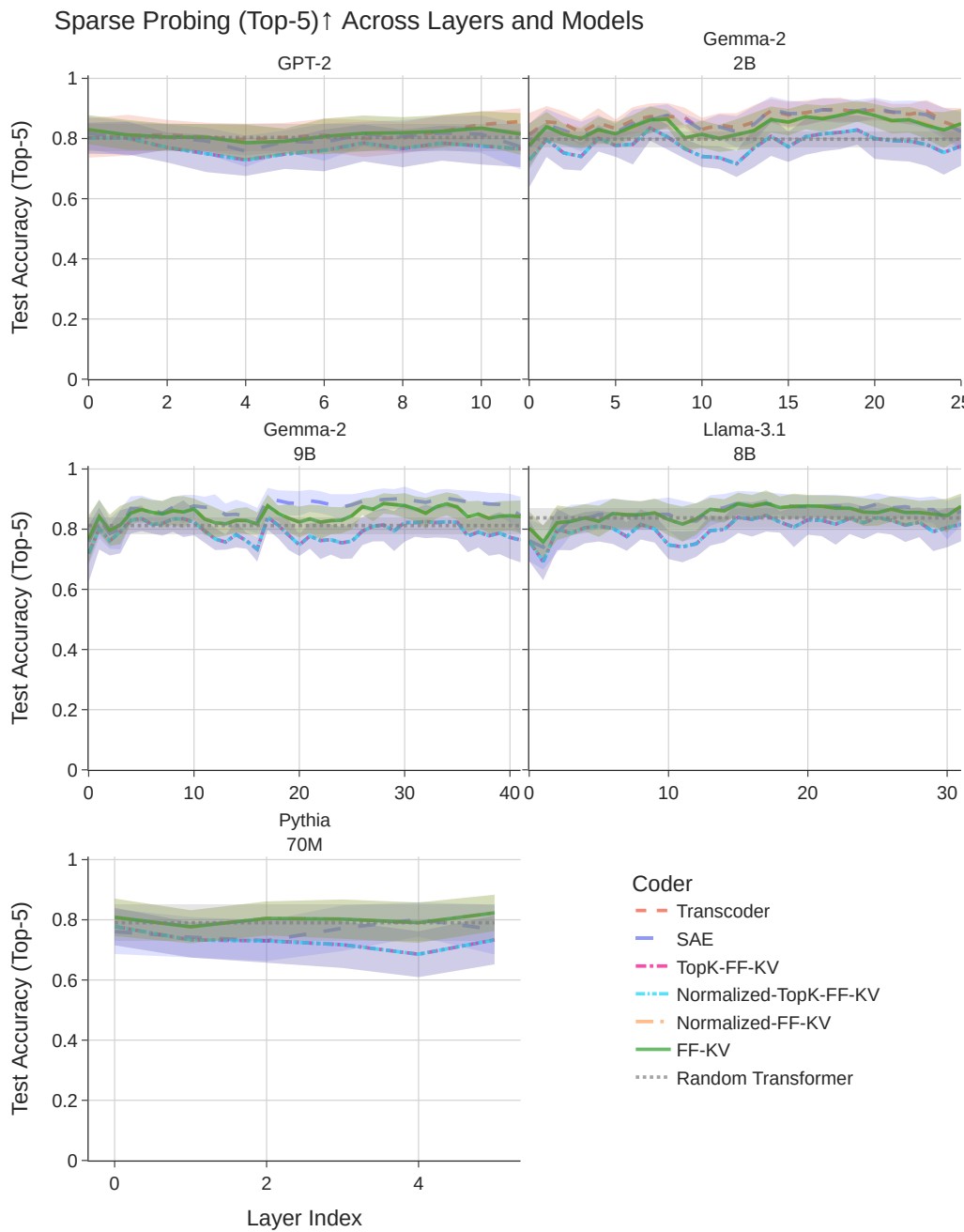

Figure 13: Detailed sparse probing scores on all tested models, across all layers, with **the same hyperparameter choice** as the main result in Table 1.

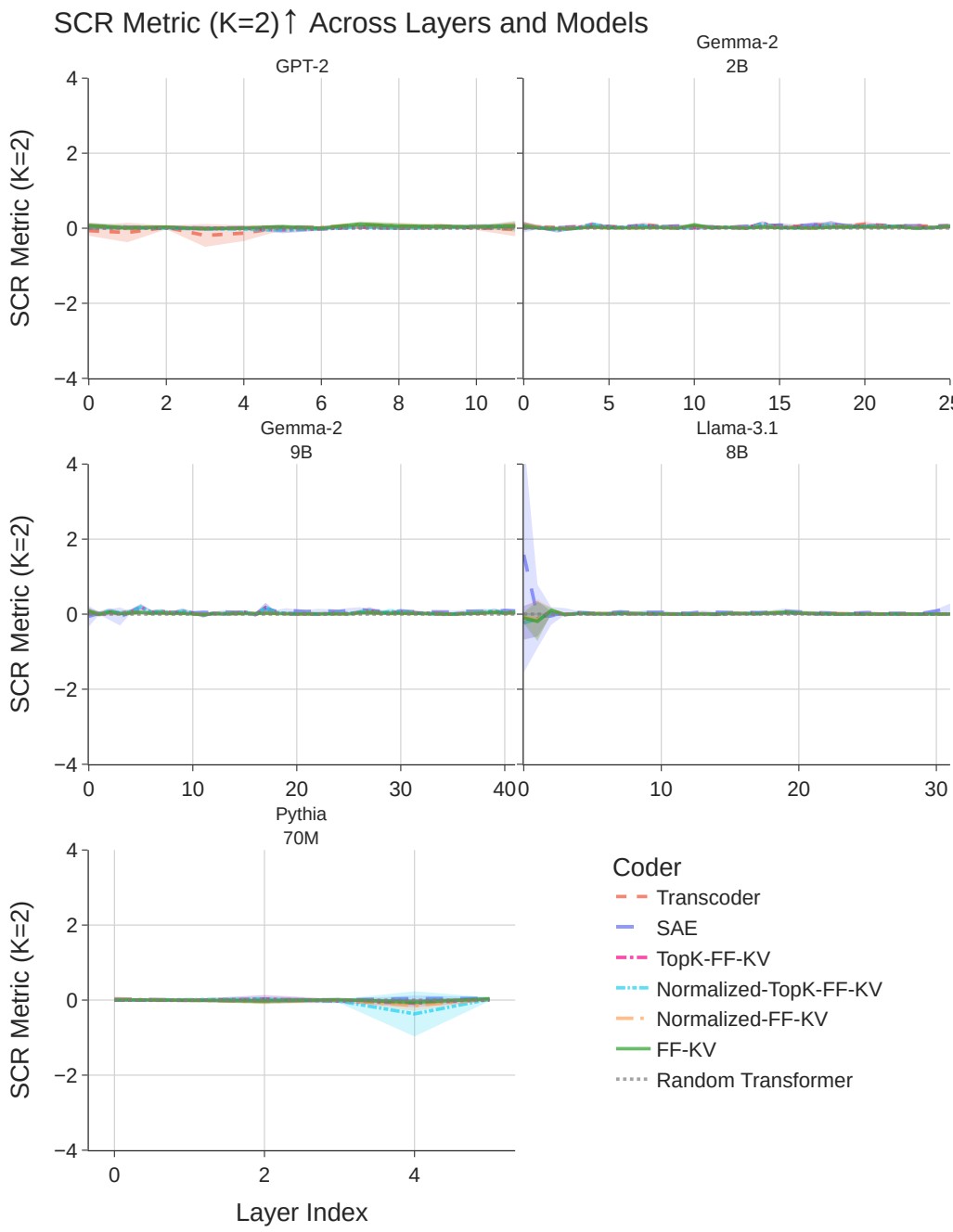

Figure 14: Detailed SCR scores on all tested models, across all layers, and various hyperparameter ($K$) choices.

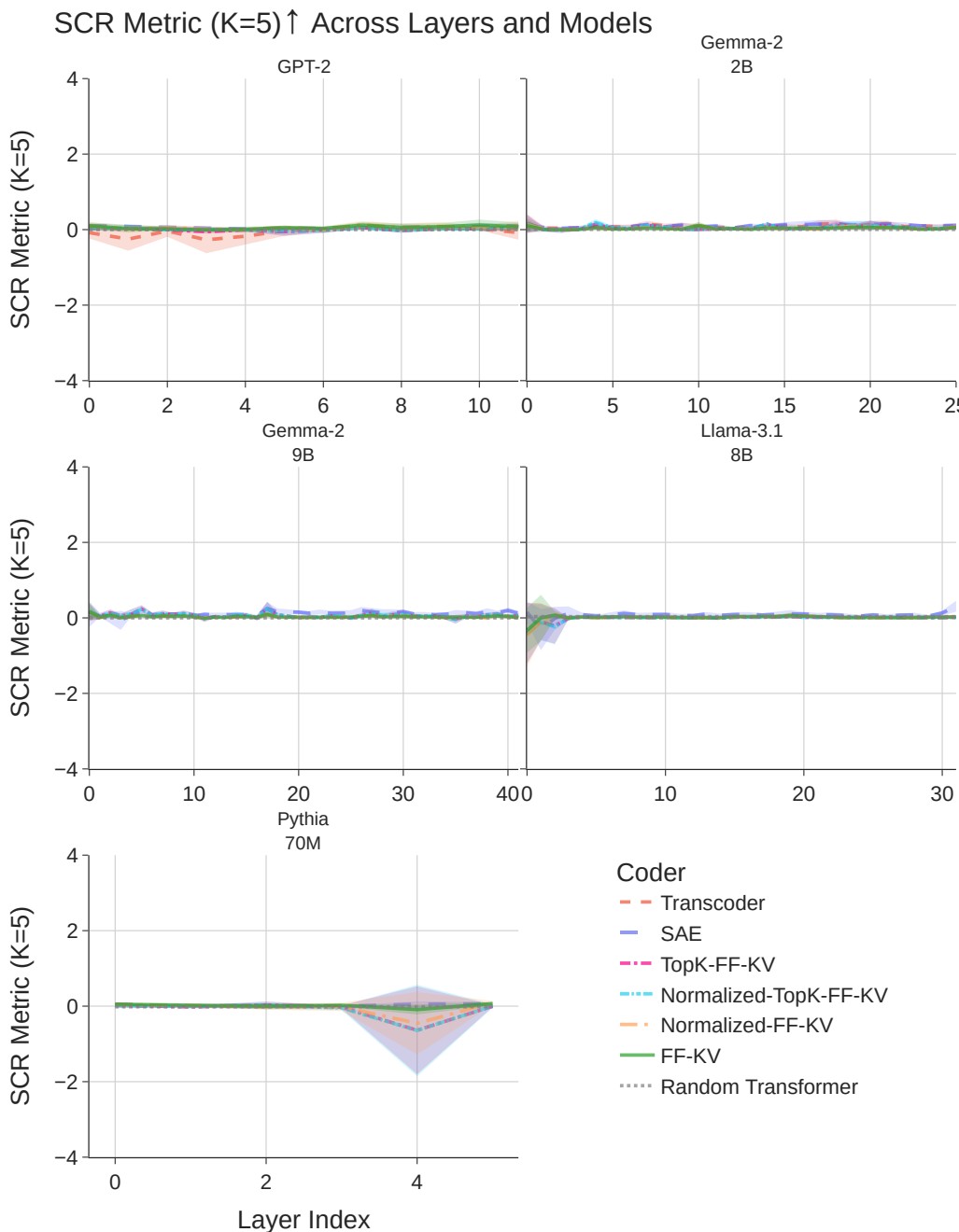

Figure 15: Detailed SCR scores on all tested models, across all layers, and various hyperparameter ($K$) choices.

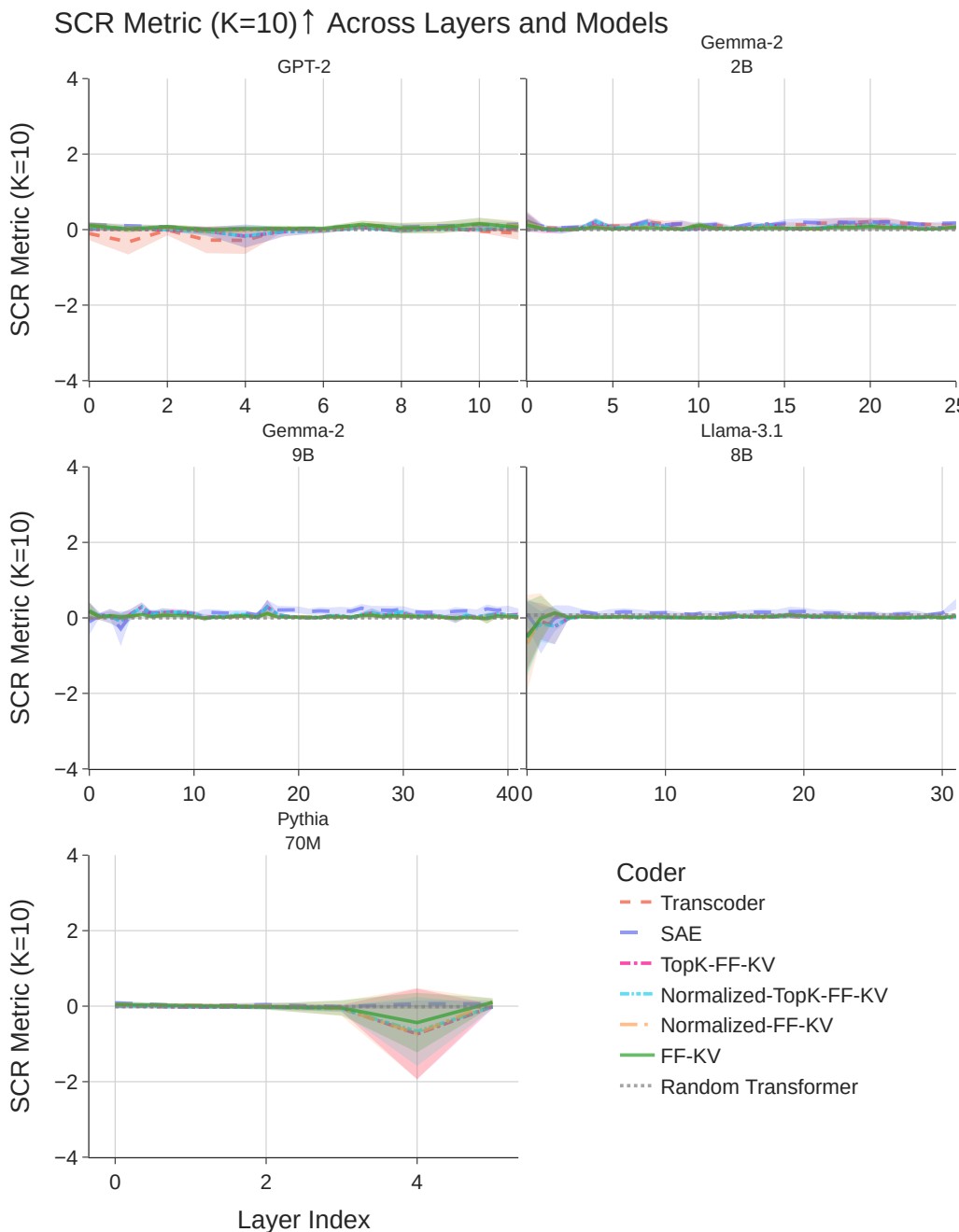

Figure 16: Detailed SCR scores on all tested models, across all layers, and various hyperparameter ($K$) choices.

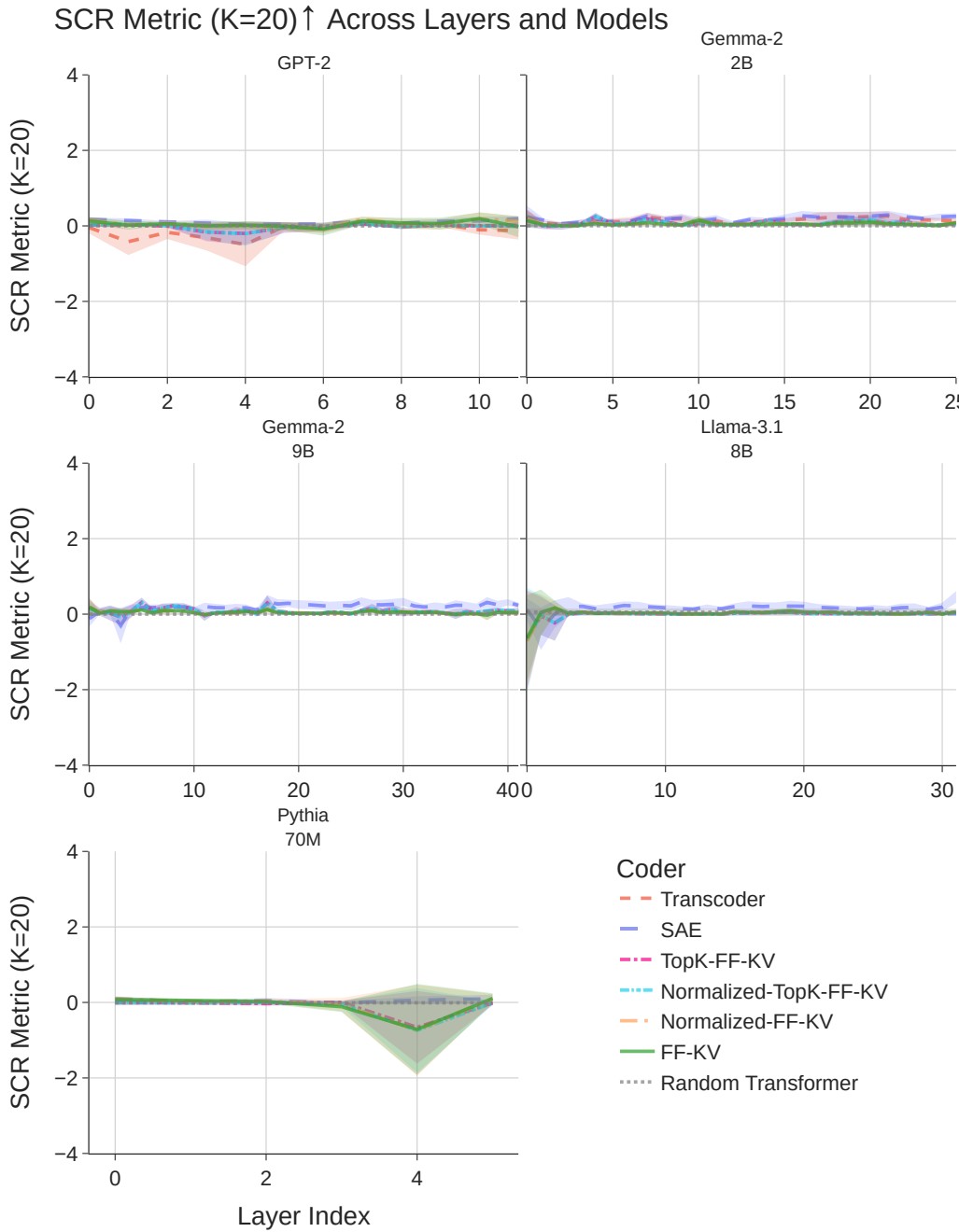

Figure 17: Detailed SCR scores on all tested models, across all layers, with **the same hyperparameter choice** as the main result in Table 1.

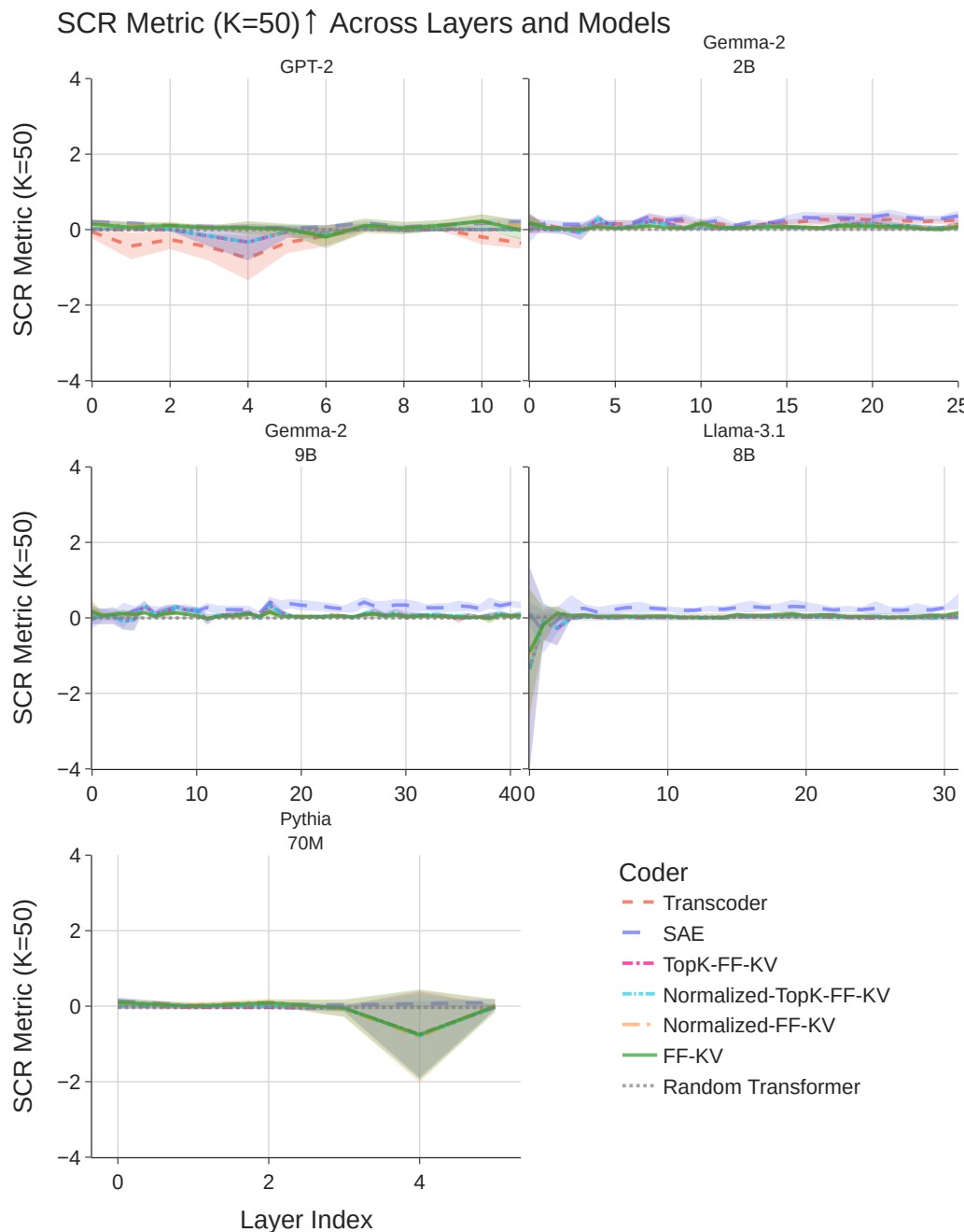

Figure 18: Detailed SCR scores on all tested models, across all layers, and various hyperparameter ($K$) choices.

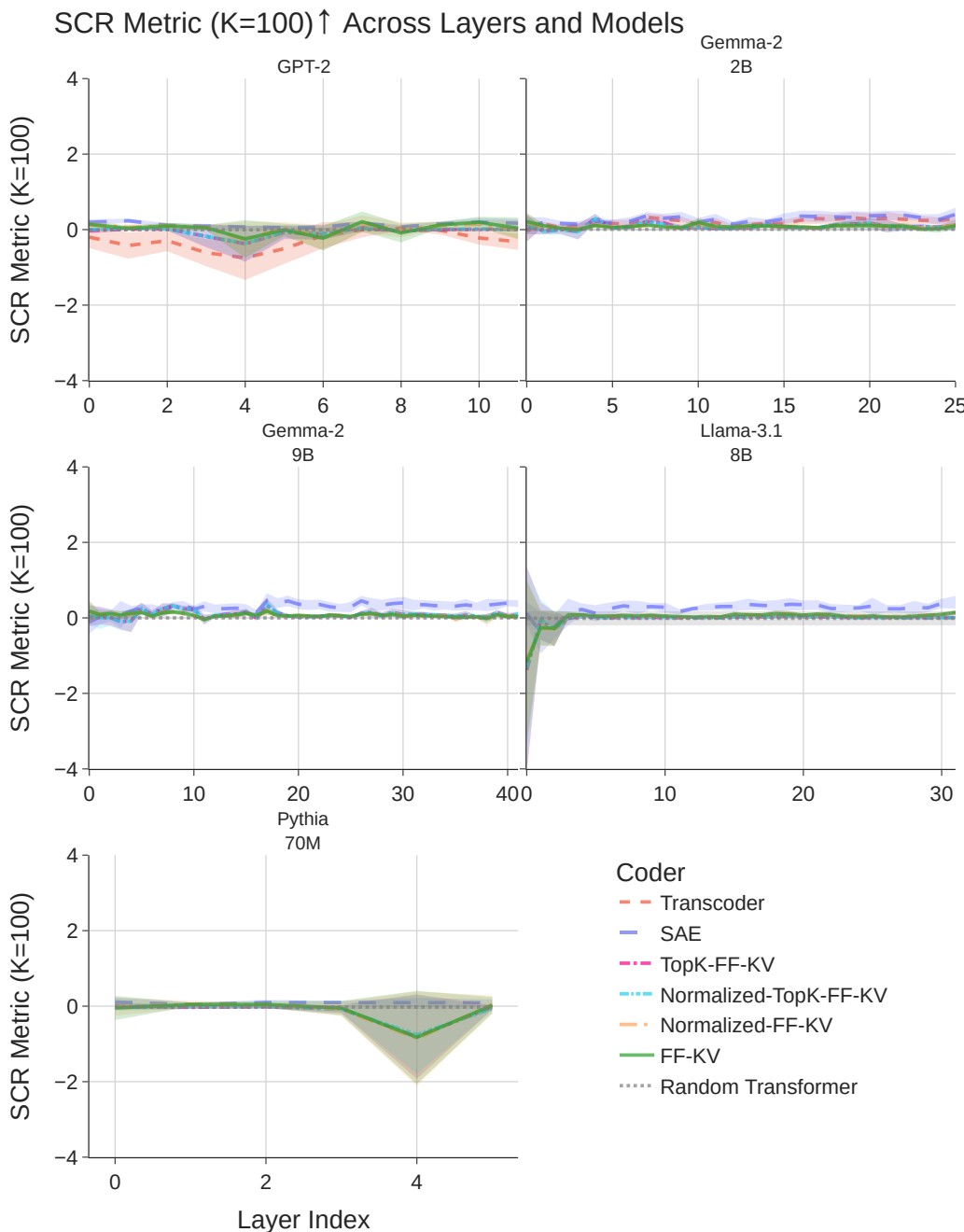

Figure 19: Detailed SCR scores on all tested models, across all layers, and various hyperparameter ($K$) choices.

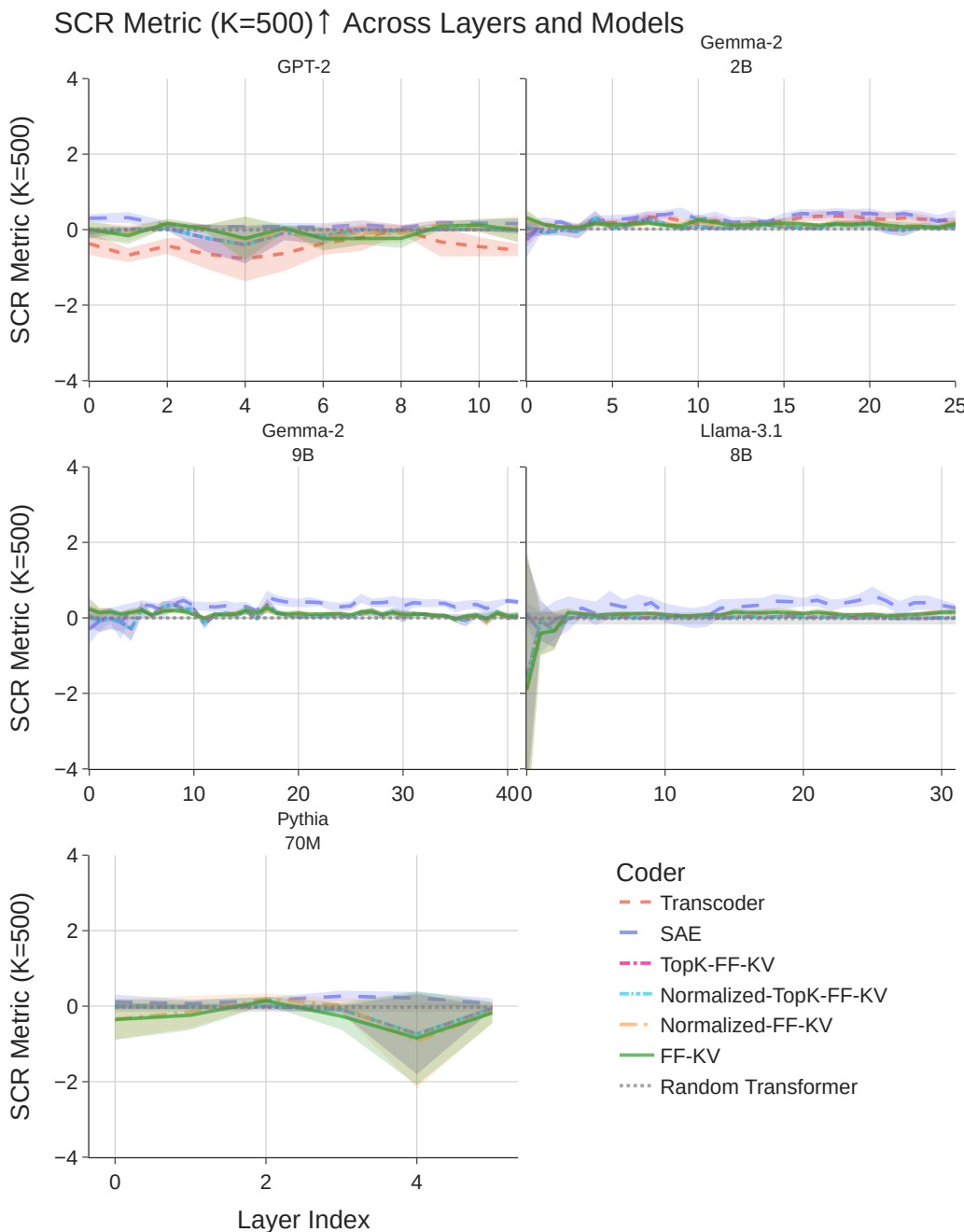

Figure 20: Detailed SCR scores on all tested models, across all layers, and various hyperparameter ($K$) choices.

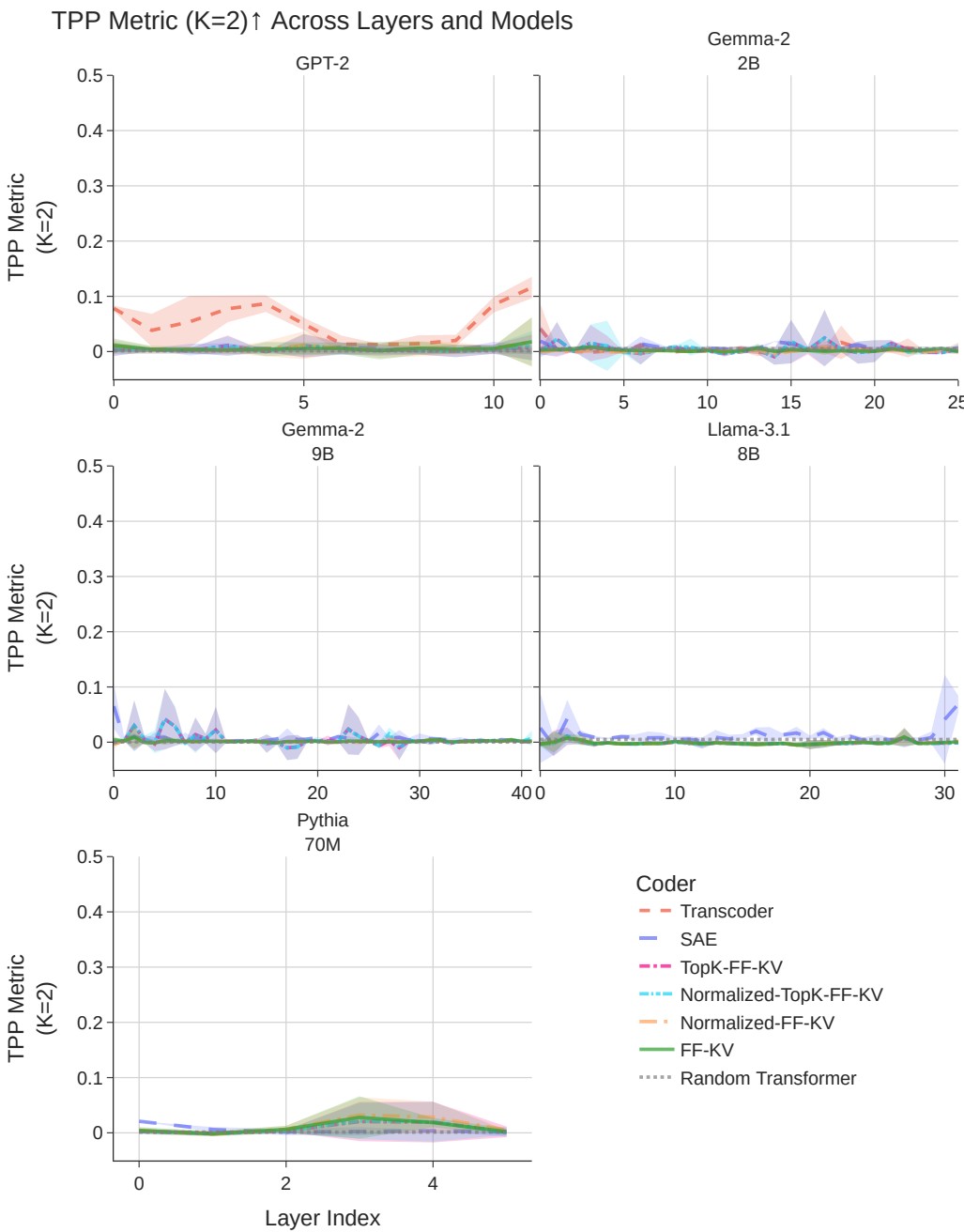

Figure 21: Detailed TPP scores on all tested models, across all layers, and various hyperparameter ($K$) choices.

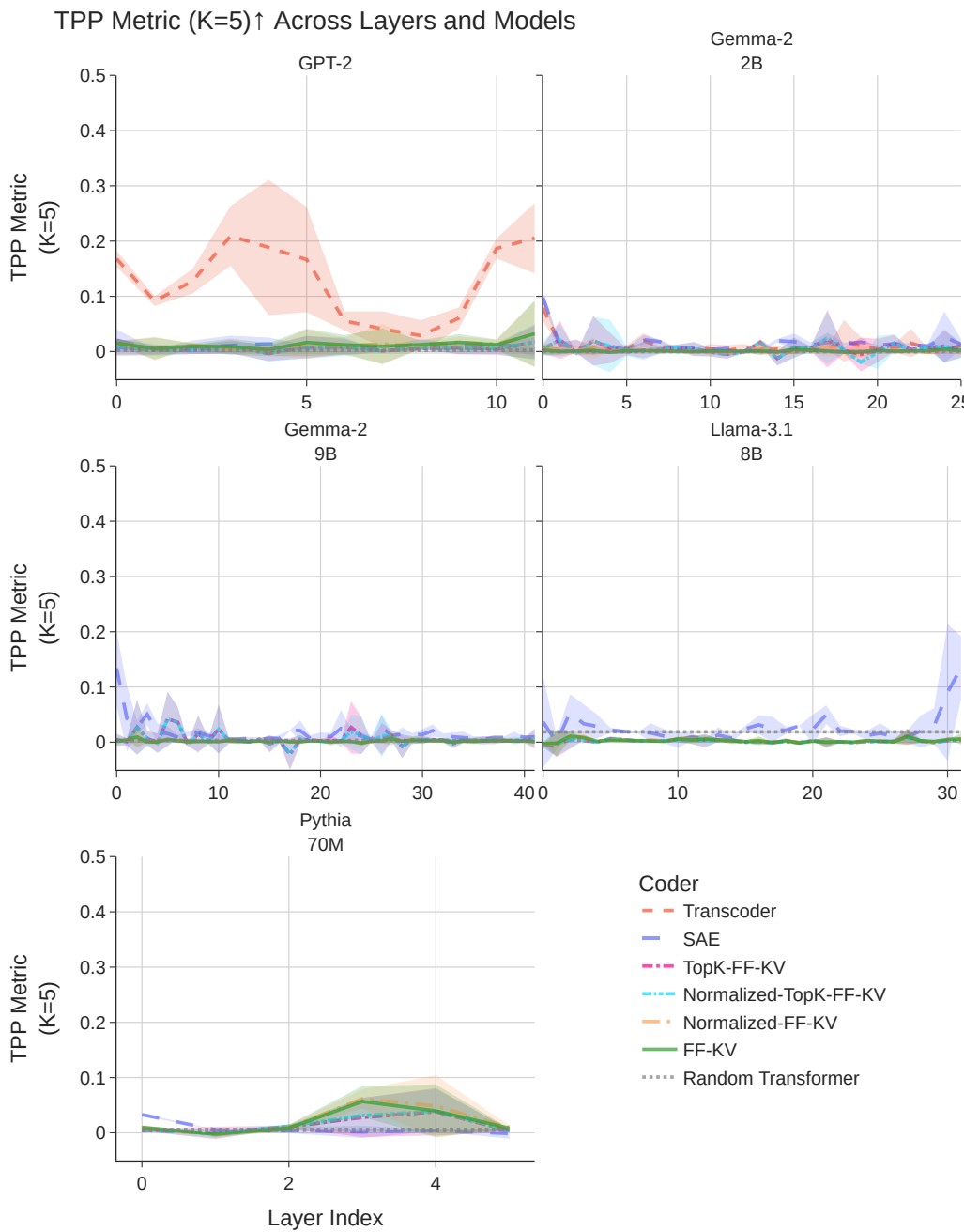

Figure 22: Detailed TPP scores on all tested models, across all layers, and various hyperparameter ($K$) choices.

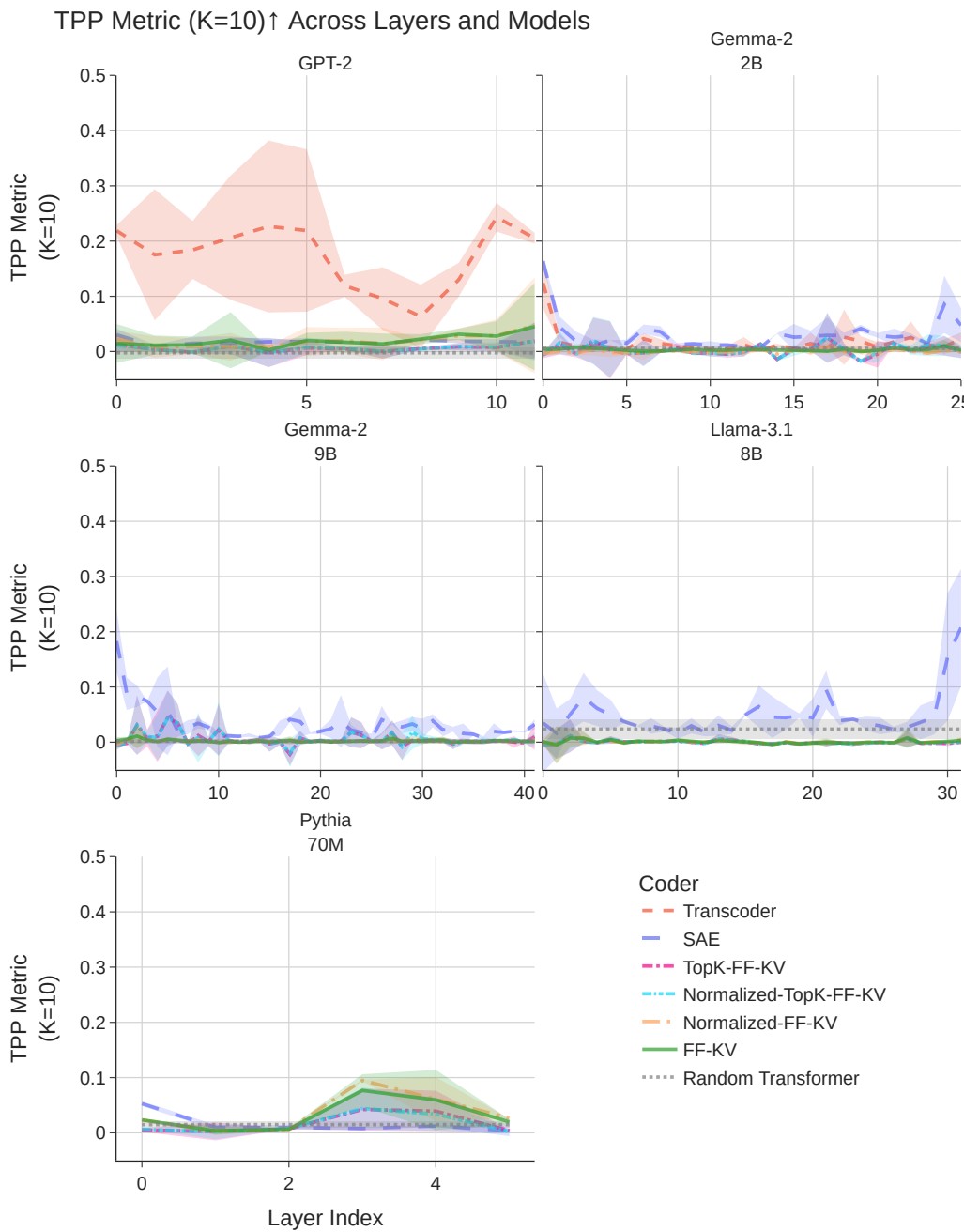

Figure 23: Detailed TPP scores on all tested models, across all layers, and various hyperparameter ($K$) choices.

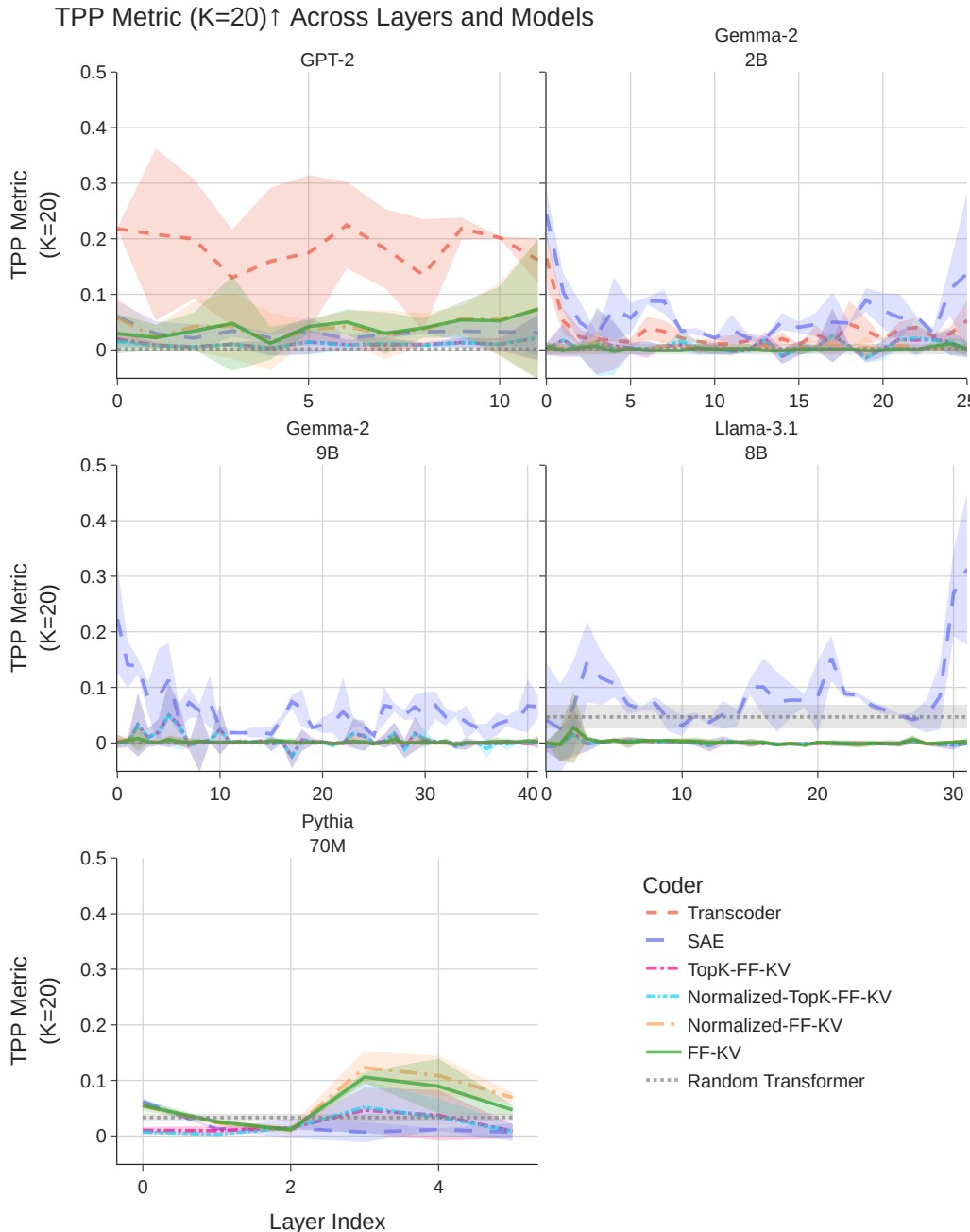

Figure 24: Detailed TPP scores on all tested models, across all layers, with **the same hyperparameter choice** as the main result in Table 1.

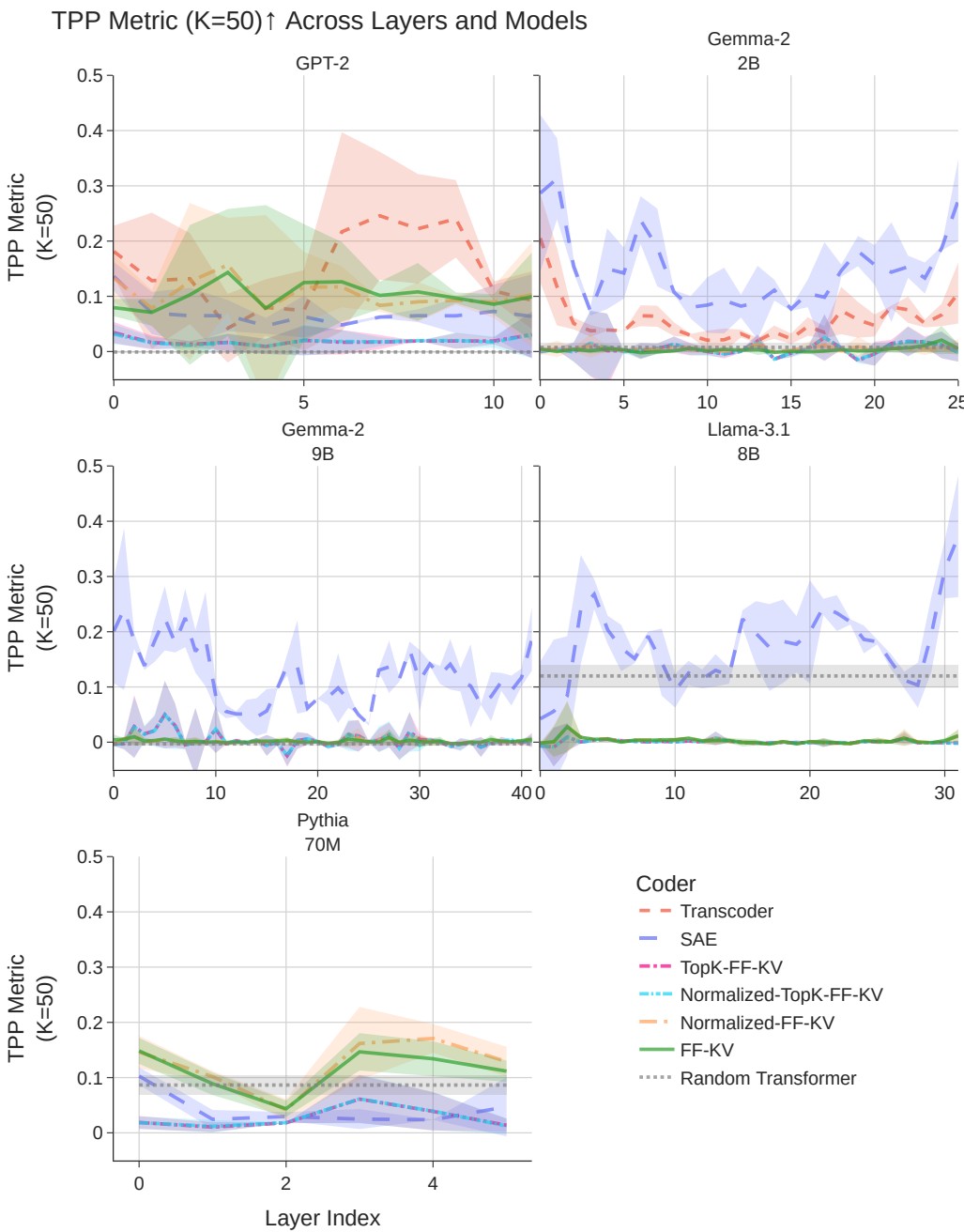

Figure 25: Detailed TPP scores on all tested models, across all layers, and various hyperparameter ($K$) choices.

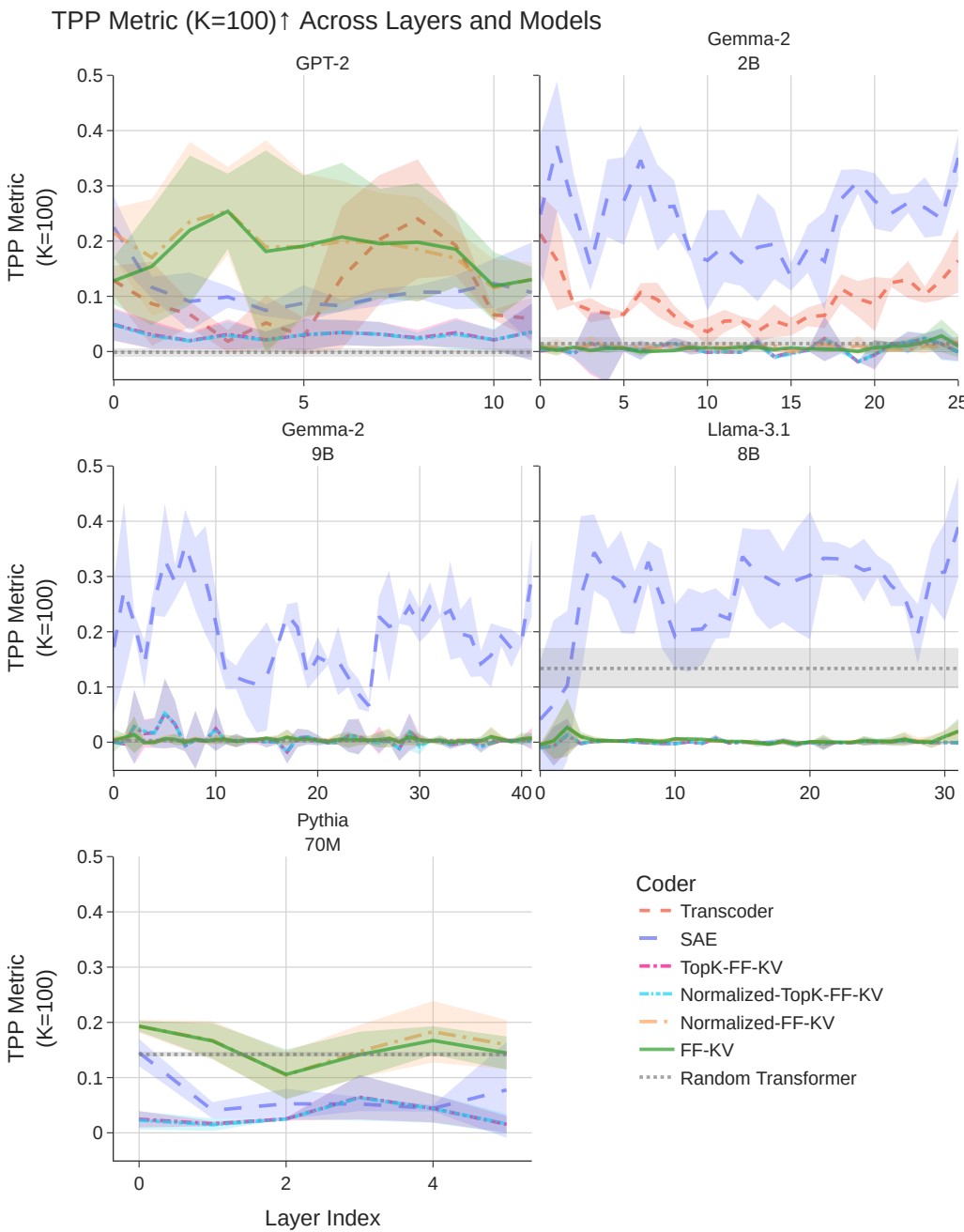

Figure 26: Detailed TPP scores on all tested models, across all layers, and various hyperparameter ($K$) choices.

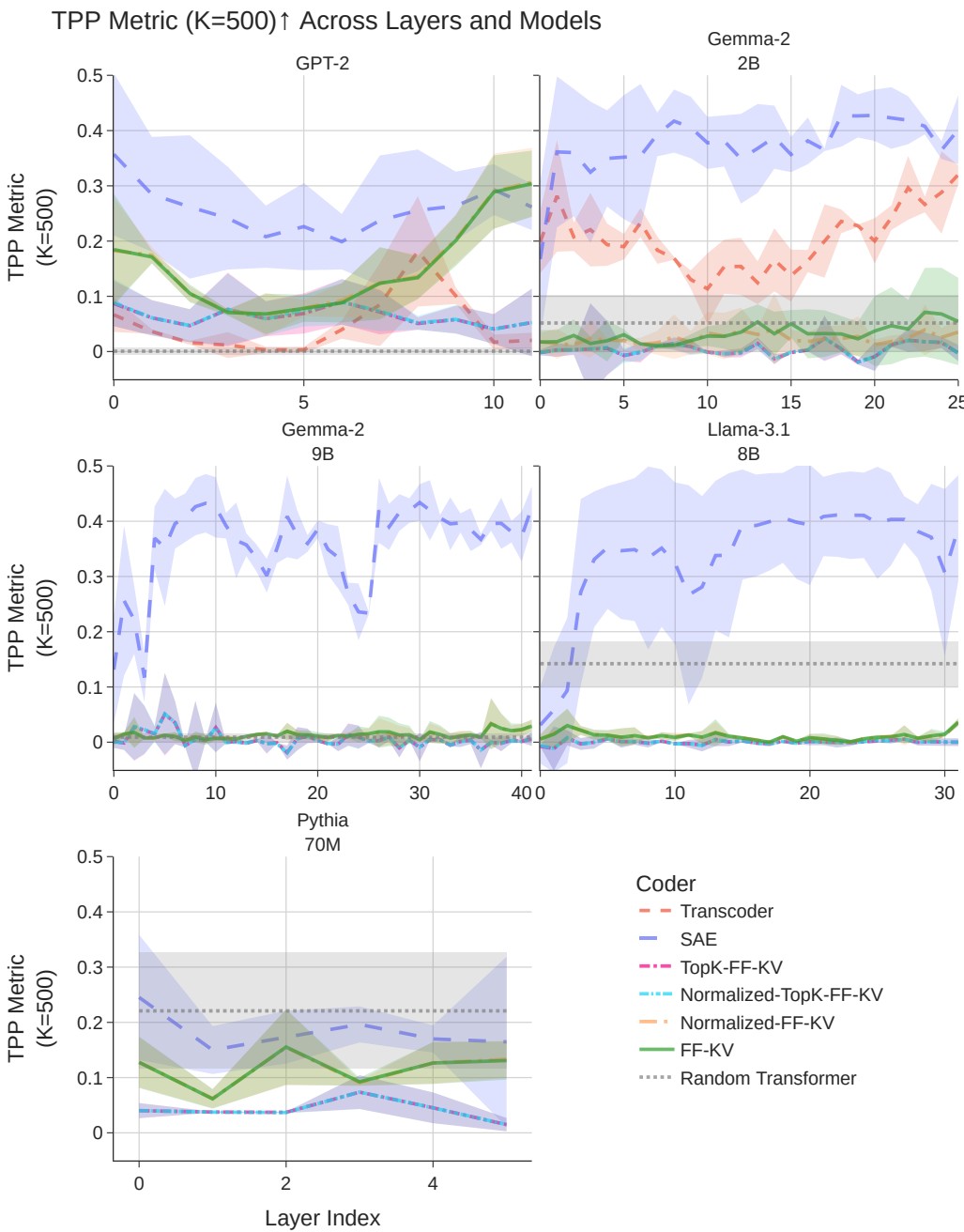

Figure 27: Detailed TPP scores on all tested models, across all layers, and various hyperparameter ($K$) choices.

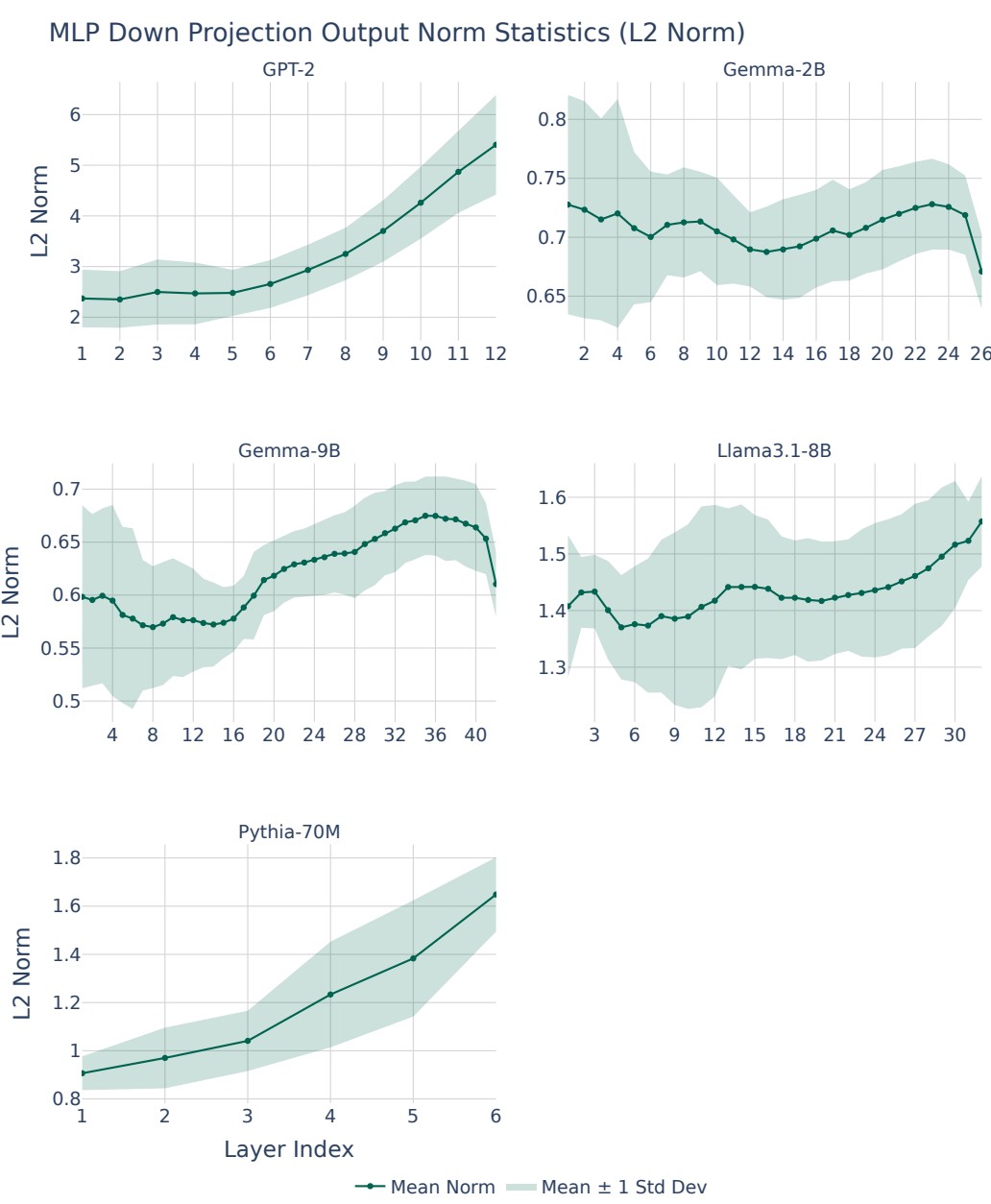

Figure 28: Distribution of the L2 norms of all tested models' FF-KV decoder weights (i.e., $W_2$ in its FF sublayer). Although the norms are not exactly one, they are concentrated in a narrow range.

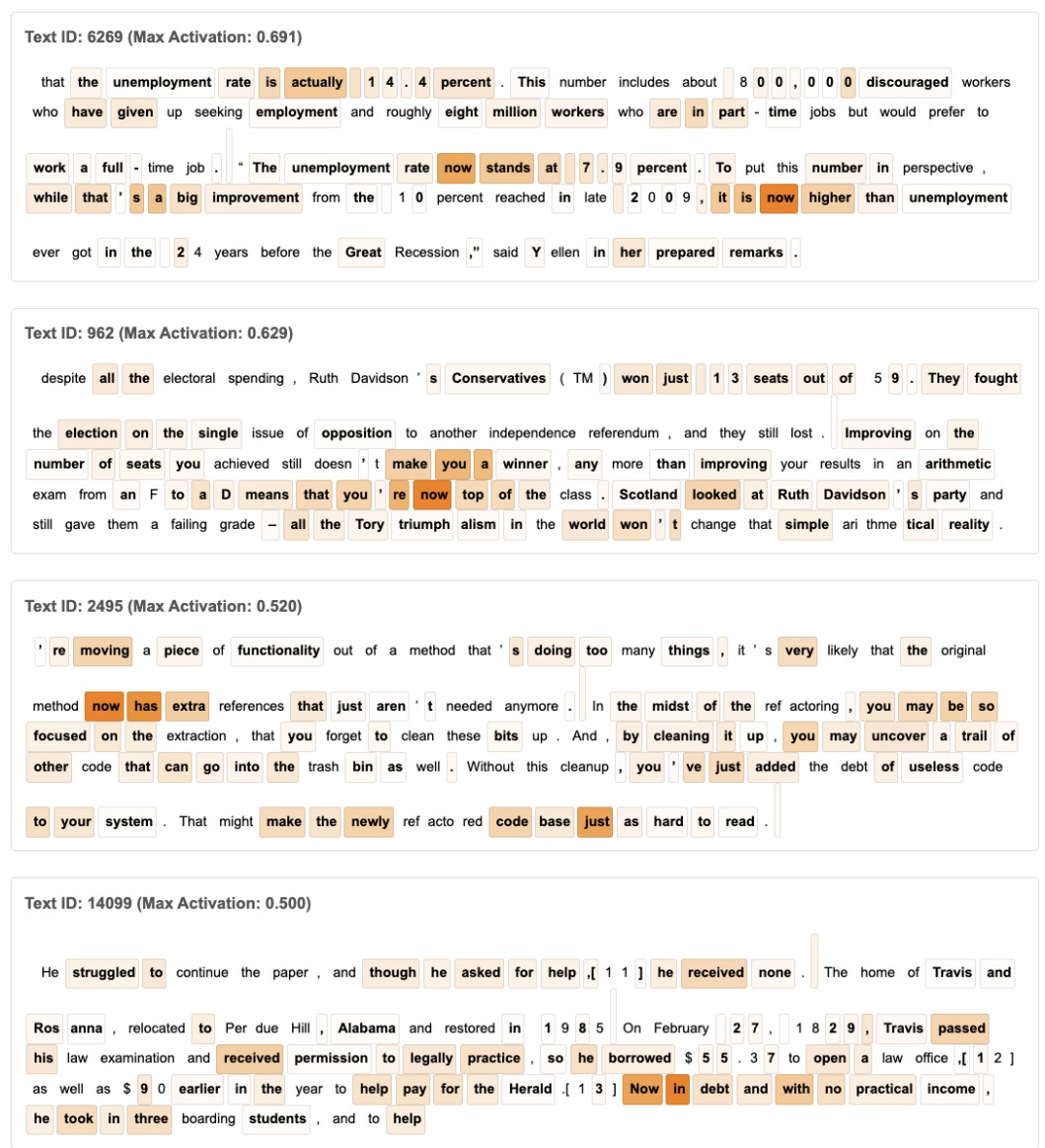

Figure 29: Top-4 activating examples for a particular feature in **FF-KV** annotated as **"superficial"**. This feature specifically activates most on the word "now", in various contexts.

**Text ID: 11936 (Max Activation: 2.609)**

Claudio Ran ieri as coach of the first team . " Following this move , **the** financial contract with Ran ieri , whose deal was coming to an end on 3 0 June 2 0 1 1 , has ended by mutual consent . Roma wishes to thank Claudio Ran ieri for the professionalism shown and the work done [ during his time at the club ]." Ran ieri enjoyed a successful first season with Roma after replacing Luciano Spal letti in September 2 0 0 9 . **The** club had endured an horrendous start to the campaign and Ran ieri , who had been fired by Juventus months earlier , rescued **the** team and nearly led them to the scu

**Text ID: 15318 (Max Activation: 2.484)**

£ 5 9 . 7 m Angel Di Maria -- Real Madrid to Man Utd , Aug . 2 0 1 4 6 ) £ 5 6 m Kaka -- AC Milan to Real Madrid , June 2 0 0 9 Sources told ESPN FC that Chelsea have been quoted a price of 4 0 million pounds by PSG for Cav ani , and that **the** player is eager to move to Stamford Bridge -- despite a public declaration that he would prefer to stay in France . There has already been preliminary contact between **the** London club and the player ' s camp , although Chelsea are still assessing **the** best course of action . With Mourinho having regularly complained

**Text ID: 7334 (Max Activation: 2.484)**

spokesman Adam Rosen said he is ' shocked ' that an agency of first responders would enforce such an order **the** week of Sept . 1 1 . " **The** four suspended firefighters said they were told that **the** order was issued because of racial discord [ in ] the department . **The** four , who include two white firefighters , a black firefighter , and a fourth firefighter who is a Cuban é mig ré , said no such problems exist ," wrote CBS , which also reported that **the** four firefighters trace the issue " to a decision by several firefighters to replace a tattered American flag last month in one of May wood ' s fire houses . **The** new flag mysteriously

**Text ID: 12010 (Max Activation: 2.453)**

Trading standards are investigating after a couple who stayed at a hotel claimed to have been " f ined " £ 1 0 0 by a hotel which they described as a " rotten st inking ho vel " on TripAdvisor . Tony and Jan Jen kinson , from White haven in Cumbria , posted a review on the website after staying at the Broadway Hotel in Blackpool . However , **the** couple later found that £ 1 0 0 charged to their credit card , which **the** BBC reported was the result of a hotel policy in the case of " bad " reviews . **The** manager of **the** hotel was not available for comment last night . **The** Jenkins ons , who

Figure 30: Top-activating examples for a feature in $k$-**Sparse FF-KV** annotated as **"superficial"**. This feature specifically activates most on the word "the", in various contexts.

**Text ID: 11353 (Max Activation: 18.412)**

k in xrange ( 0 , high + 1 ): p = Eval Poisson P mf ( k , lam ) pm f . Set ( k , p ) pm f . Normalize () `return` pm f The range of values in the computed P mf is from 0 to high . So if the value of lam were exactly 3 . 4 , we would compute _ lam = 3 . 4 goal _ dist = think bayes . Make Poisson P mf ( lam , 1 0 ) I chose the upper bound , 1 0 , because the probability of scoring more than 1 0 goals in a game is quite low . That ' s simple

**Text ID: 8952 (Max Activation: 17.365)**

q - k + 1 )/ k if k <= p : c list . `append` `(` c ) d `*=` 1 . 0 *( q - k + 1 )/( p + q - k + 1 )/ k if k <= q : d list . append ( d ) `return` np `.` `array` `(` clist [::- 1 `]),` np . array ( d list [::- 1 ]) def arg box ( y , ymin , ymax , imin , imax ): '" find limits ( we hope ) where y [ i ] is between ymin and ymax '" ii = np . arg where ( np . logical _ and ( y > ymin , y < ymax )) ii = ii

**Text ID: 4503 (Max Activation: 17.178)**

asm __ (" f sel % 0 , % 1 , % 2 , % 3 " : "= f " ( result ) : " f " ( test ), " f " ( b ), " f " ( a )); `return` `result` `;` } Such optimizations are implementation details , but are described here because they provide a practical performance benefit to the performance - conscious user . It would be nice if std STL implementations provided such things , though Met rower ks has been known to do so in some cases . Instead of just < algorithm >, EAST L has < algorithm . h >, < sort . h >, < algo set . h >,

**Text ID: 10231 (Max Activation: 16.843)**

1 0 0 0 0 0 0 0 % 1 0 + i 1 / 1 0 0 0 % 1 0 + i 1 / 1 0 % 1 0 )) % 1 0 ) % 1 0 + 1 0 * i 0 ; printf ( "% d " , i 2 ); `return` `(` `0` ); } user @ kali : ~$ gcc - o d link - wps - gen quanta - wps - gen . c user @ kali : ~$ ./ d link - wps - gen 9 7 3 2 9 3 2 9 user @ kali : ~$ You can fetch this program at https ://

**Text ID: 9000 (Max Activation: 14.418)**

Figure 31: Top-activating examples for a feature in **SAE** annotated as **"superficial"**. This feature activates on the word "return", especially in programming-related contexts.

**Text ID: 4591 (Max Activation: 34.000)**

might be worth using as a reference . A full discussion of compiler in lining characteristics is outside the scope of

this document , but some Internet discussions regarding GCC in lining problems can be found at : http :// groups .

google . com / group / comp . lang . c ++ / browse _ frm / thread / b 7 4 **eed** 1 **6 bd 4 8 d 4** 2 e http :// groups .

google . com / group / fa . linux . kernel / browse _ frm / thread / 1 8 6 1 b 2 6 **3 4 cd** fa **6** 8 a / http :// www . pixel
glow . com / lists / archive

---

**Text ID: 3930 (Max Activation: 33.750)**

buy all of the parts needed , including the plastic case , knob , and AC adapter . You can edit your cart after
loading the project if you want to change anything . To access the shared project , go to http :// www . m ouser .
com / Project Manager / Project Detail . aspx ? Access ID = b 6 8 a 3 0 2 **3 1 c** or http :// www . m ouser . com / Tools

/ Tools . aspx and enter this access code : b 6 8 a 3 0 2 **3 1** c Upgrades I am often asked what can be done to
upgrade the designs that I publish . In this case , there

---

**Text ID: 6968 (Max Activation: 32.250)**

an environment variable JE BIO _ API KEY , or pass it as a parameter if you are importing the script as a library

). Queries return JSON output , except for download requests , that return binary attachments . The return " code "

variable is set to 0 on success , != 0 on error . Here are a few examples : Query a file hash : $ jeb io . py
check 4 2 aaa 9 3 **a 8 9 4 a 6 9** bf **cbc 2** 1 **8 2 3 b** 0 9 e 4 ea 9 **f 7 2** 3 **c 4 2 8** 4 2 aaa 9 3 **a 8 9 4** a
6

---

**Text ID: 6971 (Max Activation: 31.500)**

asta . apk " } } Note : the user details section is present only if you up lola ded the file yourself . Upload a file :

$ jeb io . py upload 1 . apk 1 . apk : { " code ": 0 , " uplo ade ven tid ": 1 5 5 } Download a file : (

subject to permission ) $ jeb io . py download a 2 ba 1 b **acc 9 9** 6 b **9** 0 b **3 7 a** 2 c 9 3 **0 8 9** 6 9 2 bf 5 **f 3**
**0 f 1 d 6** 8 a 2 ba 1 b **acc 9 9** 6 b

Figure 32: Top-activating examples for a feature in **Transcoder** annotated as **"superficial"**. This feature activates on the combination of digits and alphabet, in various contexts.

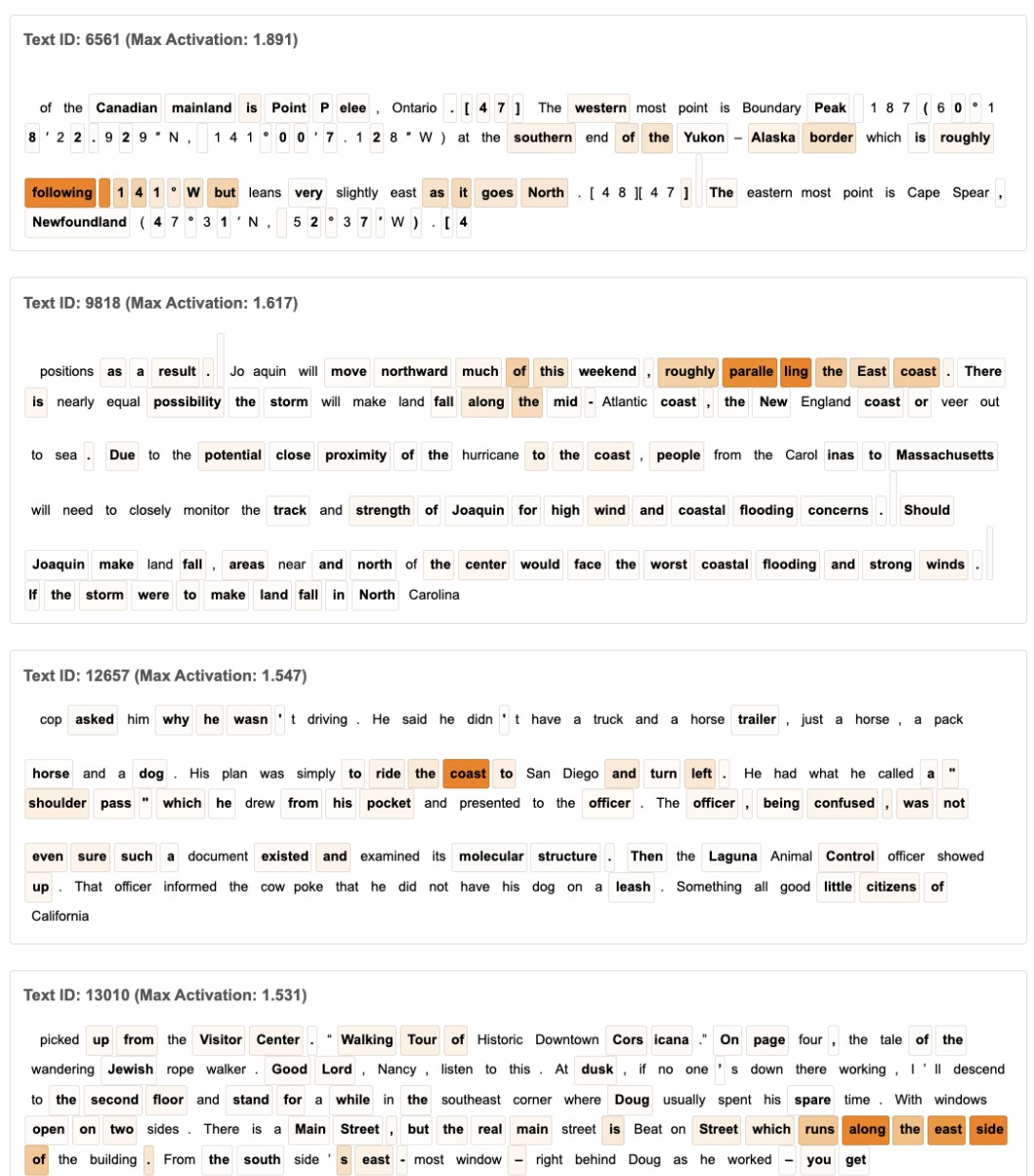

Figure 33: Top-activating examples for a feature in **FF-KV** annotated as **"conceptual"**. The specific annotation was "concept related to the coast" especially in various contexts.

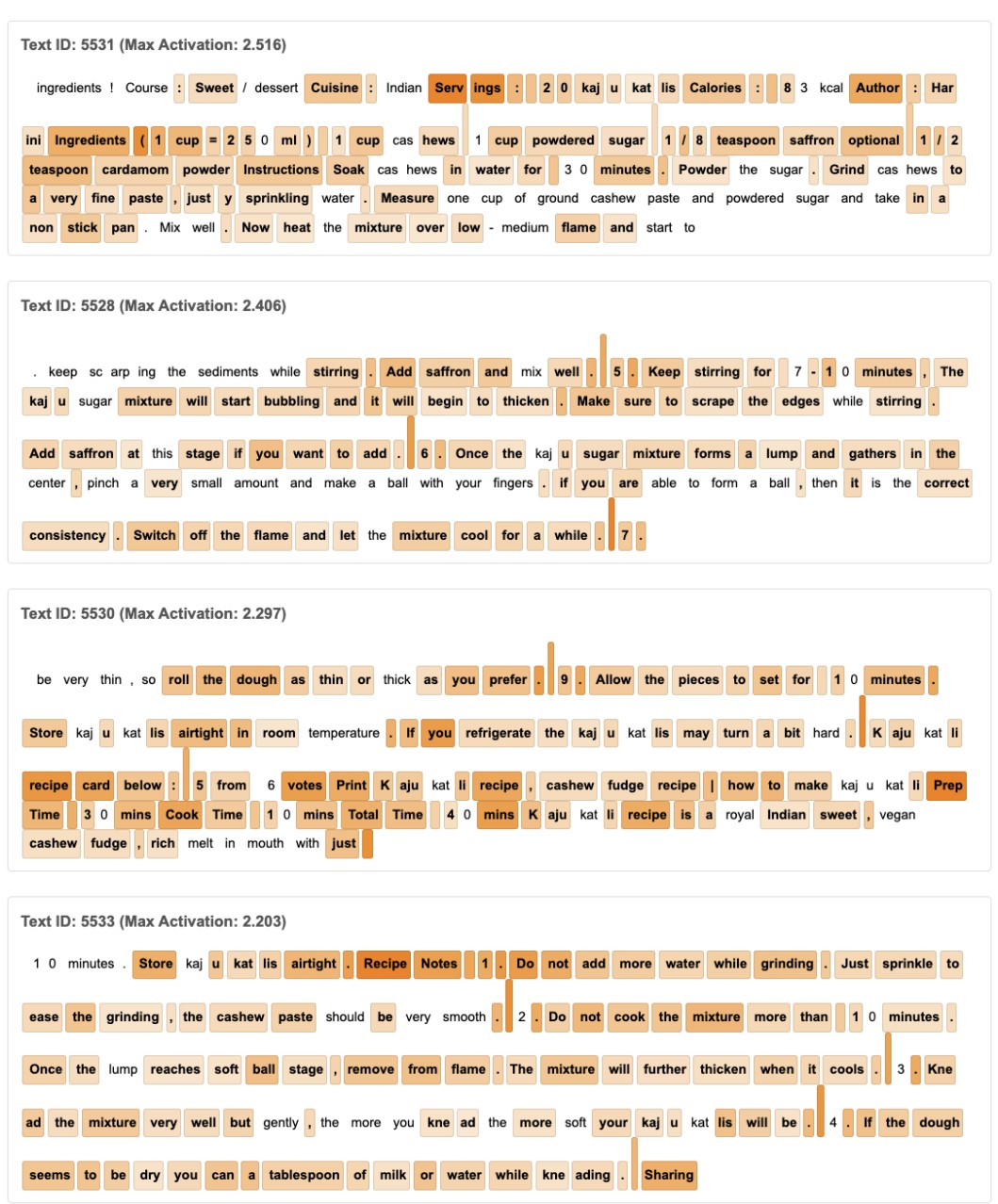

Figure 34: Top-activating examples for a feature in *k*-**Sparse FF-KV** annotated as **"conceptual"**. The specific annotation was "concept related to recipes" especially in contexts related to deserts.

**Text ID: 14508 (Max Activation: 14.855)**

I did with the Codex . WI RED : How do you deal with technology ? Sera fini : I remember my first encounter with a tablet . I was working on the opening titles of two Italian television broadcasts , O nda Verde and Enzo Bi agi ' s La L unga Marcia about his journey through China . It was a new tool , wired to a giant - size computer – quite fascinating at the time . I used it recently to illustrate Nature Stories by Jules Ren ard , but I realized my hand is much quicker . WI RED : Another encyclopedia of nature . Are you somehow obsessed with that kind of book ? Sera fini :

**Text ID: 6117 (Max Activation: 13.967)**

right , are John Horne Too ke another radical MP and Catharine Macaulay . She , like the other women , wears French tri colours . The people in this print are all linked by their support for the Revolution . The women were distinguished for ref uting Burke in print , or so it seemed . Williams who was noted for her sympathetic , eyewitness Letters Written in France had just published a poem in praise of the storming of the Bastille . Catharine Macaulay ' s forthcoming attack on Burke ' s Reflections had been announced and Barba uld , who had first opposed Burke in March 1 7 9 0 , was assumed to be writing another ref utation of his Reflections . While

**Text ID: 13267 (Max Activation: 13.466)**

and political risks which UK businesses may face when operating abroad , including in Israel and the OPT s . This includes guidance on Israeli settlements . We are advising British businesses to bear in mind the British Government ' s view on the illeg ality of settlements under international law when considering their investments and activities in the region . This is voluntary guidance to British businesses on doing business in Israel and OPT s . Ultimately it will be the decision of an individual or company whether to operate in settlements in the Occupied Territories , but the British Government would neither encourage nor offer support to such activity . When approached by businesses , we set out the UK ' s clear position on

**Text ID: 9248 (Max Activation: 13.382)**

the official said . Manchester United goalkeeper Sam Johnstone is poised to rejoin Aston Villa on Monday . The 2 4 - year - old has been a target for a number of Championship and Premier League sides but will sign for Villa on a season - long loan . John stone , who spent the second half of last season on loan at Villa Park , still has another year left on his contract at United after this one . Manchester United goalkeeper Sam Johnstone is poised to rejoin Aston Villa on Monday He has been back in training at United ' s Carrington complex and will not be part of the tour party that travels to the USA on

**Text ID: 7155 (Max Activation: 13.311)**

9 8 0 s , Evergreen transported U . S . troops on drug raids in Central and South America , the paper said . Over the years , company officials denied working for the CIA . When contacted Wednesday by The Providence Journal to discuss Evergreen ' s relationship with the CIA , a spokesman for the spy agency declined to comment . The company also had contracts to carry U . S . Mail , as well as transporting cargo and personnel for private businesses . Ever green filed for liquidation under Chapter 7 of the U . S . Bankruptcy Code on Dec . 3 1 , 2 0 1 3 , in the Delaware District .

Figure 35: Top-activating examples for a feature in **SAE** annotated as **"conceptual"**. The specific annotation was "name of country and region" in various contexts.

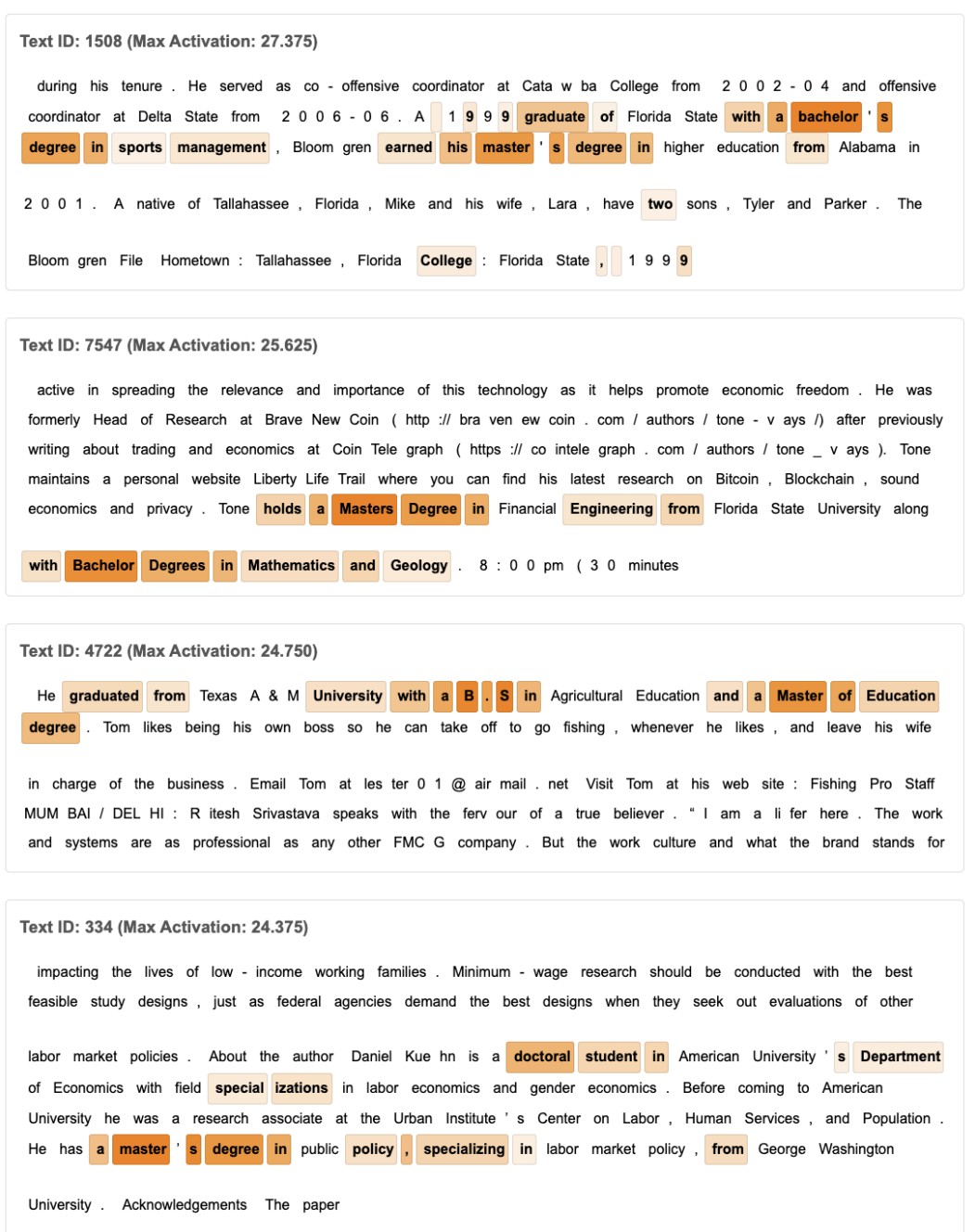

Figure 36: Top-activating examples for a feature in **Transcoder** annotated as **"conceptual"**. The specific annotation was "concept related to college degrees" in various contexts.

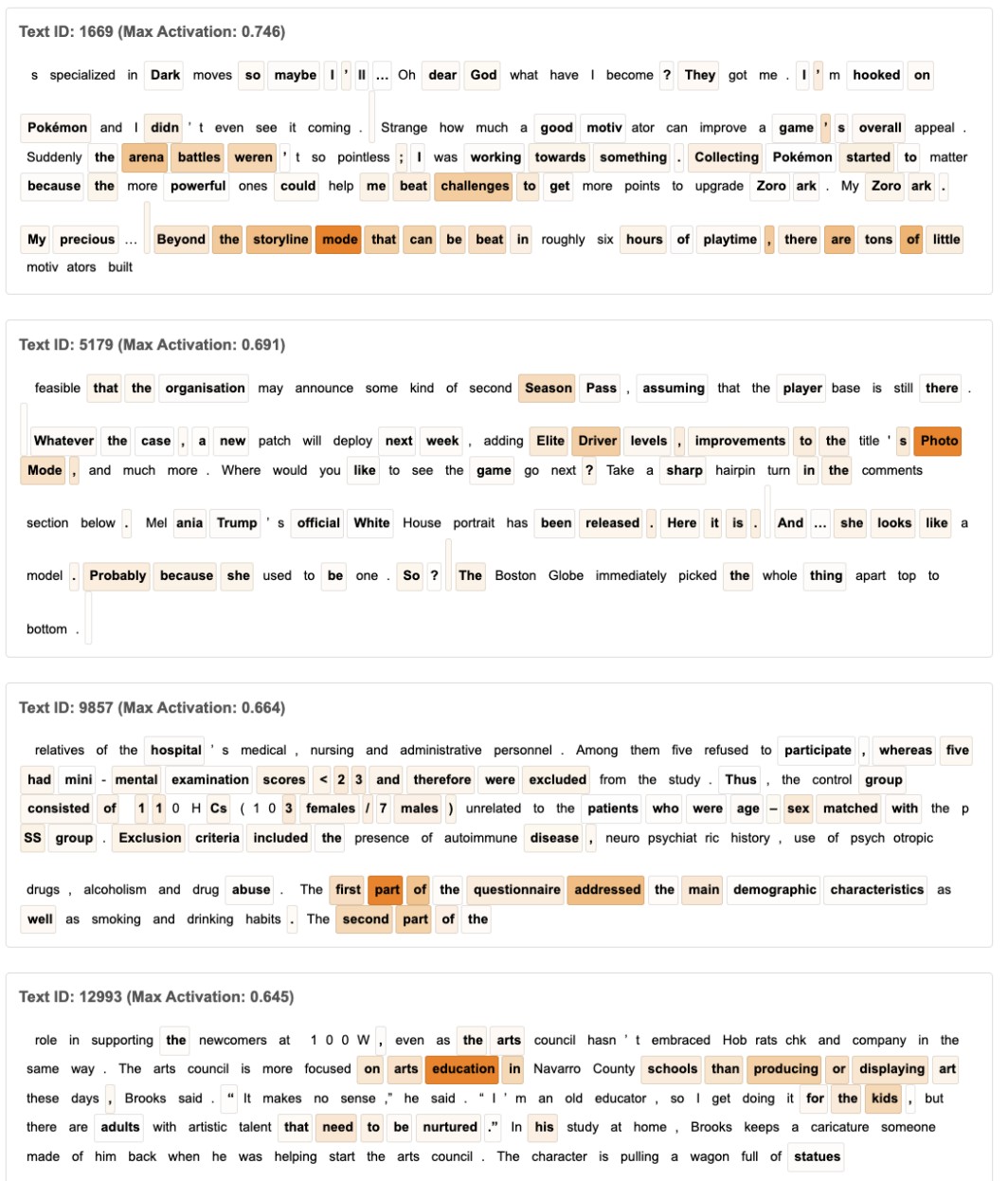

Figure 37: Top-activating examples for a feature in **FF-KV** annotated as **"Uninterpretable"**.

**Text ID: 7567 (Max Activation: 0.676)**

). I suggest that this filter plays a significant role in explaining the different patterns of deference exhibited by conservatives and liberals . The disposition to defer to others with whom we share a political or religious outlook is continuous with the disposition to use benevolence as a cue to reliability ; we are disposed to see those with **whom** we share a political outlook and / or a religious affiliation as those who are benevolent toward us and our interests ( or , perhaps , the disposition to use benevolence as a cue to reliability is just a special case of a disposition to defer to those with **whom** we share values ) . The use of political and religious affiliation as a proxy for benevolence is

**Text ID: 7727 (Max Activation: 0.609)**

, I don ' t even know who Thomas Mars is and I never have the phone records ," Robertson said . " I never find that call ." Tens of thousands of Roman ians and Bulg arians have come to the UK to work since restrictions were

lifted Peter Nicholls / The Times Net migration has reached a record 3 3 6 , 0 0 0 as figures published yesterday

showed that Roman ians are now the **third** biggest **group** coming to the UK . Officials said the latest increase was driven by a " stat istically significant " rise in the overall number of immigrants , with many of them arriving to take up jobs . The surge was partly because

**Text ID: 8655 (Max Activation: 0.605)**

Iran threat . Check out the full segment below . He is now one of five active coaches **in** the ACC **and** one of

1 5 active coaches in DI to accomplish the feat . R ALE IGH , N . C . – NC State baseball head coach Elliott Avent recorded his 1 , 0 0 0 th career win against North Carolina Central Tuesday evening , as the Wolf pack hit a

season - high four home runs en route to a 1 3 - 0 win at Do ak Field at D ail Park . Now in his 2 8 th season as a head coach and 2 1 st at NC State , Avent

**Text ID: 5332 (Max Activation: 0.605)**

. They reported that Einstein was right . Since then , his theory has been ret ouched in detail , but its essentials have been repeatedly verified . No important scientist is to be found among the skep tics , although there is every incentive to deb unk Einstein , if it can be done . Immort ality awaits the man who can overthrow Einstein . The popular uproar over the theory surprised no one **more** than the author of the theory . He had been almost a reclu se . His contacts had been with quiet , scholarly men of his own type , and his sudden glory appalled him . Interview ers , photographers , lion - hunters , cause - prom oters , testimonial

Figure 38: Top-activating examples for a feature in $k$-**Sparse FF-KV** annotated as **"Uninterpretable"**.

**Text ID: 15344 (Max Activation: 4.632)**

Aaron  Gle eman  of  Hard ball  Talk  )  reports  C  .  K  .  purchased  an  East  End  mansion  that  Babe  Ruth  spent  time  at  .

Ke il  reports  the  comedian  shelled  out  **$**  2  .  4  9  million  **for**  the  4  ,  9  5  7  -  square  foot  "  Prim rose  Cottage  "  formerly  visited  **by**  the  New  York  Yankees  legend  .  The  home  is  a  three  -  story  Tudor  originally  constructed  in  1  9  0  1  with  six  bedrooms  ,  three  -  and  -  a  -  half  **baths**  and  five  fireplaces  **.**  Yes  ,  there  are  more  fireplaces  than  bathrooms  in  this  home  ,  which  is  probably  a  zoning  requirement  in  the  Ham ptons  (  or  one  of  Ruth  '  s  ecc entri

**Text ID: 12336 (Max Activation: 4.299)**

the  new  ones  .  Additionally  ,  ssh  server  key  theft  is  another  one  -  time  vector  that  can  be  used  to  quickly  bootstrap  into  node  key  theft  .  For  this  reason  ,  node  admins  should  always  use  ssh  key  auth  for  tor  node  administration  accounts  ,  since  it  prevents  ssh  server  key  theft  from  implying  continuous  server  compromise  :  http  ://  www  .  gre m well  .  com  /  ssh  -  mit m  -  public  -  key  -  authentication  Issues  With  Ephe meral  Identity  Keys  There  are  a  few  issues  with  deploying  ephemeral  identity  keys  .  Issues  With  Ephe meral  Identity  **Keys**  :  Client  guard  node  loss  The  primary  issue  with  ephemeral  identity  keys  is  client  Guard  node  loss  .  If  your  relay  obtains  the  Guard  flag

**Text ID: 8158 (Max Activation: 4.289)**

given  Trump  '  s  sharp  criticism  and  talk  of  "  re visiting  "  the  Iran  nuclear  deal  .  The  message  is  loud  and  clear  :

The  United  States  cannot  be  trusted  .  Third  ,  the  U  .  S  .-  South  Korean  alliance  is  not  impenetrable  .  President  Trump  tweeted  his  criticism  about  the  South  Korea  -  United  States  free  trade  agreement  around  the  time  of  the  4  th  nuclear  test  and  accused  the  South  Korean  government  of  "  app ea sement  with  North  Korea  ."  South  Korea  is  finding  ,  as  I  have  told  them  ,  that  their  talk  of  appea sement  with  North  Korea  will  not  work  ,  they  only  understand  one  thing  !  —  **Donald**  J  .  Trump  (@  real  Donald  Trump

**Text ID: 4557 (Max Activation: 4.193)**

specifically  does  is  somewhat  meaningless  and  arbitrary  .  struct  Table  Based  Sorter  {  Table  Based  Sorter  (  const  int  values  [  1  2  8  ])  {  for  (  int  i  =  0  ;  i  <  1  2  8  ;  ++  i  )  m  Table  [  i  ]  =  ((  values  [  i  ]  ^  0  xff  8  0  **)**  +  1  2  8  )  -  i  ;  }  bool  operator  ()(  int  a  ,  int  b  )  const  {  return  m  **Table**  [  b  ]  <  m  Table  [  a  ];  }  int  m  Table  [  1  2  8  ];  };

std  ::  sort  (  v  ,  v  +  1  2  8  ,  Table  Based  Sorter  (  values  ));  The

Figure 39: Top-activating examples for a feature in **SAE** annotated as **"Uninterpretable"**.

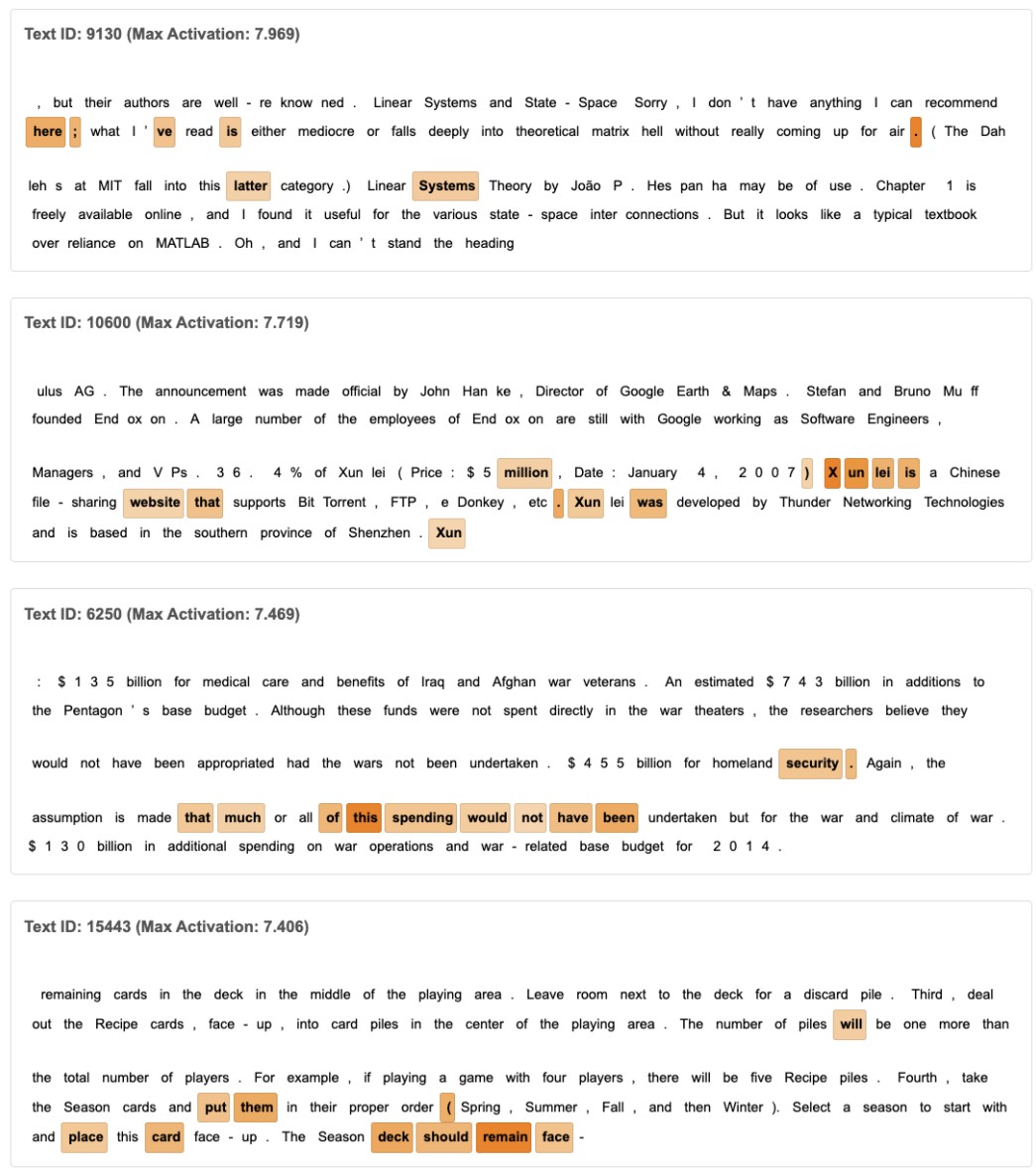

Figure 40: Top-activating examples for a feature in **Transcoder** annotated as **"Uninterpretable"**.

FF-KV  Transcoder

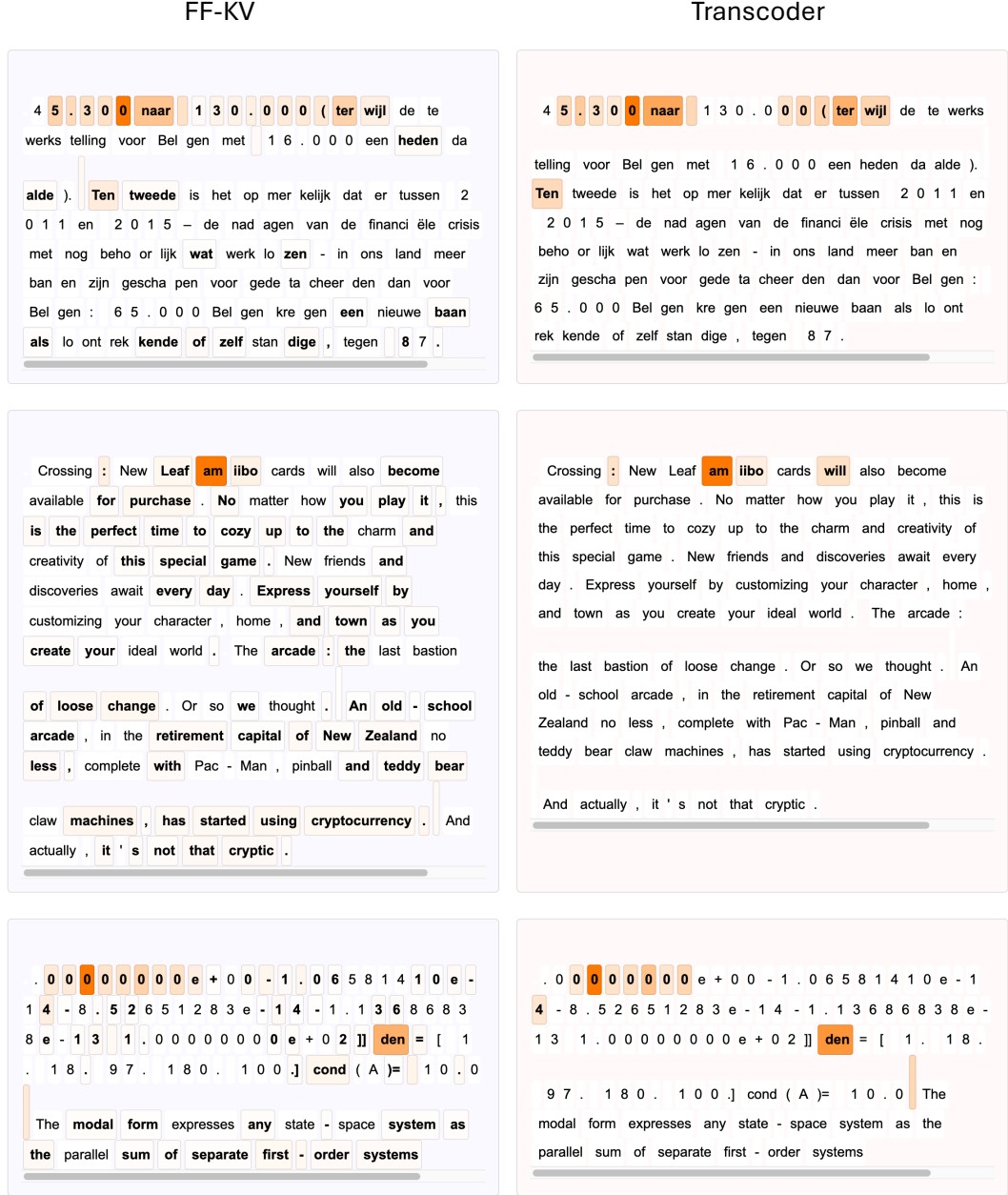

Figure 41: The first feature pair we annotate as **aligned**.

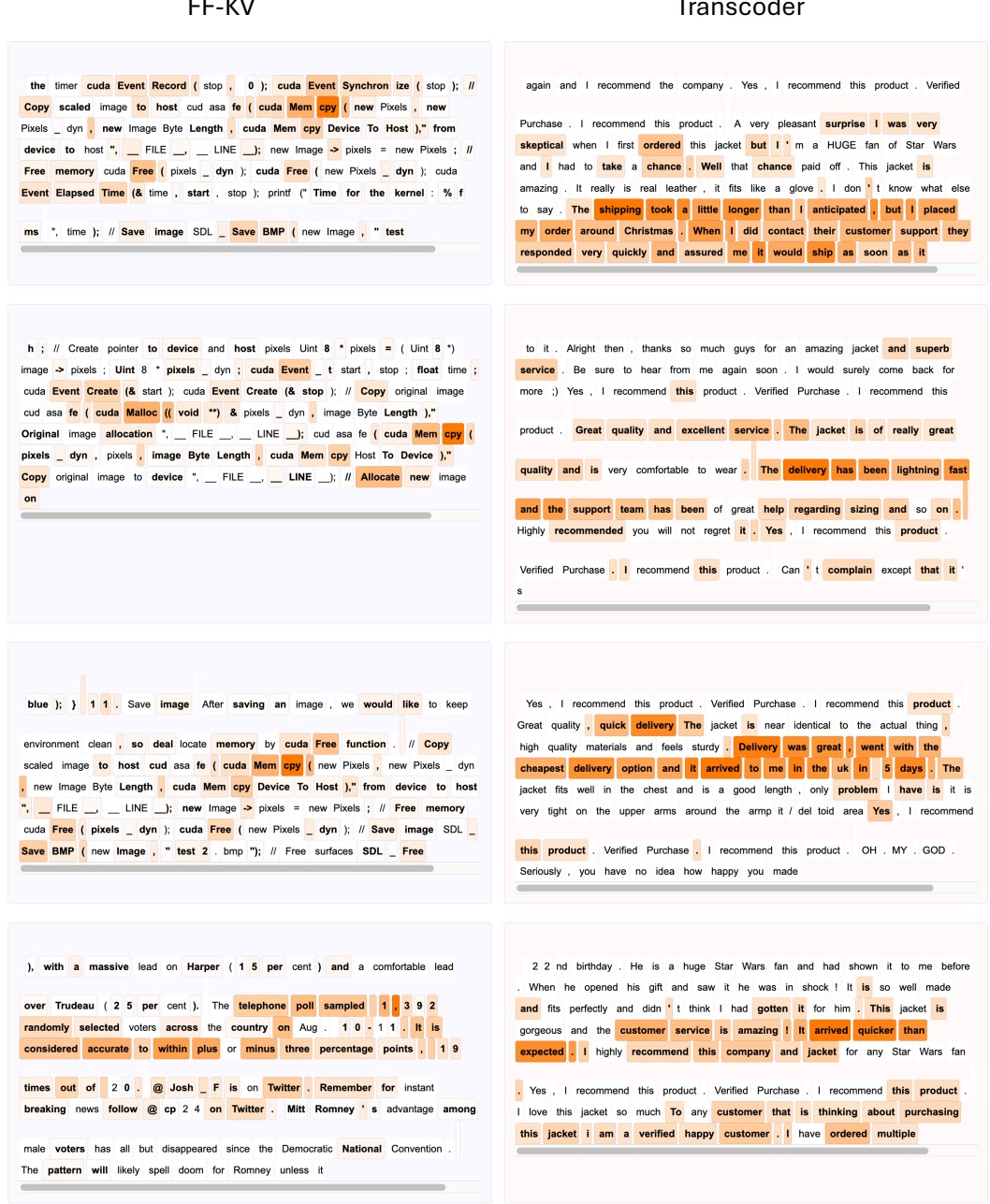

Figure 42: The first feature pair we annotate as **un-aligned**.

