# OpenReview forum: "Transformer Key-Value Memories Are Nearly as Interpretable as Sparse Autoencoders"
_NeurIPS.cc/2025/Conference — NeurIPS 2025 poster_

### Official Review · Reviewer_ToPx · 2025-06-24

**Clarity:** 3
**Significance:** 2
**Originality:** 3
**Rating:** 4
**Confidence:** 4

**Summary:**

Inspired by previous work viewing transfomer MLP layers as key-value memories, this paper explores evaluating transformer MLP layers as if they are an SAE using SAE metrics like SAEBench. This paper finds that the MLP layers already function similarly to SAEs on a number of SAE metrics, showing that transformer MLP layers do a decent job of learning interpretable features compared with SAEs.

**Questions:**

- The plots in Figure 1, containing the core results of the paper, are very compressed vertically making the results hard to read. These plots are used to justify the claim that using the MLP as an SAE is roughly as interpretable as using a real SAE, but this claim is hard to verify by looking at the plots due to them being so small.
- In Figure 1, the SCR results seem to go very negative - is this expected? What does negative SCR mean?
- What L0 and width are used for SAEs and the MLP? The paper says the MLPs are already sparse, but how sparse? Comparing SAES with very different L0 does not feel like a fair comparison in general.
- When trying to interpret the MLP layer directly as an SAE (not using top-k), it's possible for neurons to have a negative value since the non-linearity is typically not a ReLU, and typical activation functions (GeLU, SwiGLU, etc...) have negative components as well. How are these negative values handled? Negative latent values are not valid for SAEs so it seems hard to know how to compare the results to SAEs, or if SAEBench results will make sense in this case.
- It seems like there are two possible conclusions to draw from this work. Either (1) SAEs were a mistake and the field should return to directly trying to interpret neurons, or (2) SAEs aren't good enough and need to be improved. Which of these conclusions is the paper arguing for? I can see the value of using MLPs as a baseline to compare SAEs and other unsupervised techniques against, but I'm not convinced that directly interpreting neurons is a good direction.

**Ethical Concerns:**

["NO or VERY MINOR ethics concerns only"]

**Final Justification:**

Framing for this work as a baseline to judge future SAEs / interpretability methods against rather than advocating a return to trying to interpret neurons makes sense to me. Assuming this is emphasized in the final version of the paper, and the presentation issues mentioned are also fixed I will raise my score to 4.

**Limitations:**

yes

**Paper Formatting Concerns:**

no concerns

**Quality:**

2

**Strengths And Weaknesses:**

### Strengths

This paper identifies an interesting connection between transformer MLP layers and SAEs, and directly treating the MLP layer as an SAE is an interesting idea. This also has the benefit of being guaranteed to be faithful to the model, and requires no training as it's just a part of the model.

Also, although not discussed in the paper, this seems like it presents a good baseline to compare SAEs and other unsupervised interpretability methods against.

### Weaknesses

The paper does not discuss what feels like a core downside of this method, which is that the inner expansion factor of the MLP layers is typically not very large in transformer LLMs, and this means there are not many concepts that can be represented by the MLP when naively using the MLP neurons as SAE latents. The LLM can certainly represent more distinct concepts than there are MLP neurons, so using the MLP as an SAE feels fundamentally limited.

The paper also does not discuss the core motivation for SAEs, which is neuron polysemanticity. Directly interpreting neurons was one of the first things tried by interpretability researchers, but this was abandoned after it became clear that most neurons are polysemantic (meaning they fire on many different, seemingly unrelated concepts). SAEs are designed explicitly do deal with this polysemanticity. Given this paper feels like it's arguing for a return to a technique that's already been abandoned by the community, I suspect it will not have a large impact on the field.

---

> ### Author Rebuttal · Authors · 2025-07-31
>
> We appreciate constructive feedback.
>
> > Downside of FF-KV and motivation of SAEs (Weakness 1 & 2)
>
> We agree that a key motivation for SAEs is to disentangle more distinct concepts than the limited number of features permitted in MLP neurons. We agree with this theoretical advantage, and our experiments may suggest that, empirically, current SAEs may not disentangle concepts as much as expected, and the provided interpretability is nearly the same as the original MLP representations. More generally speaking, the advantage of SAE over conventional interpretability approaches, such as linear probe, has been debated, and our results will contribute to and encourage the estimation of SAE’s actual advantage.
>
> > Regarding Figure 1 (Question 1)
>
> We apologise for Figure 1’s poor readability. Please see our reply to Reviewer euvs for the table replacement and our planned visual update. Note that a more detailed breakdown of Figure 1 is shown in Appendix (but we admit the room for presentation improvement)
>
> > Negative SCR scores (Question 2)
>
> SCR can indeed be negative: a negative value simply means that using SAE features **reduces** performance relative to raw activations. Negative SCR values also appear in the SAEBench paper [1] (See their Fig. 7). In our updated table the layer-averaged SCR is non-negative; the negative points in the original figure were outliers.
>
> [1] Karvonen et al., SAEBench: A Comprehensive Benchmark for Sparse Autoencoders in Language Model Interpretability.
>
> > Sparsity of MLPs and choice of L₀ (Question 3)
>
> Reference [2] shows that MLP activations are already sparse. Appendix E lists the L0 values for all SAEs; following the Gemma-Scope recommendation, we set $L_0 \approx 100$ for every model, so sparsity levels are comparable.
>
> [2] Z. Li, C. You, S. Bhojanapalli, D. Li, A. S. Rawat, S. J. Reddi, K. Ye, F. Chern, F. Yu, R. Guo, and S. Kumar. The lazy neuron phenomenon: On emergence of activation sparsity in transformers. In The Eleventh International Conference on Learning Representations, 2023
>
> > Handling negative activations (Question 4)>
>
> Please see our response to Reviewer hRu1 for a detailed explanation of how we treat negative activations.
>
> > Conclusion of our paper (Question 5)
>
> We align with conclusion (2): SAEs themselves are not a mistake, nor do we advocate reverting to direct neuron interpretation. Rather, our findings suggest that current SAE implementations fall short of their goals, motivating both improved SAEs and new methods to achieve better conceptual disentanglement.

---

> > ### Comment · Reviewer_ToPx · 2025-08-01
> >
> > Thank you for this response. Framing for this work as a baseline to judge future SAEs / interpretability methods against rather than advocating a return to trying to interpret neurons makes sense to me. Assuming this is emphasized in the final version of the paper, and the presentation issues mentioned are also fixed I will raise my score to 4.

---

### Official Review · Reviewer_Efr6 · 2025-07-01

**Clarity:** 4
**Significance:** 4
**Originality:** 2
**Rating:** 5
**Confidence:** 4

**Summary:**

This work is a comprehensive comparison of the interpretability of SAEs and feed-forward neural networks, using a range of quantitative and qualitative (human) experiments. They find that SAEs are no better than FFs in most situations.

**Questions:**

n/a

**Ethical Concerns:**

["NO or VERY MINOR ethics concerns only"]

**Final Justification:**

After having read the other reviews and rebuttals, I'll keep my score as is.

**Limitations:**

yes

**Quality:**

3

**Strengths And Weaknesses:**

**Strengths**

* I was really excited to see this paper, as mechanistic interpretability is often critiqued for not having robust enough comparisons and evaluations. This work presents a clear side-by-side comparison, with rigorous qualitative and quantitative experiments.
* The authors compare across a reasonable number of model architectures and evaluations.
* The overall writing is really clear and easy to understand. For example, the color highlighting in the equations to match the description in the paper is great. The framing in the introduction of “bottom up vs top down” was nice, and generally I had no problems understanding what were actually pretty technically complex concepts.

**Weaknesses**

* I don’t have any significant weaknesses to report, but here are some readability suggestions and questions.
* There were a few confusingly-worded sentences, e.g.:
  * **“**Our study, in a sense, encourages the use of inherently black-box, neural LLMs with skipping the interpretability issue, as we claimed their FFs are, to some degree, interpretable”
  * “To mention differences among the methods, first, FF-KV methods can, by definition, achieve a perfect explained variance (reconstruction loss), and SAEs can not.”
* This is minor, but in the abstract, the authors claim: “how much better features such proxies have indeed discovered, especially compared to those already represented within the original model parameters — and unfortunately, such a comparison has little been made so far”. I agree that this line of work is extremely important and probably understudied, but I think it’s a stretch to say that it’s not being studied at all. For example, [https://arxiv.org/pdf/2502.16681](https://arxiv.org/pdf/2502.16681) (“Are sparse autoenencoders useful? a case study in sparse probing,”--  this paper is in your references) and [https://arxiv.org/pdf/2405.08366](https://arxiv.org/pdf/2405.08366) (“Towards Principled Evaluations of Sparse Autoencoders for Interpretability and Control” are along these lines. There’s also this thread by Neel Nanda: [https://www.lesswrong.com/posts/4uXCAJNuPKtKBsi28/negative-results-for-saes-on-downstream-tasks](https://www.lesswrong.com/posts/4uXCAJNuPKtKBsi28/negative-results-for-saes-on-downstream-tasks) (“Negative Results for SAEs On Downstream Tasks and Deprioritising SAE Research”) which isn’t peer reviewed but still relevant.
* The length of the appendix (\>50 pages) is excessive
* **“**Absorption Score evaluates how many features a particular simple concept (e.g., word starting with “S”) is split into. A higher value implies that many features are needed to emulate the targeted single concept, and thus, the feature set is redundant.” I’m confused about this, as often features are hierarchical. For example, what if the “S” concept is composed of “words starting with ST”,  “words starting with SL” and “other words starting with S”? That is, what if a concept really can be decomposed into multiple other concepts? I guess my underlying question (often brought up with SAEs/interpretability), how do you define a “simple concept”?
* Figure 1 is hard to read, as the lines are a bit pale and look similar.
* “Feature Origin Judgment.” → I’m a little confused about this. Are the authors guessing whether the feature came from an SAE vs FF-KV? If so, what does this tell us about the model interpretability?

---

> ### Author Rebuttal · Authors · 2025-07-31
>
> We thank the reviewer for recognizing our work’s contribution. We greatly appreciate these suggestions and will incorporate them in the next revision. Our answers to the questions appear inline below:
>
> >How do you define a “simple concept”?
>
> Our example in the feature‑absorption evaluation merely illustrates that a concept can appear at different levels of abstraction, following existing metrics. In practice, we believe that researchers should choose the appropriate level of absorption, depending on specific research questions.
>
> >What does “Feature Origin Judgment” tell us about the model interpretability?
>
> This experiment addresses the question: “Do SAEs produce activation patterns similar to FF-KVs qualitatively?” Our human annotators could not reliably distinguish top‑K FF‑KVs from SAEs or Transcoders, suggesting that SAEs and Transcoders do not provide fundamentally clearer activation patterns compared to FF‑KVs, supporting that FF-KV is nearly interpretable as SAEs/Transcoders.

---

### Official Review · Reviewer_euvs · 2025-07-07

**Clarity:** 3
**Significance:** 2
**Originality:** 2
**Rating:** 4
**Confidence:** 4

**Summary:**

This paper conducts a thorough comparison between the interpretability(and other features) of Sparse Autoencoder latents vs original neurons in the MLP layers of a transformer model. To make the comparison more fair, they introduce k-sparse and normalized k-sparse FF-KV for sparser activations. They find that the differences between SAEs and MLP neurons are smaller than previously thought, and claim that training an SAE might be unnecessary as existing neurons can have comparable interpretability.

**Questions:**

- Previous work on SAEs often compared interpretability against existing neurons and found very significant improvements, for example Anthropic work: https://transformer-circuits.pub/2023/monosemantic-features/index.html#global-analysis-interp . This seems to contradict the findings of this paper. Why do you think this is the case?

**Ethical Concerns:**

["NO or VERY MINOR ethics concerns only"]

**Final Justification:**

The rebuttal has clarified my understanding of the paper and I think seeing the table representation of the results is very helpful. While I still think the contributions of the paper are limited, I lean towards acceptance as the paper is well written and cleanly focuses on a particular question, so I will increase my score to 4.

**Limitations:**

Yes

**Quality:**

3

**Strengths And Weaknesses:**

Strengths:
- Very clearly written
- Adresses an interesting question with extensive experiments
- Some interesting and unexpected findings

Weaknesses:
- Fig 1 needs to be improved, hard to tell different lines apart. Partially might be a problem of plotting too many things in the same graph but clarity could also be improved by zooming in on some plots and choosing colors with more contrast, in particular I can’t tell Transcoder and Normalized-FF-KV apart at all. Markers for actual data points would also be helpful.
    - These results might be better presented as a table, I think a table of the average scores over layers of the different methods would be very useful for a reader.
    - Expanded tables in appendix are helpful but still suffer from some issues with contrast and seeing where the data points are
    - Seems like not all plots have all the methods? For example, where is transcoder in explained variance plots?
- I think paper is slightly overclaiming at several points, for example the autointerp scores are significantly higher for SAE methods but this is mostly considered a not significant difference
    - Non top-k FF-KV methods have even lower scores, could use more discussion on the significance of this difference. What is the random chance score on this evaluation? How far are these scores from that?
- Similarly, the top-k FF-KV methods have very poor explained variance compared to SAE-based methods but this is not discussed at all in the paper.
    - This is similar to Anthropic’s findings that thresholding neurons is not competitive with SAEs in terms of sparsity/reconstruction quality (See https://transformer-circuits.pub/2025/attribution-graphs/methods.html, Fig. 3)
- Human evaluation is very small scale and relies a lot on author judgement
- Limited originality. The paper mostly applies existing evaluations pipelines on existing models, though normalized k-sparse FF-KV is a neat idea.

---

> ### Author Rebuttal · Authors · 2025-07-31
>
> We thank the reviewer for the insightful comments.
>
> >Detailed plan for complete update of Figure 1 (Weakness 1)
>
> We will replace Figure 1 with tables with exact scores — sorry for the limited readability. Below, we showcase the Gemma‑2‑2B layer‑averaged results. Still, such averaged scores overlook layer-wise detailed changes; in the next version, we will also include additional figures in Appendix: one plot per coder, task, and model to improve legibility.
>
> | Model      | Coder                 | Feat. Alive $\uparrow$ | Explained Var. $\uparrow$ | Absorption $\downarrow$ | Sparse Probing $\uparrow$ | Autointerp $\uparrow$ | RAVEL-ISO $\uparrow$ | RAVEL-CAU $\uparrow$ | SCR (K=20) $\uparrow$ |
> |------------|-----------------------|-------------|----------------|------------|----------------|------------|-----------|-----------|------------|
> | Gemma-2-2B | SAE                   | 1.00        | 0.78           | 0.04       | 0.86           | 0.81       | 0.95      | 0.48      | 0.18       |
> |            | Transcoder            | 1.00        | 0.75           | 0.03       | 0.87           | 0.82       | 0.89      | 0.46      | 0.15       |
> |            | FF-KV                 | 1.00        | 1.00           | 3.57e-04   | 0.84           | 0.74       | 0.90      | 0.47      | 0.05       |
> |            | Normalized FF-KV      | 1.00        | 1.00           | 3.51e-04   | 0.84           | 0.74       | 0.90      | 0.47      | 0.05       |
> |            | TopK-FF-KV            | 0.99        | 0.08           | 1.38e-04   | 0.78           | 0.81       | 0.89      | 0.46      | 0.07       |
> |            | Normalized TopK-FF-KV | 0.99        | 0.08           | 1.41e-04   | 0.78           | 0.81       | 0.89      | 0.46      | 0.07       |
> |            | Random Transformer    | -           | -              | -        | 0.80           | 0.68       | -         | -         | 4.21e-03   |
>
> Here, we leave out some evaluation scores for random transformer, where the metric does not make sense or shows immediately bad scores (Feat. Alive, Explained Var. , RAVEL).
> Note that a detailed breakdown of Figure 1 is shown in Appendix (but we admit the room for presentation improvement)
>
>
> >Missing Transcoder in explained‑variance plots (Weakness 1)
>
> We apologize for this confusion. In Figure 5, the GPT‑2 transcoder is omitted because its explained variance is far below the plot range. The explained variance is −14 807.46 with a standard deviation of 38 729.33, indicating that the GPT‑2 transcoder is poorly trained; its result should therefore be interpreted with caution. We will make this explicit in the text.
>
>
> >paper is slightly overclaiming at several points (Weakness 2)
>
> We will adjust or clarify the conclusions based on your concern. Our main claim is that SAEs do not provide a large interpretability advantage over the existing FF-KV view overall. This asserts that a strong, simple baseline is missing in the community, and also possibly highlights the limitations of the current SAE benchmark (if one expects the advantage of SAE).
> As you pointed out, the autointerp score is slightly lower in FF-KVs, but this is one facet of our evaluations, and entirely, we hope the above take-home message and the novelty of our findings generally hold. As for the chance score, we hope that our baseline of random Transformer already provides a good reference.
>
>
> >poor explained variance on Top‑K FF‑KVs and could use more discussion (Weakness 3)
>
> Explained variance is computed on the FFN output. Because Top‑K FF‑KV zeroes all but the k largest activations (e.g., k=5 for 8000 hidden states corresponds to 0.125% of neurons), its output inevitably diverges from the original, reducing explained variance.
>
> For a quick check, increasing k for Gemma2-2B (layer 13) easily alleviates this trend, as shown below:
>
>
> | k    | explained varience |
> |------|--------------------|
> | 1    | 0.06               |
> | 2    | 0.08               |
> | 5    | 0.12               |
> | 10   | 0.16               |
> | 20   | 0.20               |
> | 50   | 0.25               |
> | 100  | 0.31               |
> | 500  | 0.58               |
> | 1000 | 0.73               |
>
> We can easily extend such quick checks to all models and layers, but during author response, we did not have enough computational resources to run a sweep across all layers in all k settings.
>
> >Human evaluation is very small scale and relies a lot on author judgement (Weakness 4)
>
> We will try to increase the annotation scale for camera-ready, but it is not so easy as the task is labour‑intensive. Let us excuse that the current manual analysis scale (200 features in total) and the setting (handled by authors) already exceed or match existing works —for example, Anthropic[1] (168 features) and the Transcoder study[2] (100 features), both annotated by the authors.
>
> [1] Bricken et al.. Towards Monosemanticity: Decomposing Language Models With Dictionary Learning.
> [2] Dunefsky et al.. Transcoders Find Interpretable LLM Feature Circuits.
>
>
> >Limited originality (Weakness 5)
>
> The NeurIPS reviewer guidelines explicitly value studies that yield new insights by rigorously evaluating existing methods. Our work fits this category, and if the reviewer is concerned about the fact that this study does not propose a novel method, we respectfully disagree with the assessment of limited originality.
>
>
> > Possible explanation of the contradiction between this paper and the Anthropic work (Question 1)
>
> We believe the discrepancy arises from different sampling strategies. Anthropic uniformly partitions each feature’s activation range into equal bins (e.g. 0–10 %, … , 90–100 %) and samples evenly from all bins, thereby analysing low, medium, and high activations. Our study, by contrast, interprets each feature using its highest‑firing inputs, as is typically focused on in other metrics. This difference likely accounts for the contrasting findings.

---

> > ### Comment · Reviewer_euvs · 2025-08-07
> >
> > Thank you for the response. This has clarified my understanding of the paper and I think seeing the table representation of the results is very helpful. While I still think the contributions of the paper are limited, I lean towards acceptance as the paper is well written and cleanly focuses on a particular question, so I will increase my score to 4.

---

### Official Review · Reviewer_hRu1 · 2025-07-07

**Clarity:** 2
**Significance:** 2
**Originality:** 1
**Rating:** 3
**Confidence:** 4

**Summary:**

This paper compares the interpretability of SAE and Transcoder features to the interpretability of feedforward network neurons. It calls into question the conventional wisdom that SAE features are significantly more interpretable than raw neuron activations.

**Questions:**

- Prior work has run experiments very similar to the ones you run here, on the same models, and found different results. Do you have an explanation for why your results seem to differ from those of Paulo et al. (2024), Table A10? Could you run your auto-interp experiments with their open source delphi library to see if the results change?
- What would count as a significant difference in autointerp scores here? The figures are quite zoomed out and it is difficult to get a good sense of how different the scores really are.
- How do you apply your method to a model like Llama, which uses SwiGLU layers rather than FFNs? Strictly speaking SwiGLU is an entirely different type of module, not an activation function for an FFN, and in the Llama variant it features three separate learned matrices, along with an elementwise multiplication step. It's not clear that this type of layer is amenable to the key-value interpretation.

**Ethical Concerns:**

["NO or VERY MINOR ethics concerns only"]

**Limitations:**

yes

**Quality:**

2

**Strengths And Weaknesses:**

Strengths:
- Simple, timely, and important research question.

Weaknesses:
- It is unclear why the "key-value memory" lens is particularly relevant to the paper; it seems that one could simply refer to the units as neurons. It's also unclear if this lens applies to the models they tested.
- The autointerp results seem to conflict with existing reported results, but this discrepancy is not remarked upon or explained.

---

> ### Author Rebuttal · Authors · 2025-07-31
>
> We thank the reviewer for the insightful comments.
>
> >Why adopt the key–value memory lens? (Weakness 1)
>
> Thanks for a good clarification question. SAE research investigates both feature activations and the geometry of their feature vectors, whereas the term “neuron” typically refers to the activations (FF keys) alone. The natural counterpart of the SAE feature vectors in the FF module is their value vectors through the lens of key-value memory, and in this sense, key-value memory will be a more accurate scope corresponding to SAE research. For example, in Section 6, we analyzed the alignment of feature vectors in SAE and FF, which will be beyond the scope of just observing activations (neurons).
>
>
> > Applicability of the KV view to SwiGLU layers (Weakness 1 & Question 3)
>
> This is a good point, and there will be at least two dimensions to be discussed:
> A. Split‑gate factorisation (gate_proj, up_proj)
> B. The presence of negative activation values produced by SwiGLU and its impact on the experimental results
>
> For the concern (A), Zhong et al. (2025) [1]  show that a SwiGLU FFN can be written as a signed key‑value sum (their Eq. 29):
>
> \begin{align}
> \mathrm{FFN_{SwiGLU}}(x)=\sum_i v_i g_i (x) \mathrm{Swish}(k_i x)
> \end{align}
> where $g_i(x) \mathrm{Swish}$ is regarded as a key (activation) and v_i as a value vector.
>
> As you may be concerned, the $\mathrm{Swish}(x)$ can take negative values, in contrast to the original key-value memory analysis in vanilla FFs — the concern (B).
>
> Fortunately, the absolute scale of negative activation is typically very small (as it’s the approximation of ReLU). Our experiments rely on the top-k activations based on their magnitude, and we empirically confirmed that negative activations were not selected at all throughout the experiments, as their magnitude was almost zero. That is, the negative activations do not affect experimental results in this study.
>
> [1] Zhong, Shu, et al. "Understanding Transformer from the Perspective of Associative Memory." arXiv preprint arXiv:2505.19488 (2025).
>
> >The inconsistency in autointerp scores with previous work (Weakness 2, Question 1, Question 2)
>
> In the SAE fields, slightly different implementations (with different activation selection processes) are used for the auto-interpretaion task, and we adopted the implementation of Karvonen et al. [2] — this is the source of the discrepancy with Paulo et al. (2024) [3]. Our autointerp results yielded a similar score scale to [2] (see their Figure 5).  Note that the implementation of [3] introduced a specific threshold (1e-5) for sampling activation (w/o reasonable justification), while [2] applies a two-step process, first a threshold-free sampling process, and second a dynamic threshold (1% of the max activation); we hope that the latter (i.e., our implementation) can handle concerns, such as different activation scale for different features/models.
>
> [2] Karvonen et al., SAEBench: A Comprehensive Benchmark for Sparse Autoencoders
> in Language Model Interpretability. Forty-Second International Conference on Machine Learning.
>
> [3] Paulo, Gonçalo Santos, et al. Automatically Interpreting Millions of Features in Large Language Models. arXiv preprint, 	arXiv:2410.13928 (2024).
>
>
> >Improvement on Figure 1
>
> We apologize for the dense, small plots and will replace Figure 1 entirely, as all reviewers noted its limited readability. For a detailed revision plan, please see our response to Reviewer euvs. Note that a more detailed breakdown of Figure 1 is shown in Appendix (but we admit the room for presentation improvement)

---

> ### Author Response · Authors · 2025-08-07
>
> Dear Reviewer hRu1,
>
> Thank you once again for dedicating your valuable time to reviewing our paper. As the deadline approaches, we kindly request your feedback on our rebuttal.
>
> We have carefully addressed all your concerns in detail and hope that you find the response satisfactory.
> Particularly, similar concerns (negative activations in SwiGLU, slight score differences with existing studies, Figure 1) were also raised by other reviewers, and they indeed recognize that our response successfully handled these concerns and ultimately raise their review scores. Now, all the other reviewers lean toward the acceptance of this paper, and we hope you also might reconsider your score based on our response.
>
> We are eager to engage in further discussion and address any additional concerns you may have. We sincerely appreciate your constructive suggestions.
>
> Thank you!
> Best Regards, Authors

---

### Note · Authors · 2025-08-14

We thank the reviewers for their constructive feedback and engagement during the discussion period.
We addressed all concerns in the author rebuttal phase and believe all reviewers, except hRu1, now lean toward acceptance.

Let us report the irresponsibility of the reviewer hRu1 since this reviewer:
- did not participate in the discussion
- did not submit Mandatory Acknowledgement form [1]

While we offered an intensive response for the given comments even with a reminder, we could not communicate with this reviewer at all to reach agreement or promote any further discussion.

Given this, we ask ACs to critically re-assess hRu1’s review in light of our response.
Notably, engaged reviewers (euvs, Efr6, ToPx) raised similar concerns, and our response was so effective as it addressed their concerns, leading them to increasing scores.

Specifically, the weaknesses and questions raised by hRu1 are handled as follows:
- Motivation for the key-value memory lens — we clarified the correspondence between SAE feature vectors and FF value vectors.
- Negative activations in SwiGLU — existing work [2] shows theoretical compatibility of key-value-based analysis with SwiGLU, and our empirical evidence shows negligible effect on our results (reviewer ToPx).
- Differences in autointerp scores with prior work — due to implementation variations across studies, mainly in activation sampling (related to reviewer euvs). Using the same implementation, we reproduced similar scores in [3]. The study [4] mentioned by the reviewer adopted a heuristic activation threshold, unsuitable for comparing models with different activation scales.
- Figure 1 readability — we will replace it with clearer tables and per-task/per-coder plots. (all reviewers)

We appreciate your hard work and would like to see the consideration of our final remark.

[1] The email for reviewers on August 10th: "Please note you will be required to confirm (by pressing a "Mandatory Acknowledgement" button) ... before the end of reviewer-AC discussion (before Aug 13). Failure to do so will be flagged as an irresponsible behavior."

[2] Zhong, Shu, et al. "Understanding Transformer from the Perspective of Associative Memory." arXiv preprint arXiv:2505.19488 (2025).

[3] Karvonen et al., SAEBench: A Comprehensive Benchmark for Sparse Autoencoders in Language Model Interpretability. ICML 2025.

[4] Paulo, Gonçalo Santos, et al. Automatically Interpreting Millions of Features in Large Language Models. ICML 2025.

---

### Decision · Program_Chairs · 2025-09-17

**Decision:**

Accept (poster)

**Comment:**

This paper examines whether sparse autoencoders and Transcoders truly provide more interpretable features than the native feed-forward key–value structure in transformer MLPs. By adapting SAEBench metrics directly to FF-KV and its variants, the authors report that FF features perform comparably to SAE features across a range of interpretability tasks, with each side showing small but consistent advantages on particular metrics. A human study and alignment analysis further suggest that proxy features often overlap with, or fail to meaningfully improve on, FF features. The core contribution is to establish FF-KV as a strong baseline against which interpretability methods should be measured.

The strengths of the paper are the systematic analysis of mechanistic interpretability with systematic experiments across several model families. The paper establishes a fair evaluation framework that treats FF as an SAE and it presents useful insights about where SAEs add value and where they do not. The writing is clear, and the authors have provided extensive supplementary material. Reviewers found the idea of reframing FF layers as an interpretability baseline both simple and powerful, with strong potential to recalibrate community assumptions.

Weaknesses noted by reviewers (and with which I agree with) are the limited originality since the work repurposes existing evaluation pipelines rather than introducing new methodology and the modest scale and potential bias in the human evaluation. Some reviewers also point out the  over claims in the framing, particularly regarding whether SAEs are unnecessary.  There were also questions about fairness of capacity comparisons and how negative activations in SwiGLU layers were handled, and finally discrepancies with prior auto-interpretation results. Some presentation issues in the results figures are also brought up by the reviews.

The rebuttal and discussion were constructive and helpful as the authors clarified the motivation for the key–value lens and provided theoretical and empirical justification for handling SwiGLU. The authors also satisfacorily explained implementation differences underlying the auto-interp discrepancies. They also committed to replacing confusing figures with tables and clearer plots. Reviewers euvs and ToPx found these clarifications satisfactory. Reviewer Efr6 was positive throughout, emphasizing the clarity and significance of the comparison. Reviewer hRu1 remained unconvinced and did not engage in discussion, but most of their concerns were directly addressed in the rebuttal and were similar to those raised by other reviewers who did change their stance.

While the paper does not propose a new interpretability method, I believe it does provides valuable empirical insight, showing that FF-KV is a surprisingly strong baseline, and that current SAE approaches may not deliver as much additional interpretability as assumed. The limitations around human evaluation scale and originality are real, but not enough to warrant rejecting the paper. The authors have also shown a willingness to address presentation and framing concerns, and the reviewers who engaged in discussion seemed to agree on acceptance, which I agree with.